# A Direct Approach for Handling Contextual Bandits with Latent State Dynamics

**Zhen Li** [* 1]   **Gilles Stoltz** [* 2 3]

## Abstract

We consider a linear contextual bandit model where contexts and rewards are governed by a finite hidden Markov chain. We first revisit the simplified model by Nelson et al. (2022), in which rewards are linear functions of the posterior probabilities over the hidden states given the observed contexts (called beliefs), rather than functions of the hidden states themselves. This simplified model may be handled through a direct reduction to standard linear contextual bandits. We extend the theoretical analysis of this reduction to take into account the estimation of the parameters of the hidden Markov model [HMM] in the regret bound and to provide high-probability bounds not depending anymore on the reward functions and only depending on the model through the estimation of the HMM parameters. Second, and most importantly, we instead study the more natural and more complex model incorporating direct dependencies in the hidden states (on top of dependencies on the observed contexts, as is natural for contextual bandits). Under a classic HMM forgetting condition, the main algorithmic tool introduced to cope with the various statistical dependencies that the reward structure introduces is to only periodically update reward-model parameters.

## 1. Introduction and Related Works

We consider a linear contextual bandit model where contexts and rewards are governed by a finite hidden Markov chain. Before we compare in detail our work to the earlier one by Nelson et al. (2022), we position the problem within the

---
[1]BNP Paribas Corporate and Institutional Banking, Paris, France [2]Université Paris-Saclay, CNRS, Inria, Laboratoire de mathématiques d'Orsay, 91405, Orsay, France [3]HEC Paris, Jouy-en-Josas, France. Correspondence to: Zhen Li <zhen.li@bnpparibas.com>, Gilles Stoltz <gilles.stoltz@universite-paris-saclay.fr>.

*Proceedings of the 43ʳᵈ International Conference on Machine Learning*, Seoul, South Korea. PMLR 306, 2026. Copyright 2026 by the author(s).

broader context of stochastic bandits, and more particularly, of stochastic bandits in changing environments.

In finitely-armed stochastic bandits (introduced by Thompson, 1933 and Robbins, 1952; see also the survey monograph by Lattimore & Szepesvári, 2020), rewards are drawn i.i.d. from fixed but unknown distributions indexed by arms, and the learner must perform some trade-off between exploration (to estimate the distributions) and exploitation (to pull more often better-performing arms). A first extension of interest is called linear contextual bandits (see Chu et al., 2011 and Abbasi-Yadkori et al., 2011, where the celebrated LinUCB strategy was introduced), where the learner observes a context (possibly chosen adversarially), selects an action, and receives a reward modeled as a linear function of (some function of) the context and action.

A second extension of interest is when this linear contextual model depends on some latent state changing over time. Two modelings and approaches were considered: first, some change-point detection approaches, relying on infrequent changes, with regret bounds typically functions of the root number of changes (see Wu et al., 2018 or Austin & Morgan, 2025); second, a modeling of the latent state as following some partially known dynamic, typically a Markov chain. Contexts are then assumed to follow a hidden Markov model [HMM]: they are drawn independently at random given the latent state. Much of this literature focuses on context-free reward models, where rewards depend on the actions and latent states but not on the observed contexts (when present, the latter are used only for state inference). For example, Azizzadenesheli et al. (2016) and Zhou et al. (2021) study such a setting and combine a LinUCB-style exploration with a spectral method for reward estimation. A common limitation due to the spectral method is to have to consider finite reward spaces, with Zhou et al. (2021) focusing, in particular, on binary rewards. In terms of guarantees, Azizzadenesheli et al. (2016) obtained an $\sqrt{T}$–high-probability regret bound against a memoryless policy benchmark, whereas Zhou et al. (2021) provided a $T^{2/3}$ regret bound against a stronger oracle that knows the true belief (the posterior distribution over latent states given past and present contexts) and the state-dependent expected rewards.

**Motivation.** We are interested in a general setting of linear contextual bandits in which the latent state governs the

context distribution and, jointly with context and action, determines the rewards. If one is ready to believe that classic linear contextual bandits form a setting of practical interest, then the extension considered here to some latent-state dynamic wishes to cover the cases where rewards are not functions of actions and contexts only. For instance, in economic problems (see our case study in Section 5), underlying economic states correspond to crises or growth periods and directly influence both the contexts and the rewards.

**Specific literature review.** To the best of our knowledge, only a few works have studied linear contextual bandits with a latent-state dynamic. Nelson et al. (2022) do so with a HMM modeling of contexts, but further assume that rewards are linear in (some function of) the action and in the belief, rather than being linear in (some function of) the action and the actual latent state. This is a seemingly harmless but actually major simplification of the problem, as we explain throughout this article. Nelson et al. (2022) propose Thompson-sampling and LinUCB-style algorithms and also provide some partial elements for a theoretical analysis (however, not discussing the estimation of HMM parameters). Finally, in their setting, rewards do not depend directly on the contexts, they do so only indirectly through beliefs.

Two recent works include Hong et al. (2020b) and Galozy et al. (2025): they evaluate their algorithms against a strong benchmark that knows the realized latent states and the latent-state-dependent reward-model parameters. However, their regret guarantees require a sublinear number of latent-state changes and degrade to linear when switches occur at a linear rate. Hong et al. (2020a), on the contrary, assumes that the latent state remains constant over time (and does not tackle the estimation of the reward-model parameters).

**Main Contributions; Comparison to Nelson et al. (2022)**

The contributions of this article are twofold: first, being able to obtain sublinear high-probability regret bounds in a complex model more challenging than existing models; second, achieving an elementary, more direct, and more efficient treatment of the simpler model by Nelson et al. (2022) as a special case of the methodological developments made to tackle the more general problem considered.

**Contribution 1: General model.** First, we introduce around Equation (1) a general setting of linear contextual bandits with latent-state dynamics, where expected rewards depend linearly on (functions of) the contexts and actions, as well as on unobserved states (that follow a HMM). The rewards are continuously-valued. We explain why the seemingly similar dependence of rewards in Nelson et al. (2022) on beliefs, rather than directly on states, is actually an important simplification of the model. Therefore, prior work

either assumed infrequent state changes, or considered reward models depending only on states and actions but not on contexts (with contexts only used for inference), or depending on beliefs and actions but not directly on states, or restricted the rewards to a finite space. None provided sublinear regret bounds in our setting where rewards depends jointly on states, contexts, and actions, where states may switch arbitrarily often, and where these rewards are continuously-valued.

Our solution relies on an extremely careful analysis of the statistical dependencies induced by the reward model, leading to a strategy proceeding in stages to carefully balance decent estimation of the reward-model parameters and of the beliefs and limited dependencies in the past.

**Contribution 2: Comparison.** We formally compare the belief-dependent linear reward model of Nelson et al. (2022) to the state-dependent model studied here, showing in that the former reduces to linear contextual bandits whereas the latter does not.

We further show in Appendix A that, under the belief-dependent linear reward model of Nelson et al. (2022), extended to action-context-belief-dependent rewards, the regret may be handled through a direct LinUCB-like analysis, up to an extra belief error term. Namely, we obtain a high-probability regret bound of order $T^{3/4}$, where the extra $T^{1/4}$ factor is shown to only come from belief estimation. The analysis proposed in Appendix A is simple (much simpler than in the original reference) and directly exploits the reduction to linear contextual bandits.

We also discuss in Section 4.1 how the obtained regret bound is sharper and more general than the one by Nelson et al. (2022): in particular, it holds with high probability, not only in expectation, and is reward-model free.

**Outline**

Section 2 introduces the setting and the general model of HMM-generated contextual bandits with rewards being state-dependent linear functions of the contexts and actions, as well as the simplified version considered by Nelson et al. (2022). In particular, this section introduces the corresponding notions of regret.

Because of technicalities discussed in detail in Appendix B, including intricate dependencies on observed quantities to the hidden states, we resort to a staged algorithm. Namely, a staged LinUCB-like strategy is formally stated in Section 3; it relies on belief-estimation subroutines, for which reminders are provided in Appendix C. A special case of this strategy, when stages only contain a single round, is able to handle the simplified model of Nelson et al. (2022).

Section 4 states and provides sketches of proofs of the regret

bounds of the strategies considered: a $T^{3/4}$ regret bound in the simplified model by Nelson et al. (2022), with full details in Appendix A, and a $T^{7/8}$ regret bound in the general model, with full details provided in Appendix E. The second regret bound relies on forgetting properties of HMMs, for which reminders are provided in Appendix D.

Section 5 and Appendix F provide some numerical simulations, with the mere aim to illustrate the theory developed.

## 2. Settings, Notation, and Regret Definitions

We first describe the considered finite-armed contextual bandit setting with latent-state dynamic and then state two versions of the reward functions: our own, more complex, version and the original, simplified, version by Nelson et al. (2022). After highlighting some issues arising from the statistical dependencies at stake, we discuss two notions of regret: the same notion of pseudo-regret as in Nelson et al. (2022), and regret in terms of actual rewards.

**Notation.** The short-hand $\boldsymbol{x}_{s:t}$ stands for the sequence $\boldsymbol{x}_s, \ldots, \boldsymbol{x}_t$. We let $[H] = \{1, \ldots, H\}$. The vectors $(1, \ldots, 1)$ with all elements equal to 1 are denoted, independently of the lengths, by $\mathbf{1}$. The identity matrix of size $s \times s$ is denoted by $\boldsymbol{I}_s$. We denote the tensor product of two vectors $\boldsymbol{u} = (u_h)_{h \in [H]} \in \mathbb{R}^H$ and $\boldsymbol{v} \in \mathbb{R}^d$ by $\boldsymbol{u} \otimes \boldsymbol{v} = (u_h \boldsymbol{v})_{h \in [H]} \in \mathbb{R}^{dH}$. The Euclidean norm is denoted by $\| \cdot \|_2$, and the $\ell^1$–norm by $\| \cdot \|_1$. For matrices $M$, the norm considered is the Frobenius norm, i.e., the Euclidean norm of the coefficients all written into a column vector; this is why we use the same notation $\|M\|_2$ for this matrix norm. The norm induced on $\mathbb{R}^s$ by a symmetric definite positive matrix $G$ of size $s \times s$ is defined by

$$\forall \boldsymbol{u} \in \mathbb{R}^s, \quad \|\boldsymbol{u}\|_G = \sqrt{\boldsymbol{u}^\top G \boldsymbol{u}} \,.$$

For two symmetric matrices $G, G'$, we write $G \succeq G'$ when $G - G'$ is a symmetric positive semi-definite matrix.

### 2.1. Latent Dynamic and Learning Protocol

We consider a finite-armed contextual bandit problem, with a finite action set $\mathcal{A}$ and with a $m$–dimensional context space $\mathcal{X} \subseteq \mathbb{R}^m$, equipped with the Borel $\sigma$–algebra. At each round $t \geqslant 1$, the learner observes some context $\boldsymbol{x}_t \in \mathcal{X}$, generated by a hidden Markov model [HMM].

**HMM modeling.** More formally, there exists an underlying state $h_t \in [H]$, where $[H]$ denotes the finite latent state space; this space $[H]$ is known to the learner. The first state $h_1$ is distributed according to some initial distribution denoted by $\boldsymbol{\pi}$. At each round $t \geqslant 1$, the context $\boldsymbol{x}_t$ is drawn independently at random given $h_t$, according to an emission distribution over $\mathcal{X}$ denoted by $\nu_{h_t}$. The next latent state $h_{t+1}$ is then drawn according to a Markov model indexed by the transition matrix $\boldsymbol{M} = (M_{h,h'})_{(h,h') \in [H]^2}$,

where $M_{h,h'}$ is the probability from moving from state $h$ to state $h'$.

The (homogeneous) HMM is thus parameterized by the initial distribution $\boldsymbol{\pi}$, the transition matrix $\boldsymbol{M}$, and the emission distributions $\nu = (\nu_h)_{h \in [H]}$, all unknown to the learner.

**Reward model—most complex one.** We consider the following linear model: there exist a known transfer function $\boldsymbol{\varphi} : \mathcal{A} \times \mathcal{X} \to \mathbb{R}^d$ and some unknown parameters $\boldsymbol{\theta}_h^\star \in \mathbb{R}^d$, where $h \in [H]$, such that the reward $r_t(a)$ obtained with action $a \in \mathcal{A}$ at round $t$ equals

$$r_t(a) = \boldsymbol{\varphi}(a, \boldsymbol{x}_t)^\top \boldsymbol{\theta}_{h_t}^\star + \eta_t(a) \,, \tag{1}$$

where $\eta_t(a)$ is a noise term, which we discuss below. It is handy to assume some boundedness.

**Assumption 2.1** (bounds on the reward model). There exists $C_{\boldsymbol{\theta}^\star} \in (0, +\infty)$ such that for all $a \in \mathcal{A}, \ \boldsymbol{x} \in \mathcal{X}, \ h \in [H]$,

$$|\boldsymbol{\varphi}(a, \boldsymbol{x})^\top \boldsymbol{\theta}_h^\star| \leqslant 1 \,, \quad \|\boldsymbol{\varphi}(a, \boldsymbol{x})\|_2 \leqslant 1 \,, \quad \|\boldsymbol{\theta}_h^\star\|_2 \leqslant C_{\boldsymbol{\theta}^\star} \,.$$

**Learning protocol and information available.** At each round $t \geqslant 1$, the learner observes the context $\boldsymbol{x}_t$ (but not the latent state $h_t$), picks an action $a_t \in \mathcal{A}$ based on $\boldsymbol{x}_t$ and on the information available from past rounds, and obtains and observes the reward $r_t(a_t)$, but not the rewards $r_t(a)$ for actions $a \neq a_t$.

The information available when picking $a_t$ consists therefore of the past and present contexts $(\boldsymbol{x}_\tau)_{1 \leqslant \tau \leqslant t}$ and of the past rewards $(r_\tau(a_\tau))_{1 \leqslant \tau \leqslant t-1}$. We denote the filtration generated by this information by

$$\mathcal{F}_t^{\text{obs}} = \sigma\Big( \big(\boldsymbol{x}_\tau, r_\tau(a_\tau)\big)_{\tau \leqslant t-1}, \ \boldsymbol{x}_t \Big)$$

(where "obs" stands for observed): the action $a_t$ is thus $\mathcal{F}_t^{\text{obs}}$–measurable.

**Assumptions on the noise term.** A classic assumption in linear bandits (e.g., Abbasi-Yadkori et al., 2011) on the noise terms $\eta_t(a)$ is that these terms are conditionally sub-Gaussian, see Assumption 2.2. It turns out that following Nelson et al. (2022), a milder assumption on conditional first and second moments may be enough, see Assumption 2.3. We will consider the second assumption for stating our main results, though the stronger Assumption 2.2 will be useful for discussions and comparison to prior results.

For both assumptions, conditionings are taken with respect to all priori random variables, whether they are observed or not: we consider the filtration

$$\mathcal{F}_t^{\text{all}} = \sigma\left( \Big( h_\tau, \boldsymbol{x}_\tau, \big(\eta_\tau(a)\big)_{a \in \mathcal{A}} \Big)_{\tau \leqslant t-1}, \ h_t, \boldsymbol{x}_t \right).$$

**Assumption 2.2** (conditionally sub-Gaussian noise). There exists $v_\eta$ such that for all $a \in \mathcal{A}$,

$$\mathbb{E}\big[e^{\lambda \eta_t(a)} \mid \mathcal{F}_t^{\text{all}}\big] \leqslant e^{\lambda^2 v_\eta^2/2} \,.$$

Note that this entails that $\mathbb{E}\big[\eta_t(a) \mid \mathcal{F}_t^{\mathrm{all}}\big] = 0$.

**Assumption 2.3** (Bounded conditional second-order moment)**.** There exists $C_\eta$ such that for all $a \in \mathcal{A}$,

$$\mathbb{E}\big[\eta_t(a) \mid \mathcal{F}_t^{\mathrm{all}}\big] = 0 \quad \text{and} \quad \mathbb{E}\big[\eta_t(a)^2 \mid \mathcal{F}_t^{\mathrm{all}}\big] \leqslant C_\eta \,.$$

### 2.2. The Simplified Model by Nelson et al. (2022)

Consider the beliefs (the posterior probabilities over the hidden states given the observed contexts)

$$\boldsymbol{b}_t : h \in [H] \longmapsto \boldsymbol{b}_t(h) = \mathbb{P}(h_t = h \mid \boldsymbol{x}_{1:t}) \,.$$

Nelson et al. (2022) consider the same latent dynamics and learning protocol as above but rather study the following reward model: there exist scalars $(\theta_{h,a}^\star)_{h \in [H], a \in \mathcal{A}}$ such that

$$r'_t(a) = \sum_{h \in [H]} \boldsymbol{b}_t(h) \, \theta_{h,a}^\star + \eta'_t(a) \,, \qquad (2)$$

where the noise terms $\eta'_t(a)$ satisfy Assumption 2.3.

We rather consider an immediate generalization where expected rewards can depend directly also on contexts and where general transfer functions are considered as in (1):

$$r'_t(a) = \sum_{h \in [H]} \boldsymbol{b}_t(h) \, \boldsymbol{\varphi}(a, \boldsymbol{x}_t)^\top \boldsymbol{\theta}_h^\star + \eta'_t(a) \,, \qquad (3)$$

The original model (2) corresponds to the special case where $\boldsymbol{\theta}_h^\star = (\theta_{h,a}^\star)_{a \in \mathcal{A}}$ and $\boldsymbol{\varphi}(a, \boldsymbol{x}_t) \in \{0,1\}^\mathcal{A}$ with the $a$–th component equal to 1 and all other components being null.

The difference between the model (1) we study in this article and the immediate generalization (2) of the model by Nelson et al. (2022) lies in replacing $\boldsymbol{\varphi}(a, \boldsymbol{x}_t)^\top \boldsymbol{\theta}_{h_t}^\star$ by

$$\mathbb{E}\big[\boldsymbol{\varphi}(a, \boldsymbol{x}_t)^\top \boldsymbol{\theta}_{h_t}^\star \mid \boldsymbol{x}_{1:t}\big] = \boldsymbol{\varphi}(a, \boldsymbol{x}_t)^\top \sum_{h \in [H]} \boldsymbol{b}_t(h) \, \boldsymbol{\theta}_h^\star \,.$$

This substitution looks harmless at first sight but has important consequences: the problem can be reduced to contextual bandits, as exploited by Nelson et al. (2022). Without this substitution, and when keeping the direct dependencies on the hidden states $h_t$, no such reduction holds and an improved analysis is required. We now detail these claims, as the technical discussions that follow will clarify why and how we consider two notions of regret in Section 2.3.

**Reduction of model (3) to linear contextual bandits.** Introduce

$$\mathcal{F}_t^{'\mathrm{obs}} = \sigma\Big(\big(\boldsymbol{x}_\tau, r'_\tau(a_\tau)\big)_{\tau \leqslant t-1}, \boldsymbol{x}_t\Big) \,.$$

The action $a_t$ picked in the model (3) is $\mathcal{F}_t^{'\mathrm{obs}}$–measurable. Also, the assumptions on the noise entail, by the tower rule, that for all $a \in \mathcal{A}$,

$$\mathbb{E}\big[\eta'_t(a) \mid \mathcal{F}_t^{'\mathrm{obs}}\big] = 0 \,, \quad \text{thus} \quad \mathbb{E}\big[\eta'_t(a_t) \mid \mathcal{F}_t^{'\mathrm{obs}}\big] = 0 \,.$$

Because of the specific form of the reward model (3), these equalities translate into:

$$\forall a \in \mathcal{A}, \quad \mathbb{E}\big[r'_t(a) \mid \mathcal{F}_t^{'\mathrm{obs}}\big] = \boldsymbol{\varphi}(a, \boldsymbol{x}_t)^\top \sum_{h \in [H]} \boldsymbol{b}_t(h) \, \boldsymbol{\theta}_h^\star$$

$$\text{and} \qquad \mathbb{E}\big[r'_t(a_t) \mid \mathcal{F}_t^{'\mathrm{obs}}\big] = \boldsymbol{\varphi}(a_t, \boldsymbol{x}_t)^\top \sum_{h \in [H]} \boldsymbol{b}_t(h) \, \boldsymbol{\theta}_h^\star \,.$$

The vectors $\big(\boldsymbol{\varphi}(a, \boldsymbol{x}_t) \boldsymbol{b}_t(h)\big)_{h \in [H]}$ act as contexts in linear contextual bandits. Nelson et al. (2022) only provide an analysis when these contexts are known (because the HMM parameters are assumed to be known in their theoretical analysis) but with the techniques introduced in this article, these contexts may be estimated and the reduction to linear bandits can be saved.

See Appendix A for details and a regret analysis taking care of estimation errors: under Assumption 2.2, we obtain a high probability regret bound of $\widetilde{\mathcal{O}}(T^{3/4})$, where the extra $\widetilde{\mathcal{O}}(T^{1/4})$ term is due to belief estimation (recovering $\widetilde{\mathcal{O}}(T^{1/2})$ if the belief were known).

**No such reduction for model (1).** There is no such reduction in the reward model (1) primarily studied in this article, where reward depend directly on the hidden states $h_t$. For this model, for all $a \in \mathcal{A}$,

$$\mathbb{E}\big[r_t(a) \mid \mathcal{F}_t^{\mathrm{obs}}\big]$$
$$= \boldsymbol{\varphi}(a, \boldsymbol{x}_t)^\top \sum_{h \in [H]} \mathbb{P}(h_t = h \mid \mathcal{F}_t^{\mathrm{obs}}) \boldsymbol{\theta}_h^\star \,, \quad (4)$$

but $\mathbb{P}(h_t = h \mid \mathcal{F}_t^{\mathrm{obs}})$ is a complex quantity, depending on the strategy implemented (as the actions played are $\mathcal{F}_t^{\mathrm{obs}}$–measurable), that cannot be easily estimated, and that is in general different from the belief $\boldsymbol{b}_t(h)$. Appendix B further details the issues that arise.

### 2.3. Two Notions of Regret

**Pseudo-regret based on beliefs.** The literature of bandits with latent space dynamics considers benchmarks involving posterior probabilities over the states of the form

$$\boldsymbol{b}_t^{\mathrm{bnk}} : h \in [H] \longmapsto \boldsymbol{b}_t^{\mathrm{bnk}}(h) = \mathbb{P}(h_t = h \mid \mathcal{F}_t^{\mathrm{bnk}})$$

for filtrations $\sigma(\boldsymbol{x}_{1:t}) \subseteq \mathcal{F}_t^{\mathrm{bnk}} \subseteq \mathcal{F}_t^{\mathrm{obs}}$ discussed below; these posterior probabilities rely on the knowledge of the HMM parameters. The associated benchmarks are of the form of sums of

$$\max_{a \in \mathcal{A}} \sum_{h \in [H]} \mathbb{E}\big[r_t(a) \mid \mathcal{F}_t^{\mathrm{bnk}}\big] = \max_{a \in \mathcal{A}} \sum_{h \in [H]} \boldsymbol{b}_t^{\mathrm{bnk}}(h) \, \boldsymbol{\varphi}(a, \boldsymbol{x}_t)^\top \boldsymbol{\theta}_h^\star \,,$$

where the equality holds by the tower rule.

Zhou et al. (2021) consider a model with $\{0,1\}$–valued rewards and (only) because of that, may take $\mathcal{F}_t^{\mathrm{bnk}} = \mathcal{F}_t^{\mathrm{obs}}$.

This choice however is somewhat unnatural, as the benchmark is not intrinsic and depends on the strategy used.

Nelson et al. (2022) consider a more intrinsic choice, which also does not constrain rewards to take finitely many values: $\mathcal{F}_t^{\mathrm{bnk}} = \sigma(\boldsymbol{x}_{1:t})$, i.e., the posterior probabilities equal the beliefs $\boldsymbol{b}_t$ and are based only on contexts. Thus, no additional information from the complex dependencies of rewards on the hidden states is exploited. More formally, they consider associated pseudo-regret defined by

$$R_T = \sum_{t=1}^{T} \max_{a \in \mathcal{A}} \sum_{h \in [H]} \boldsymbol{b}_t(h) \, \boldsymbol{\varphi}(a, \boldsymbol{x}_t)^{\top} \boldsymbol{\theta}_h^{\star} -$$
$$\sum_{t=1}^{T} \sum_{h \in [H]} \boldsymbol{b}_t(h) \, \boldsymbol{\varphi}(a_t, \boldsymbol{x}_t)^{\top} \boldsymbol{\theta}_h^{\star} \,. \quad (5)$$

**Regret based on actual rewards.** The first sum in the definition (5) admits some natural interpretation as the sum of actual rewards achieved, up to some high-probability $\sqrt{T}$–deviation terms, by an oracle that would know the HMM parameters and the reward-model parameters $\boldsymbol{\theta}_h^{\star}$, and would pick its actions based on the contexts observed. Indeed, for all $a \in \mathcal{A}$,

$$\mathbb{E}\big[r_t(a) \mid \boldsymbol{x}_{1:t}\big] = \boldsymbol{\varphi}(a, \boldsymbol{x}_t)^{\top} \sum_{h \in [H]} \boldsymbol{b}_t(h) \, \boldsymbol{\theta}_h^{\star} \,.$$

However, it is actually difficult to interpret the second sum in (5), because in general, it is difficult to relate

$$\boldsymbol{\varphi}(a_t, \boldsymbol{x}_t)^{\top} \sum_{h \in [H]} \boldsymbol{b}_t(h) \, \boldsymbol{\theta}_h^{\star}$$

to conditional expectations like $\mathbb{E}\big[r_t(a_t) \mid \boldsymbol{x}_{1:t}, a_t\big]$ or $\mathbb{E}\big[r_t(a_t) \mid \boldsymbol{x}_{1:t}\big]$. This is due, exactly as in Equation (4), to the complex dependencies between the actions taken and the hidden states, through the rewards observed. See Appendix B for details.

However, Appendix E.3 proves, by adapting the proof of Theorem 4.2 (and in particular, the one of Lemma E.2), that for the strategy considered in Box A (which proceeds in stages), the second sum in Equation (5) is close to the sum $\sum_{t \in [T]} r_t(a_t)$ of actual rewards, with high-probability and up to an additive term of order $T^{5/8}$ up to poly-logarithmic factors. Put differently, the regret bounds on $R_T$ stated later in this article also yield bounds on the actual regret

$$R_T^{\mathrm{actual}} = \sum_{t=1}^{T} r_t(a_t^{\star}) - r_t(a_t) \,,$$
$$\text{where} \qquad a_t^{\star} \in \underset{a \in \mathcal{A}}{\operatorname{argmax}} \sum_{h \in [H]} \boldsymbol{b}_t(h) \, \boldsymbol{\varphi}(a, \boldsymbol{x}_t)^{\top} \boldsymbol{\theta}_h^{\star} \,.$$

# 3. Algorithm(s): Staged LinUCB on Estimated Beliefs

In this section, we both present our main algorithm (Box A) addressing the most complex reward model of Equation (1), as well as a special case thereof addressing the simplified model of Equation (3) but in a more generic way than in Nelson et al. (2022), as we do not fix a specific belief estimation subroutine (online expectation-maximization in their case) but consider any efficient such subroutine (see Assumption 3.1). We discuss these subroutines first (in Section 3.1) and then state the strategies (in Section 3.2).

## 3.1. Belief Estimation Subroutines

As justified in Appendix B and as in Nelson et al. (2022), due to the complex dependencies between rewards and hidden states, we estimate beliefs only based on contexts. We therefore define a belief estimation subroutine $\mathcal{B}$ as a sequence of functions where the $t$–th function

$$\boldsymbol{x}_1, \dots, \boldsymbol{x}_t \longmapsto \widehat{\boldsymbol{b}}_t = \big(\widehat{\boldsymbol{b}}_t(h)\big)_{h \in [H]}$$

associates with the contexts $\boldsymbol{x}_1, \dots, \boldsymbol{x}_t$ a probability distribution $\widehat{\boldsymbol{b}}_t$ over the hidden state spaces $[H]$.

We provide no methodological development on the estimation of beliefs and instead resort to known results, up to one addition. The estimation of HMM parameters, and thus of beliefs, requires knowing the number $H$ of hidden states but only provides estimates that are correct up to permutations of the hidden states (as the latter have no specific ordering). This is why estimation guarantees are only formulated in norms. However, the strategy considered (see Box A) must keep track of specific states, as it will maintain estimators for each parameter $\boldsymbol{\theta}^{\star}$. That the labeling of hidden states is consistent throughout time will be vital. We achieve this through an additional alignment step. See details on this issue and on the solution in Appendix C

To make our arguments generic, we consider the following assumption on the belief-estimation subroutine $\mathcal{B}$; examples and pointers below explain why it is a reasonable assumption (and to which large classes of hidden Markov chains it applies).

**Assumption 3.1** (belief estimation error). The belief estimation procedure $\mathcal{B}$ is such that for all hidden Markov chains $(\boldsymbol{\pi}, \boldsymbol{M}, \nu)$ in a wide class, there exist

- a constant $T_{\mathcal{B}, \boldsymbol{M}, \nu}$ not necessarily known to the learner,
- a fully known belief error function $U_{\mathrm{belief}}$ on $\{1, 2, \dots\} \times (0, 1)$, where $U_{\mathrm{belief}}(t, \delta)$ depends logarithmically on $\delta$ and, up to poly-log factors, for each $\delta \in (0, 1)$,

$$\sum_{t \in [T]} U_{\mathrm{belief}}(t, \delta) = \widetilde{\mathcal{O}}\big(T^{1/2}\big) \,,$$

such that for all $\delta \in (0,1)$, with probability at least $1-\delta$, the following statements hold for all $t \geqslant T_{\mathcal{B},\boldsymbol{M},\nu}\big(1 + \ln(1/\delta)\big)$:

- first, the labeling of hidden states is consistent over the rounds considered;
- second, $\big\|\widehat{\boldsymbol{b}}_t - \boldsymbol{b}_t\big\|_1 \leqslant U_{\mathrm{belief}}(t,\delta)$.

In the HMM literature, belief estimation is more commonly referred to as the estimation of the filtering distributions. The hidden state space is typically assumed to be finite, while the context space may be finite or continuous. For the sake of exposition, and since the belief estimation is used here only as an independent subroutine, we will mostly focus on the case of a finite context set.

**Example 1: Spectral method for finite context sets $\mathcal{X}$.** The so-called spectral method was proposed by Hsu et al. (2012) and further developed by Anandkumar et al. (2012) and Anandkumar et al. (2014). It provides estimates of the HMM transition matrix $\boldsymbol{M}$ and of the emission distributions $\nu_h$. De Castro et al. (2017) show how the performance of these estimates, combined with the Bayes' update rule, transfers into a performance bound on estimated beliefs of the form of Assumption 3.1. This is formally stated in Lemma 3.3 below.

Assume that the context set $\mathcal{X}$ is finite, so that each emission distribution $\nu_h$ on $\mathcal{X}$ may be seen as a column vector, and let $\boldsymbol{E}$ denote the emission matrix, indexed by $\mathcal{X} \times [H]$, obtained by concatenating the vectors $\nu_h$ as $h \in [H]$. We assume below that $\boldsymbol{E}$ has full column rank: this imposes, in particular, that $H$ is smaller than the cardinality $X$ of $\mathcal{X}$.

Recall that $\sigma$ is a singular value of $\boldsymbol{E}$ if $\sigma^2$ is an eigenvalue of the square matrix $\boldsymbol{E}^\top \boldsymbol{E}$.

**Assumption 3.2.** The context set $\mathcal{X}$ is finite, with cardinality denoted by $|\mathcal{X}| = X$.

The emission matrix $\boldsymbol{E}$ has full column rank with smallest singular value $\sigma_{\min}(\boldsymbol{E}) > 0$, and its smallest element satisfies
$$e_{\nu,\min} \overset{\mathrm{def}}{=} \min_{h \in [H]} \min_{x \in \mathcal{X}} \nu_h(x) > 0 \,.$$

The transition matrix $\boldsymbol{M}$ is invertible, with smallest eigenvalue denoted by $\sigma_{\min}(\boldsymbol{M}) > 0$, and the smallest element of $\boldsymbol{M}$ is positive: $\varepsilon_{\boldsymbol{M}} = \min_{h,h'} M_{h,h'} > 0$.

Finally, the initial distribution $\boldsymbol{\pi}$ is the (unique) stationary distribution of $\boldsymbol{M}$.

Appendix C reviews the literature necessary to obtain the guarantee stated in Lemma 3.3 (whose proof may be found in Appendices C.1 and C.2), and also provides more details on the underlying belief estimation procedure (namely, the spectral method combined with a Bayes' update rule).

**Lemma 3.3.** *Assumption 3.1 is satisfied for all hidden Markov chains of Assumption 3.2, for the spectral*

*method (followed by an alignment step) combined with the Bayes' update rule, with the known belief error function $U_{\mathrm{belief}}(t,\delta) =$*

$$\ln(t)\left( H\sqrt{X}\sqrt{\frac{2\ln\big(6Xt(t+1)/\delta\big)}{t}} + \mathrm{e}^{-\sqrt{t-1}} \right)$$

*and the unknown threshold $T_{\mathcal{B},\boldsymbol{M},\nu}$ whose closed-form expression is provided in Equation (31).*

**Example 2: More general context sets $\mathcal{X}$.** De Castro et al. (2017) extended the spectral method and its analysis to the case of continuously-valued contexts, under an assumption that contexts are continuously projectable into a finite-dimensional feature space via basis functions such as splines, trigonometric functions, or wavelets.

### 3.2. LinUCB Strategies on Estimated Beliefs

For any probability distribution $\boldsymbol{b}$ over $[H]$, we use the shorthand notation, for all $a \in \mathcal{A}$ and $\boldsymbol{x} \in \mathcal{X}$,
$$\boldsymbol{b} \otimes \boldsymbol{\varphi}(a,\boldsymbol{x}) = \big(\boldsymbol{b}(h)\boldsymbol{\varphi}(a,\boldsymbol{x})\big)_{h \in [H]} \in \mathbb{R}^{dH} \,,$$
so that
$$\sum_{h \in [H]} \boldsymbol{b}(h)\,\boldsymbol{\varphi}(a,\boldsymbol{x})^\top \boldsymbol{\theta}_h^\star = \big(\boldsymbol{b} \otimes \boldsymbol{\varphi}(a,\boldsymbol{x})\big)^\top \boldsymbol{\theta}^\star \,.$$

Note that by Assumption 2.1, which considers the Euclidean norm in $\mathbb{R}^d$, and the fact that $\boldsymbol{b}$ is a probability distribution, we also have, for the Euclidean norm in $\mathbb{R}^{dH}$,
$$\forall a \in \mathcal{A}, \ \forall \boldsymbol{x} \in \mathcal{X}, \qquad \big\|\boldsymbol{b} \otimes \boldsymbol{\varphi}(a,\boldsymbol{x})\big\|_2 \leqslant 1 \,. \quad (6)$$

We estimate the stacked vector $\boldsymbol{\theta}^\star = (\boldsymbol{\theta}_h^\star)_{h \in [H]} \in \mathbb{R}^{dH}$ through a LinUCB-style (Abbasi-Yadkori et al., 2011) approach: let $\lambda > 0$ and introduce, for $t \geqslant 1$, the (symmetric definite positive thus invertible) Gram matrix
$$G_t \overset{\mathrm{def}}{=} \sum_{\tau=1}^t \big(\widehat{\boldsymbol{b}}_\tau \otimes \boldsymbol{\varphi}(a_\tau,\boldsymbol{x}_\tau)\big)\big(\widehat{\boldsymbol{b}}_\tau \otimes \boldsymbol{\varphi}(a_\tau,\boldsymbol{x}_\tau)\big)^\top + \lambda \boldsymbol{I}_{dH} \,,$$
based on which we define the estimates
$$\widehat{\boldsymbol{\theta}}_t = G_t^{-1} \sum_{\tau=1}^t \big(\widehat{\boldsymbol{b}}_\tau \otimes \boldsymbol{\varphi}(a_\tau,\boldsymbol{x}_\tau)\big)\, r_\tau(a_\tau) \,. \quad (7)$$

**Strategy in the most complex reward model (1).** As justified in Appendix B, we consider a strategy that works in stages of lengths $\ell \geqslant 1$ and only performs the estimations (7) periodically, at rounds $t$ multiple of $\ell$. This defines stages, where stage $s \geqslant 1$ gather rounds $(s-1)\ell+1$ to $s\ell$. Within a stage, rewards are estimated by estimates of their conditional means $\boldsymbol{\varphi}(a,\boldsymbol{x}_t)^\top \boldsymbol{\theta}_{h_t}^\star$, of the form
$$\sum_{h \in [H]} \widehat{\boldsymbol{b}}_t(h)\boldsymbol{\varphi}(a,\boldsymbol{x}_t)^\top \widehat{\boldsymbol{\theta}}_{(s-1)\ell,h} + \varepsilon_{t,a} \,,$$

where the $\varepsilon_{t,a}$ are confidence bonuses. The strategy considered is optimistic and plays arms that maximize the upper confidence estimates defined above.

The resulting strategy, called *staged LinUCB on estimated beliefs*, is formally stated in Box A.

---

**BOX A: STAGED LINUCB ON ESTIMATED BELIEFS**

**Known parameters:** finite action set $\mathcal{A}$; context set $\mathcal{X}$; transfer function $\boldsymbol{\varphi} : \mathcal{A} \times \mathcal{X} \rightarrow \mathbb{R}^d$; finite state space $[H]$

**Unknown parameters:** HMM parameters, given by a transition matrix $\boldsymbol{M} = (M_{h,h'})_{(h,h') \in [H]}$ and emission distributions $(\nu_h)_{h \in [H]}$ over $\mathcal{X}$; reward parameters $\boldsymbol{\theta}_h^\star \in \mathbb{R}^d$, for $h \in [H]$

**Inputs:** risk $\delta \in (0,1)$; belief estimation subroutine $\mathcal{B}$; stage length $\ell \geqslant 1$; regularization parameter $\lambda > 0$; closed-form expression for the confidence bonuses $\varepsilon_{t,a}$, possibly depending on $\delta$, $\lambda$, and $\ell$

**Initialization:** set $\widehat{\boldsymbol{\theta}}_0 = (1/\lambda)\,\mathbf{1} \in \mathbb{R}^{dH}$

**For stages** $s = 1, 2, \dots$ **:**
**For rounds** $t = (s-1)\ell + 1, \dots, s\ell$, **the learner:**
1. Observes the context $\boldsymbol{x}_t$, drawn independently by the environment from $\nu_{h_t}$;
2. Obtains the belief estimate $\widehat{\boldsymbol{b}}_t$ by feeding $\boldsymbol{x}_1, \dots, \boldsymbol{x}_t$ to the subroutine $\mathcal{B}$;
3. Computes estimated rewards: for all $a \in \mathcal{A}$,
$$\widehat{r}_t(a) = \sum_{h \in [H]} \widehat{\boldsymbol{b}}_t(h) \boldsymbol{\varphi}(a, \boldsymbol{x}_t)^\top \widehat{\boldsymbol{\theta}}_{(s-1)\ell, h};$$
4. Picks an action $a_t \in \underset{a \in \mathcal{A}}{\mathrm{argmax}}\{\widehat{r}_t(a) + \varepsilon_{t,a}\}$;
5. Obtains and observes the reward
$$r_t(a_t) = \boldsymbol{\varphi}(a_t, \boldsymbol{x}_t)^\top \boldsymbol{\theta}_{h_t}^\star + \eta_t(a_t);$$
**end**
Computes $\widehat{\boldsymbol{\theta}}_{s\ell}$ as in Equation (7).
**end**

---

**Strategy in the simplified model** (3). Our generic version of the strategy by Nelson et al. (2022) is given by the Box-A strategy run with $\ell = 1$, i.e., updating the LinUCB estimates of $\boldsymbol{\theta}^\star$ at each round, and with the reward-obtention step (numbered 5 in Box A) of course replaced by Equation (3). For the sake of clarity, we state separately this strategy in Box B of Appendix A.

## 4. Regret Bounds

In this section, we present regret analyses both for the main strategy of Box A addressing the most complex reward model (1), as well as its special case addressing the simplified model of Equation (3) (see the paragraph above). We start with the latter as it can be performed with no additional assumption.

### 4.1. Regret Bound for the Simplified Model (3)

In Appendix A, we state (Theorem A.1) and show that under Assumption 2.2 (sub-Gaussian noise), Assumption 3.1 (on the belief estimation subroutine), and Assumption 2.1 (boundedness of rewards), with proper inputs, the strategy in the simplified model (3) described above satisfies, with probability at least $1 - \delta$, up to poly-log factors,

$$R_T = \widetilde{\mathcal{O}}\big(T^{3/4}\big),$$

where a closed-form expression of the regret bound may be found in the proof, see Equation (21).

**Comparison to Nelson et al. (2022, Theorem 2).** First, Nelson et al. (2022, Theorem 2) do not take into account the belief estimation error into account in their regret bound, which, in addition, only holds in expectation; they obtain a $\sqrt{T}$ rate and the proof of Theorem A.1 shows that the worsened rate $T^{3/4}$ is only due to the belief estimation error.

Second, Nelson et al. (2022, Theorem 2) consider a milder noise condition (Assumption 2.3 instead of Assumption 2.2) but to do so, require a forgetting condition (as Assumption 4.1 below). We instead provide a more direct analysis, close to the standard LinUCB analysis and not requiring this forgetting condition; see Appendix A.

Third, the bound of Nelson et al. (2022, Theorem 2) is an expected bound, and not a bound in high probability; it involves constants that heavily depend on the problem, in particular, on the reward gaps, while the bound achieved in Theorem A.1 is model-free for the part not linked to the estimation of HMM parameters, see Equation (21).

Fourth, Nelson et al. (2022, Theorem 1) also impose a non-degeneracy assumption on its population design matrix, which can be stated as follows in our extended setting, denoting by $\lambda_{\min}$ the smallest eigenvalue: for all actions $a \in \mathcal{A}$,

$$\liminf_{T \to \infty} \lambda_{\min}\big(\tilde{G}_t^{(a)}\big) > 0, \qquad \text{where}$$

$$\tilde{G}_t^{(a)} = \frac{1}{T} \sum_{t=1}^T \mathbb{E}\Big[\mathbb{1}_{\{a = a_t^\star\}} \big(\widehat{\boldsymbol{b}}_t \otimes \boldsymbol{\varphi}(a, \boldsymbol{x}_t)\big)\big(\widehat{\boldsymbol{b}}_t \otimes \boldsymbol{\varphi}(a, \boldsymbol{x}_t)\big)^\top\Big].$$

Note that the sum in the definition of $\tilde{G}_t^{(a)}$ is restricted to rounds $t$ such that $a_t^\star = a$, where

$$a_t^\star \in \underset{a \in \mathcal{A}}{\mathrm{argmax}} \sum_{h \in [H]} \boldsymbol{b}_t(h)\, \boldsymbol{\varphi}(a, \boldsymbol{x}_t)^\top \boldsymbol{\theta}_h^\star.$$

This implies that the population design matrix grows linearly in all directions. We do not impose such a coverage assumption; instead, we only use $1/\lambda_{\min}(G_t) \leqslant 1/\lambda$, which leads to a larger regret rate but avoids this additional condition.

In a nutshell, we leverage the reduction to linear contextual bandits proposed by Nelson et al. (2022) a in a more direct and more efficient way.

## 4.2. Regret Bound for the Most Complex Model (1)

We require a final, classic (see Cappé et al., 2005), assumption on the HMM: that it satisfies some fast forgetting property. Details, exemples, and further references (including the alternative forgetting condition assumed by Nelson et al., 2022) are provided in Appendix D.3.

**Assumption 4.1** (exponentially fast forgetting of initial condition)**.** There exists a constant $\gamma \in [0, 1)$ so that, for all $s \leqslant t$, for all pairs $h, h' \in [H]$ of hidden states,

$$\sum_{j \in [H]} \Big| \mathbb{P}_{\{h_s = h\}}(h_t = j \mid \boldsymbol{x}_{s+1:t})$$
$$- \mathbb{P}_{\{h_s = h'\}}(h_t = j \mid \boldsymbol{x}_{s+1:t}) \Big| \leqslant 2\gamma^{t-s} \,.$$

We may now state our main result. The $T^{7/8}$ rate achieved therein must be contrasted with the $T^{3/4}$ rate discussed in Section 4.1 above: the price to pay for facing the actual latent model (and not an overly simplified version thereof) is a $T^{1/8}$ factor with our method, mostly due to of proceeding in stages. While Appendix B explains how handy it is to proceed in stages, this might be avoidable and the regret bound might be improvable. In particular, we do not provide any matching regret lower bound.

**Theorem 4.2.** *Assume the horizon $T$ is known to the learner and fix $\delta \in (0, 1)$. Consider the strategy of Box A with a belief estimation subroutine satisfying Assumption 3.1, with parameters $\lambda = T^{3/4}$ and $\ell = \lceil T^{3/4} \rceil$, as well as the confidence bonuses $\varepsilon_{t,a} = 1 + \sqrt{d}/\lambda$ for $t \in [1, \ell]$ and for $t \geqslant \ell + 1$,*

$$\varepsilon_{t,a} = U_{\text{belief}}(t, \delta/2) + f_t \left\| G^{-1}_{(s_t-1)\ell} \Big( \widehat{\boldsymbol{b}}_t \otimes \boldsymbol{\varphi}(a, \boldsymbol{x}_t) \Big) \right\|_2$$

*where* $\quad f_t = \lambda\sqrt{H}\, C_{\boldsymbol{\theta}^\star} + 4\sqrt{\dfrac{s_T(s_t-1)(1+s_t\gamma)\ell}{\delta(1-\gamma)}} + $

$$\sqrt{\frac{4s_T}{\delta} C_\eta (s_t-1)\ell} + \frac{2(s_t-1)\gamma}{1-\gamma} + \sum_{\tau=1}^{(s_t-1)\ell} U_{\text{belief}}(\tau, \delta/2)$$

*and where $s_t = \lceil t/\ell \rceil$ denotes the stage to which round $t$ belongs. Then, under Assumption 2.1 (boundedness of rewards), Assumption 2.3 (noise with bounded conditional second-order moments), Assumption 3.1 (controlled belief estimation error), and Assumption 4.1 (exponentially fast forgetting), with probability at least $1 - \delta$, up to poly-log factors,*

$$R_T = \widetilde{\mathcal{O}}\big(T^{7/8}\big) \,.$$

*A closed-form expression of the regret bound may be found in the proof, see Equations (48) and (49).*

## 4.3. Proof Sketch for Theorem 4.2

The full proof of Theorem 4.2 may be found in Appendix E.

We introduce a filtration augmented by the algorithmic updates, by considering the estimates $\widehat{\boldsymbol{\theta}}_{s\ell}$ computed at the end of past complete stages $s \leqslant s_t - 1$ on top of contexts $\boldsymbol{x}_{1:t}$:

$$\mathcal{U}_t = \sigma\Big( \boldsymbol{x}_{1:t}, \big( \widehat{\boldsymbol{\theta}}_{s\ell} \big)_{s \leqslant s_t - 1} \Big) \,.$$

The key of the proof, as discussed in Appendix B, is that this filtration is such that $a_t$ is $\mathcal{U}_t$–measurable (by design, thanks to staging in Box A) while

$$\overline{\boldsymbol{b}}_t(h) = \mathbb{P}(h_t = h \mid \mathcal{U}_t) \quad \text{and} \quad \boldsymbol{b}_t(h) \qquad (8)$$

are close enough; this may be guaranteed by Assumption 4.1 (exponentially fast forgetting condition).

**Summing confidence bounds.** The core of the proof is to show that the confidence bonuses $\varepsilon_{t,a}$ in Theorem 4.2 satisfy, with high probability, uniformly over $T_0 \leqslant t \leqslant T$ and $a \in \mathcal{A}$, that

$$\Bigg| \sum_{h \in [H]} \boldsymbol{b}_t(h) \boldsymbol{\varphi}(a, \boldsymbol{x}_t)^\top \boldsymbol{\theta}^\star_h$$
$$- \sum_{h \in [H]} \widehat{\boldsymbol{b}}_t(h) \boldsymbol{\varphi}(a, \boldsymbol{x}_t)^\top \widehat{\boldsymbol{\theta}}_{(s_t-1)\ell, h} \Bigg| \leqslant \varepsilon_{t,a} + 2T_0/\lambda \,, \quad (9)$$

where $T_0$ is essentially the unknown constant threshold of Assumption 3.1. Based on that, classic manipulations entail that the pseudo-regret $R_T$ is essentially bounded by

$$2 \sum_{t \in [T]} \varepsilon_{t,a_t} \quad \text{which is seen} \quad = \widetilde{\mathcal{O}}\big(T^{7/8}\big)$$

by substituting classic linear-algebra bounds (the so-called elliptic potential lemma, adapted to stages, see Abbasi-Yadkori et al., 2011, Section C) and by carefully picking $\lambda$ and $\ell$ to optimize the bound.

Thus, the core of the proof is to show (9).

**Three sums, including a difficult one.** The left-hand side of (9) is bounded by the sum of two terms; first, $\|\boldsymbol{b}_t - \widehat{\boldsymbol{b}}_t\|_1$, which is manageable thanks to the estimation Assumption 3.1; and second,

$$\Big| \big( \widehat{\boldsymbol{b}}_t \otimes \boldsymbol{\varphi}(a, \boldsymbol{x}_t) \big)^\top \big( \boldsymbol{\theta}^\star - \widehat{\boldsymbol{\theta}}_{(s_t-1)\ell} \big) \Big|$$
$$= \big( \widehat{\boldsymbol{b}}_t \otimes \boldsymbol{\varphi}(a, \boldsymbol{x}_t) \big)^\top G^{-1}_{(s_t-1)\ell} \times$$
$$\big( S'_{\text{diff},(s_t-1)\ell} + S'_{\text{belief},(s_t-1)\ell} + S'_{\text{eta},(s_t-1)\ell} - \lambda \boldsymbol{I}_{dH} \boldsymbol{\theta}^\star \big)$$

where the equality follows by substituting the very definition of $\widehat{\boldsymbol{\theta}}_{(s_t-1)\ell}$ and where the exact definitions of the three $S$ terms are in Appendix E. We handle the term in the display

above by a Cauchy-Schwarz inequality and the boundedness Assumption 2.1, together with the fact the Euclidean norms

$$\|S'_{\text{diff},(s_t-1)\ell}\|, \quad \|S'_{\text{belief},(s_t-1)\ell}\|, \quad \|S'_{\text{eta},(s_t-1)\ell}\|$$

behave respectively as the absolute values of

$$S^\Delta = \sum_{\tau=1}^{(s-1)\ell} \varphi(a_\tau, x_\tau)^\top \left( \theta^\star_{h_\tau} - \sum_{h'\in[H]} \bar{b}_\tau(h')\theta^\star_{h'} \right),$$

$$S^b = \sum_{\tau=1}^{(s-1)\ell} \sum_{h'\in[H]} \varphi(a_\tau, x_\tau)^\top \theta^\star_{h'} \left( \bar{b}_\tau(h') - \widehat{b}_\tau(h') \right),$$

$$S^\eta = \sum_{\tau=1}^{(s-1)\ell} \eta_\tau(a_\tau).$$

Now, the term $S^b$ may be bounded by

$$\sum_{\tau=1}^{(s-1)\ell} \left\| \bar{b}_\tau - b_\tau \right\|_1 + \sum_{\tau=1}^{(s-1)\ell} \left\| b_\tau - \widehat{b}_\tau \right\|_1,$$

where Equation (8) and Assumption 3.1 take respective care of each sum. The term $S^\eta$ could be bounded by resorting to martingale arguments (like in the LinUCB analysis, though we rather mimic for it in Appendix E the proof scheme used for $S^\Delta$ and described next).

The term $S^\Delta$ is the term that is difficult to control, see the discussions in Appendix B: LinUCB-type analyses are not applicable. Indeed, denoting

$$z_\tau = \varphi(a_\tau, x_\tau)^\top \left( \theta^\star_{h_\tau} - \sum_{h\in[H]} \bar{b}_\tau(h)\theta^\star_h \right),$$

where $|z_\tau| \leqslant 2$ by Assumption 2.1; we have that

$$\mathbb{E}\big[z_\tau \mid U_\tau\big] = 0$$

but $z_\tau$ is not $U_\tau$–measurable (as it explicitly depends on $h_\tau$). However, we follow instead an approach by Nelson et al. (2022), which consists of controlling $S^\Delta$ in $\mathbb{L}^2$–norm and applying Markov's inequality: thanks to Assumption 4.1 (exponentially fast forgetting condition), $\mathbb{E}[z_\tau z_{\tau'}]$ is exponentially small when $\tau$ and $\tau'$ are separated, so that

$$\mathbb{E}\big[(S^\Delta)^2\big] \quad \text{is of order} \quad s^2\ell,$$

hence, $|S^\Delta|$ is smaller than $s\sqrt{\ell/\delta_s}$ with probability at least $1 - \delta_s$.

Collecting all elements, together with careful union bounds (taking $\delta_s = \delta/s_T$, where $s_T$ denotes the stage of $T$), concludes the proof. Again, the complete proof of Theorem 4.2 may be found in Appendix E.

## 5. Numerical Simulations

We consider a partially simulated but realistic data set derived from the UCI "Default of Credit Card Clients" dataset (Yeh, 2009; Yeh & Lien, 2009), in a banking marketing setup with three actions (calling a client; emailing a client; not reaching out). Two latent states, inflation and recession, affect both context distributions and the rewards.

Figure 1 reports empirically estimated pseudo-regrets of the Box A strategy (with stages of length $\ell = 37$ or without stages, i.e., for $\ell = 1$) versus a baseline formed by the LinUCB strategy by Abbasi-Yadkori et al. (2011) in its standard form (referred to as *Plain LinUCB* in the picture). This baseline ignores the latent-state dynamics altogether and therefore does not exploit either the HMM structure or the belief estimates; as a result, it suffers linear pseudo-regret. By contrast, the strategies developed achieve sublinear pseudo-regrets.

Full simulation details, hyperparameter definitions, and robustness checks with respect to hyperparameter grids are provided in Section F.

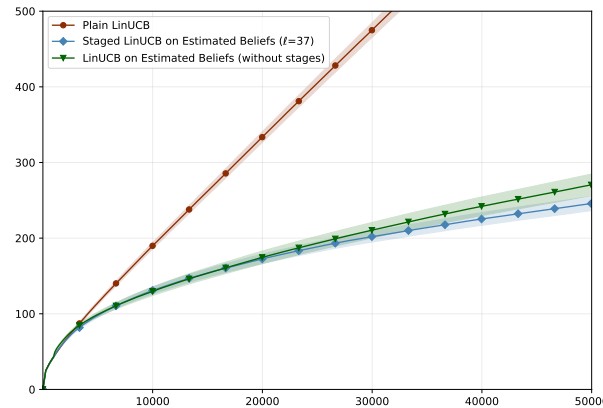

*Figure 1.* Pseudo-regrets averaged over 100 runs. Solid lines correspond to averages and shaded areas to $\pm 2$ standard errors.

## 6. Limitations and Future Work

The main open questions are around optimality: first, showing that a $T^{3/4}$ rate on the pseudo-regret is inevitable, even in the simplified reward model, due to belief estimation; second, possibly improving the $T^{7/8}$ rate in the most complex reward model into a $T^{3/4}$ rate by finding a more efficient theoretical argument than the $\mathbb{L}^2$–Markov exhibited or, on the algorithmic front, by avoiding proceeding in stages. Indeed, the $\mathbb{L}^2$–Markov argument entails dependencies on the probabilities of failure as $1/\sqrt{\delta_s}$, instead of typical $\sqrt{\ln(1/\delta_s)}$ dependencies under exponential-martingale arguments, and this worsened dependency comes at a polynomial cost in the final regret bound.

## Impact Statement

This paper presents work whose goal is to advance the field of Machine Learning. There are many potential societal consequences of our work, none which we feel must be specifically highlighted here.

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

## A. Algorithm and Analysis for the Simplified Reward Model of Nelson et al. (2022)

Section 2.2 indicated that (a generalized version of) the simplified reward model by Nelson et al. (2022) may be stated as

$$r'_t(a) = \sum_{h \in [H]} \boldsymbol{b}_t(h)\, \boldsymbol{\varphi}(a, \boldsymbol{x}_t)^\top \boldsymbol{\theta}^\star_h + \eta'_t(a)\,, \qquad \text{where} \qquad \boldsymbol{b}_t(h) = \mathbb{P}(h_t = h \mid \boldsymbol{x}_{1:t})$$

and where here, we assume that the noise terms satisfy a sub-Gaussian assumption as in Assumption 2.2: denoting by

$$\mathcal{F}'^{\text{all}}_t = \sigma\left( \Big( h_\tau,\, \boldsymbol{x}_\tau,\, \big(\eta'_\tau(a)\big)_{a \in \mathcal{A}} \Big)_{\tau \leqslant t-1},\, h_t,\, \boldsymbol{x}_t \right)$$

the filtration with respect to all random variables anterior to the $\eta'_t(a)$, there exists $v_\eta$ such that for all $a \in \mathcal{A}$,

$$\mathbb{E}\big[\eta'_t(a) \mid \mathcal{F}'^{\text{all}}_t\big] = 0 \qquad \text{and} \qquad \mathbb{E}\big[\mathrm{e}^{\lambda \eta'_t(a)} \mid \mathcal{F}'^{\text{all}}_t\big] \leqslant \mathrm{e}^{\lambda^2 v_\eta^2 / 2}\,. \qquad (10)$$

**Aim of this appendix.** This appendix recalls the main claim by Nelson et al. (2022), namely, how a reduction to standard linear contextual bandits may be performed for the reward model above. Unlike Nelson et al. (2022), we also provide a straightforward analysis based on the LinUCB analysis, taking into account the belief estimation error (see Appendix C for a description of a belief estimation routine and its associated guarantees), and yielding high-probability bounds (not only bounds in expectation); no HMM forgetting properties are required to that end.

Actually, the more complex analysis by Nelson et al. (2022), which, in particular, relies on HMM forgetting properties (as in Assumption 4.1, see more generally Appendix D.3), is only required because of the relaxation considered on the noise terms: Nelson et al. (2022) only assume that conditional second-order moments are bounded, as in Assumption 2.3. We see this relaxation as unimportant.

**Algorithm.** We use the reduction to linear contextual bandits pointed out by Nelson et al. (2022), discussed in Section 2.2, and relying on the rewriting

$$r'_t(a) = \sum_{h \in [H]} \boldsymbol{b}_t(h)\, \boldsymbol{\varphi}(a, \boldsymbol{x}_t)^\top \boldsymbol{\theta}^\star_h + \eta'_t(a) = \big(\boldsymbol{b}_t \otimes \boldsymbol{\varphi}(a, \boldsymbol{x}_t)\big)^\top \boldsymbol{\theta}^\star + \eta'_t(a)\,, \qquad \text{where} \qquad \boldsymbol{\theta}^\star = \big(\boldsymbol{\theta}^\star_h\big)_{h \in [H]};$$

the quantities $\boldsymbol{b}_t \otimes \boldsymbol{\varphi}(a, \boldsymbol{x}_t)$ act as (unknown) contexts and mean rewards depend linearly on them, via the $dH$-dimensional parameter $\boldsymbol{\theta}^\star$. We also consider a belief estimation subroutine $\mathcal{B}$, as discussed in Section 3.1 (see also Appendix C), so as to replace the unknown contexts by known estimated contexts. Fix a regularization parameter $\lambda > 0$. At the end of each round $t \geqslant 1$, the algorithm computes

$$\widehat{\boldsymbol{\theta}}_t = \big(\widehat{\boldsymbol{\theta}}_{t,h}\big)_{h \in [H]} \stackrel{\text{def}}{=} G_t^{-1} \sum_{\tau=1}^{t} \big(\widehat{\boldsymbol{b}}_\tau \otimes \boldsymbol{\varphi}(a_\tau, \boldsymbol{x}_\tau)\big)\, r'_\tau(a_\tau)\,,$$

$$\text{where} \qquad G_t \stackrel{\text{def}}{=} \sum_{\tau=1}^{t} \big(\widehat{\boldsymbol{b}}_\tau \otimes \boldsymbol{\varphi}(a_\tau, \boldsymbol{x}_\tau)\big)\big(\widehat{\boldsymbol{b}}_\tau \otimes \boldsymbol{\varphi}(a_\tau, \boldsymbol{x}_\tau)\big)^\top + \lambda \boldsymbol{I}_{dH}\,, \quad (11)$$

based, in particular, on the estimated belief $\widehat{\boldsymbol{b}}_t$ obtained from $\mathcal{B}$ at the beginning of round $t$. Then, in the next round $t + 1$,

$$\text{mean rewards} \quad \sum_{h \in [H]} \boldsymbol{b}_{t+1}(h)\, \boldsymbol{\varphi}(a, \boldsymbol{x}_{t+1})^\top \boldsymbol{\theta}^\star_h \qquad \text{are estimated by} \qquad \widehat{r}_{t+1}(a) \stackrel{\text{def}}{=} \sum_{h \in [H]} \widehat{\boldsymbol{b}}_{t+1}(h)\, \boldsymbol{\varphi}(a, \boldsymbol{x}_{t+1})^\top \widehat{\boldsymbol{\theta}}_{t,h}\,.$$

The action $a_{t+1}$ to be played at round $t + 1$ is picked in an optimistic way as the action maximizing $\widehat{r}_{t+1}(a)$ plus some confidence bonus over $a \in \mathcal{A}$. The corresponding algorithm is formally stated in Box B. It corresponds to a LinUCB approach (Abbasi-Yadkori et al., 2011) with contexts computed based on estimated beliefs; the mere difference to the main algorithm of Box A is that it does not proceed in stages.

---

### BOX B: LINUCB ON ESTIMATED BELIEFS (WITHOUT STAGES)

**Known parameters:** finite action set $\mathcal{A}$; finite state space $[H]$; context space $\mathcal{X}$; transfer function $\boldsymbol{\varphi} : \mathcal{A} \times \mathcal{X} \to \mathbb{R}^d$

**Unknown parameters:** HMM parameters, given by a transition matrix $\boldsymbol{M} = (M_{h,h'})_{(h,h') \in [H]}$ and emission distributions $(\nu_h)_{h \in [H]}$ over $\mathcal{X}$; reward parameters $\boldsymbol{\theta}_h^\star \in \mathbb{R}^d$, for $h \in [H]$

**Inputs:** risk $\delta \in (0, 1)$; belief estimation subroutine $\mathcal{B}$ (see Section 3.1); regularization parameter $\lambda > 0$; closed-form expression for the confidence bonuses $\varepsilon_{t,a}$, possibly depending on $\delta$ and $\lambda$

**Initialization:** the learner sets $\widehat{\boldsymbol{\theta}}_0 = (1/\lambda)\,\mathbf{1} \in \mathbb{R}^{dH}$

**For** rounds $t \geqslant 1$ **the learner:**

1. Observes the context $\boldsymbol{x}_t$, drawn independently by the environment from $\nu_{h_t}$;

2. Obtains the belief estimate $\widehat{\boldsymbol{b}}_t$ by feeding $\boldsymbol{x}_1, \ldots, \boldsymbol{x}_t$ to the subroutine $\mathcal{B}$;

3. Computes the estimated mean rewards $\quad \widehat{r}_t(a) = \sum_{h \in [H]} \widehat{\boldsymbol{b}}_t(h)\,\boldsymbol{\varphi}(a, \boldsymbol{x}_t)^\top \widehat{\boldsymbol{\theta}}_{t-1,h} \quad$ for all $a \in \mathcal{A}$;

4. Picks an action $a_t \in \underset{a \in \mathcal{A}}{\operatorname{argmax}} \big\{ \widehat{r}_t(a) + \varepsilon_{t,a} \big\}$;

5. Obtains and observes the reward $\quad r_t'(a_t) = \sum_{h \in [H]} \boldsymbol{b}_t(h)\,\boldsymbol{\varphi}(a_t, \boldsymbol{x}_t)^\top \boldsymbol{\theta}_h^\star + \eta_t'(a_t)$;

6. Computes $\widehat{\boldsymbol{\theta}}_t$ as in Equation (11).

**Analysis.** At a high level, the analysis adapts the LinUCB proof to handle the substitution of the true contexts $\boldsymbol{b}_t \otimes \boldsymbol{\varphi}(a, \boldsymbol{x}_t)$ by estimations thereof. We follow closely classic analyses of LinUCB (the original reference by Abbasi-Yadkori et al., 2011, the monograph by Lattimore & Szepesvári, 2020, Chapters 19 and 20, as well as the extension by Brégère et al., 2019), with occasional simplifications or shortcuts—e.g., we avoid stating confidence ellipsoids on the $\boldsymbol{\theta}_h^\star$ and rather focus on confidence intervals on the mean payoffs, as studied in Lemma A.2 below. The bounds obtained in the sequel corresponds to the classic bound when $\widehat{\boldsymbol{b}}_t = \boldsymbol{b}_t$, i.e., if there was no estimation error for the beliefs. The formal aim is to prove the following theorem.

**Theorem A.1.** *Assume that the horizon $T$ is known to the learner and fix $\delta \in (0, 1)$. Consider the strategy of Box B with a belief estimation subroutine satisfying Assumption 3.1, with $\lambda = T^{1/2}$ and with the confidence bonuses (14). Then, under the boundedness stated in Assumption 2.1 and under the sub-Gaussian noise assumption (10), with probability at least $1 - \delta$, up to poly-log factors,*

$$R_T = \widetilde{\mathcal{O}}\big(T^{3/4}\big),$$

*where a closed-form expression of the regret bound may be found in the proof, see Equation (21).*

The total regret bound is basically given by 2 times the sum of the upper confidence bounds of Lemma A.2 below, which we prove first.

**Lemma A.2.** *Under Assumption 2.1 and for sub-Gaussian noise terms as in Equation (10), with probability at least $1 - \delta$, for all $t \geqslant 2$,*

$$\forall a \in \mathcal{A}, \qquad \left| \sum_{h \in [H]} \boldsymbol{b}_t(h)\,\boldsymbol{\varphi}(a, \boldsymbol{x}_t)^\top \boldsymbol{\theta}_h^\star - \sum_{h \in [H]} \widehat{\boldsymbol{b}}_t(h)\,\boldsymbol{\varphi}(a, \boldsymbol{x}_t)^\top \widehat{\boldsymbol{\theta}}_{t-1,h} \right|$$

$$\leqslant \big\| \boldsymbol{b}_t - \widehat{\boldsymbol{b}}_t \big\|_1 + \big\| \widehat{\boldsymbol{b}}_t \otimes \boldsymbol{\varphi}(a, \boldsymbol{x}_t) \big\|_{G_{t-1}^{-1}} \left( \frac{1}{\sqrt{\lambda}} \sum_{\tau=1}^{t-1} \| \boldsymbol{b}_\tau - \widehat{\boldsymbol{b}}_\tau \|_1 + \sqrt{\lambda H}\, C_{\boldsymbol{\theta}^\star} + v_\eta \sqrt{2 \ln(1/\delta) + dH \ln\big(1 + t/(\lambda dH)\big)} \right).$$

In the case $t = 1$, by several triangle inequalities, Assumption 2.1, the fact that $\boldsymbol{b}_1$ and $\widehat{\boldsymbol{b}}_1$ are probability vectors, and a Cauchy-Schwarz inequality, we have, with probability 1: for all $a \in \mathcal{A}$,

$$\left| \sum_{h \in [H]} \boldsymbol{b}_1(h)\,\boldsymbol{\varphi}(a, \boldsymbol{x}_1)^\top \boldsymbol{\theta}_h^\star - \sum_{h \in [H]} \widehat{\boldsymbol{b}}_1(h)\,\boldsymbol{\varphi}(a, \boldsymbol{x}_t)^\top \widehat{\boldsymbol{\theta}}_{0,h} \right| \leqslant \max_{h \in [H]} \overbrace{\big| \boldsymbol{\varphi}(a, \boldsymbol{x}_t)^\top \boldsymbol{\theta}_h^\star \big|}^{\leqslant 1} + \max_{h \in [H]} \overbrace{\big| \boldsymbol{\varphi}(a, \boldsymbol{x}_t)^\top \widehat{\boldsymbol{\theta}}_{0,h} \big|}^{\leqslant \sqrt{d}/\lambda} \leqslant 1 + \frac{\sqrt{d}}{\lambda},$$

where we resorted to the Cauchy-Schwarz inequality $\left\| \boldsymbol{\varphi}(a, \boldsymbol{x}_1) \right\|_2 \left\| \widehat{\boldsymbol{\theta}}_{0,h} \right\|_2 \leqslant 1 \times \sqrt{d}/\lambda$, since $\widehat{\boldsymbol{\theta}}_{0,h} = \mathbf{1} \in \mathbb{R}^d$.

*Proof.* By a triangle inequality and by the boundedness stated in Assumption 2.1,

$$
\left| \sum_{h \in [H]} \boldsymbol{b}_t(h) \, \boldsymbol{\varphi}(a, \boldsymbol{x}_t)^\top \boldsymbol{\theta}_h^\star - \sum_{h \in [H]} \widehat{\boldsymbol{b}}_t(h) \, \boldsymbol{\varphi}(a, \boldsymbol{x}_t)^\top \widehat{\boldsymbol{\theta}}_{t-1,h} \right|
$$
$$
\leqslant \underbrace{\left| \sum_{h \in [H]} \big( \boldsymbol{b}_t(h) - \widehat{\boldsymbol{b}}_t(h) \big) \overbrace{\boldsymbol{\varphi}(a, \boldsymbol{x}_t)^\top \boldsymbol{\theta}_h^\star}^{|\cdot| \leqslant 1} \right|}_{\leqslant \|\boldsymbol{b}_t - \widehat{\boldsymbol{b}}_t\|_1} + \underbrace{\left| \sum_{h \in [H]} \widehat{\boldsymbol{b}}_t(h) \boldsymbol{\varphi}(a, \boldsymbol{x}_t)^\top \big( \boldsymbol{\theta}_h^\star - \widehat{\boldsymbol{\theta}}_{t-1,h} \big) \right|}_{= (\widehat{\boldsymbol{b}}_t \otimes \boldsymbol{\varphi}(a, \boldsymbol{x}_t))^\top (\boldsymbol{\theta}^\star - \widehat{\boldsymbol{\theta}}_{t-1})} . \quad (12)
$$

The rest of the proof bounds the second term in the upper bound of Equation (12). To that end, we rewrite $r'_\tau(a_\tau)$ as

$$
r'_\tau(a_\tau) = \big( (\boldsymbol{b}_\tau - \widehat{\boldsymbol{b}}_\tau) \otimes \boldsymbol{\varphi}(a_\tau, \boldsymbol{x}_\tau) \big)^\top \boldsymbol{\theta}^\star + \big( \widehat{\boldsymbol{b}}_\tau \otimes \boldsymbol{\varphi}(a_\tau, \boldsymbol{x}_\tau) \big)^\top \boldsymbol{\theta}^\star + \eta'_\tau(a_\tau)
$$

and also note that by the definition of $G_{t-1}$ in Equation (11),

$$
\boldsymbol{\theta}^\star - \widehat{\boldsymbol{\theta}}_{t-1} = G_{t-1}^{-1} \left( G_{t-1} \boldsymbol{\theta}^\star - \sum_{\tau=1}^{t-1} \big( \widehat{\boldsymbol{b}}_\tau \otimes \boldsymbol{\varphi}(a_\tau, \boldsymbol{x}_\tau) \big) r'_\tau(a_\tau) \right)
$$
$$
= G_{t-1}^{-1} \left( \lambda \boldsymbol{\theta}^\star - \sum_{\tau=1}^{t-1} \big( \widehat{\boldsymbol{b}}_\tau \otimes \boldsymbol{\varphi}(a_\tau, \boldsymbol{x}_\tau) \big) \Big( r'_\tau(a_\tau) - \big( \widehat{\boldsymbol{b}}_\tau \otimes \boldsymbol{\varphi}(a_\tau, \boldsymbol{x}_\tau) \big)^\top \boldsymbol{\theta}^\star \Big) \right) .
$$

Thanks to these two equalities, we may decompose the second term in the upper bound of Equation (12) as

$$
\big( \widehat{\boldsymbol{b}}_t \otimes \boldsymbol{\varphi}(a, \boldsymbol{x}_t) \big)^\top \big( \boldsymbol{\theta}^\star - \widehat{\boldsymbol{\theta}}_{t-1} \big) = \big( \widehat{\boldsymbol{b}}_t \otimes \boldsymbol{\varphi}(a, \boldsymbol{x}_t) \big)^\top G_{t-1}^{-1} \big( \lambda \boldsymbol{\theta}^\star - S_{\text{diff},t-1} - S_{\text{eta},t-1} \big)
$$
$$
\text{where} \qquad S_{\text{diff},t-1} = \sum_{\tau=1}^{t-1} \big( \widehat{\boldsymbol{b}}_\tau \otimes \boldsymbol{\varphi}(a_\tau, \boldsymbol{x}_\tau) \big) \big( (\boldsymbol{b}_\tau - \widehat{\boldsymbol{b}}_\tau) \otimes \boldsymbol{\varphi}(a_\tau, \boldsymbol{x}_\tau) \big)^\top \boldsymbol{\theta}^\star \quad \text{and} \quad S_{\text{eta},t-1} = \sum_{\tau=1}^{t-1} \big( \widehat{\boldsymbol{b}}_\tau \otimes \boldsymbol{\varphi}(a_\tau, \boldsymbol{x}_\tau) \big) \eta'_\tau(a_\tau) .
$$

A Cauchy-Schwarz inequality for the inner product induced by $G_{t-1}^{-1}$, together with a triangle inequality, entails

$$
\left| \big( \widehat{\boldsymbol{b}}_t \otimes \boldsymbol{\varphi}(a, \boldsymbol{x}_t) \big)^\top \big( \boldsymbol{\theta}^\star - \widehat{\boldsymbol{\theta}}_{t-1} \big) \right| \leqslant \left\| \widehat{\boldsymbol{b}}_t \otimes \boldsymbol{\varphi}(a, \boldsymbol{x}_t) \right\|_{G_{t-1}^{-1}} \left( \lambda \| \boldsymbol{\theta}^\star \|_{G_{t-1}^{-1}} + \| S_{\text{diff},t-1} \|_{G_{t-1}^{-1}} + \| S_{\text{eta},t-1} \|_{G_{t-1}^{-1}} \right) .
$$

Since $G_{t-1} \succeq \lambda \boldsymbol{I}_{dH}$,

$$
\forall \boldsymbol{u} \in \mathbb{R}^{dH} , \qquad \| \boldsymbol{u} \|_{G_{t-1}^{-1}} \leqslant \frac{\| \boldsymbol{u} \|_2}{\sqrt{\lambda}} . \qquad (13)
$$

Therefore, using Assumption 2.1,

$$
\lambda \| \boldsymbol{\theta}^\star \|_{G_{t-1}^{-1}} \leqslant \sqrt{\lambda} \, \| \boldsymbol{\theta}^\star \|_2 , \leqslant \sqrt{\lambda H} \, C_{\boldsymbol{\theta}^\star}
$$

and, again by Assumption 2.1, by Equation (6), and by a triangular inequality,

$$
\| S_{\text{diff},t-1} \|_{G_{t-1}^{-1}} \leqslant \frac{1}{\sqrt{\lambda}} \left\| \sum_{\tau=1}^{t-1} \sum_{h \in [H]} \overbrace{\boldsymbol{\varphi}(a_\tau, \boldsymbol{x}_\tau)^\top \boldsymbol{\theta}_h^\star}^{|\cdot| \leqslant 1} \big( \boldsymbol{b}_\tau(h) - \widehat{\boldsymbol{b}}_\tau(h) \big) \overbrace{\widehat{\boldsymbol{b}}_\tau \otimes \boldsymbol{\varphi}(a_\tau, \boldsymbol{x}_\tau)}^{\| \cdot \|_2 \leqslant 1} \right\|_2 \leqslant \frac{1}{\sqrt{\lambda}} \sum_{\tau=1}^{t-1} \left\| \boldsymbol{b}_\tau - \widehat{\boldsymbol{b}}_\tau \right\|_1 .
$$

As for the final term $\| S_{\text{eta},t-1} \|_{G_{t-1}^{-1}}$, it is exactly of the form discussed in Abbasi-Yadkori et al. (2011, Theorem 1 and Lemma 10, recalled below as Lemma A.3), with

$$
d' = dH , \qquad G = \lambda \boldsymbol{I}_{dH} , \qquad X_\tau = \widehat{\boldsymbol{b}}_\tau \otimes \boldsymbol{\varphi}(a_\tau, \boldsymbol{x}_\tau) , \qquad \eta_\tau = \eta'_\tau(a_\tau) , \qquad \mathcal{F}_\tau = \mathcal{F}'^{\text{all}}_{\tau+1} ;
$$

indeed, $a_\tau$ and $X_\tau$ are $\mathcal{F}_\tau'^{\text{all}}$–measurable while the $\eta_t'(a)$, and thus also $\eta_t'(a_t)$, are $\mathcal{F}_{\tau+1}'^{\text{all}}$–measurable and $v_\eta$–sub-Gaussian conditionally to $\mathcal{F}_\tau'^{\text{all}}$, as stated in Equation (10). In addition, Equation (6) guarantees that $\|X_\tau\|_2 \leqslant 1$. We get that probability at least $1 - \delta$, for all $t \geqslant 1$,

$$\left\| \sum_{\tau=1}^{t-1} \eta_\tau'(a_\tau) \, \widehat{\boldsymbol{b}}_\tau \otimes \boldsymbol{\varphi}(a_\tau, \boldsymbol{x}_\tau) \right\|_{G_{t-1}^{-1}} \leqslant v_\eta \sqrt{2\ln(1/\delta) + dH \ln\big(1 + t/(\lambda dH)\big)} \,.$$

The proof is concluded by collecting all the bounds above. $\qquad\square$

For the convenience of the reader, we restate the key classic deviation inequality used above. We recall that sub-Gaussian random variables are necessarily centered. Lemma 10 by Abbasi-Yadkori et al. (2011) is exactly Lemma A.5 stated at the end of this appendix.

**Lemma A.3** (Abbasi-Yadkori et al., 2011, Theorem 1 and Lemma 10). *Consider a filtration* $(\mathcal{F}_t)_{t \geqslant 0}$ *and two stochastic processes, a scalar-valued process* $(\eta_t)_{t \geqslant 1}$ *such that* $\eta_t$ *is* $\mathcal{F}_t$–*measurable and* $v_\eta$–*sub-Gaussian conditionally to* $\mathcal{F}_{t-1}$, *and a* $d'$–*vector-valued process* $(X_t)_{t \geqslant 1}$ *such that* $X_t$ *is* $\mathcal{F}_{t-1}$–*measurable and* $\|X_t\|_2 \leqslant 1$ *a.s. For* $\lambda > 0$, *let*

$$S_t = \sum_{\tau=1}^t \eta_\tau X_\tau \qquad and \qquad G_t = \lambda \boldsymbol{I}_{d'} + \sum_{\tau=1}^t X_\tau X_\tau^\top \,.$$

*Then, with probability at least* $1 - \delta$, *for all* $t \geqslant 1$,

$$\|S_t\|_{G_t^{-1}} \leqslant v_\eta \sqrt{2\ln(1/\delta) + d' \ln\big(1 + t/(\lambda d')\big)} \,.$$

We consider the Box B strategy with a belief estimation subroutine satisfying Assumption 3.1 (for which we recall that the belief error function $U_{\text{belief}}$ is known) and the confidence bonuses, for $t \geqslant 2$ and $a \in \mathcal{A}$,

$$\varepsilon_{t,a} = U_{\text{belief}}(t, \delta/2) \tag{14}$$
$$+ \left\| \widehat{\boldsymbol{b}}_t \otimes \boldsymbol{\varphi}(a, \boldsymbol{x}_t) \right\|_{G_{t-1}^{-1}} \left( \frac{1}{\sqrt{\lambda}} \sum_{\tau=1}^{t-1} U_{\text{belief}}(\tau, \delta/2) + \sqrt{\lambda H} \, C_{\boldsymbol{\theta}^\star} + v_\eta \sqrt{2\ln(2/\delta) + dH \ln\big(1 + t/(\lambda dH)\big)} \right)$$

and $\varepsilon_{1,a} = 1 + \sqrt{d}/\lambda$. The confidence bonuses $\varepsilon_{t,a}$ correspond to the upper bounds of Lemma A.2, denoted by $\varepsilon_{t,a}'$ in the proof below, up to the replacements of $\delta$ by $\delta/2$ and of the unknown $\|\boldsymbol{b}_\tau - \widehat{\boldsymbol{b}}_\tau\|_1$ by their high-probability bounds $U_{\text{belief}}(\tau, \delta/2)$. Recall the statement of our theorem, which we may now prove below.

**Theorem A.1.** *Assume that the horizon* $T$ *is known to the learner and fix* $\delta \in (0, 1)$. *Consider the strategy of Box B with a belief estimation subroutine satisfying Assumption 3.1, with* $\lambda = T^{1/2}$ *and with the confidence bonuses* (14). *Then, under the boundedness stated in Assumption 2.1 and under the sub-Gaussian noise assumption* (10), *with probability at least* $1 - \delta$, *up to poly-log factors,*

$$R_T = \widetilde{\mathcal{O}}\big(T^{3/4}\big) \,,$$

*where a closed-form expression of the regret bound may be found in the proof, see Equation* (21).

*Proof.* We denote by

$$\varepsilon_{t,a}' = \left\| \boldsymbol{b}_t - \widehat{\boldsymbol{b}}_t \right\|_1$$
$$+ \left\| \widehat{\boldsymbol{b}}_t \otimes \boldsymbol{\varphi}(a, \boldsymbol{x}_t) \right\|_{G_{t-1}^{-1}} \left( \frac{1}{\sqrt{\lambda}} \sum_{\tau=1}^{t-1} \left\| \boldsymbol{b}_\tau - \widehat{\boldsymbol{b}}_\tau \right\|_1 + \sqrt{\lambda H} \, C_{\boldsymbol{\theta}^\star} + v_\eta \sqrt{2\ln(2/\delta) + dH \ln\big(1 + t/(\lambda dH)\big)} \right)$$

the upper bound read in Lemma A.2 for the risk $\delta/2$, and let $\varepsilon_{1,a}' = 1 + \sqrt{d}/\lambda$. With this piece of notation, Lemma A.2 guarantees, in particular, that with probability at least $1 - \delta/2$, for all $t \geqslant 1$,

$$\max_{a \in \mathcal{A}} \sum_{h \in [H]} \boldsymbol{b}_t(h) \, \boldsymbol{\varphi}(a, \boldsymbol{x}_t)^\top \boldsymbol{\theta}_h^\star \leqslant \max_{a \in \mathcal{A}} \left\{ \varepsilon_{t,a}' + \sum_{h \in [H]} \widehat{\boldsymbol{b}}_t(h) \, \boldsymbol{\varphi}(a, \boldsymbol{x}_t)^\top \widehat{\boldsymbol{\theta}}_{t-1,h} \right\} \tag{15}$$

and

$$\sum_{h \in [H]} \widehat{\boldsymbol{b}}_t(h) \, \boldsymbol{\varphi}(a_t, \boldsymbol{x}_t)^\top \widehat{\boldsymbol{\theta}}_{t-1,h} \leqslant \varepsilon_{t,a_t}' + \sum_{h \in [H]} \boldsymbol{b}_t(h) \, \boldsymbol{\varphi}(a_t, \boldsymbol{x}_t)^\top \boldsymbol{\theta}_h^\star \,. \tag{16}$$

On the other hand, Assumption 3.1 ensures that with probability at least $1 - \delta/2$,

$$\forall t \in [T_0,\ T], \qquad \left\| \boldsymbol{b}_t - \widehat{\boldsymbol{b}}_t \right\|_1 \leqslant U_{\text{belief}}(t, \delta/2), \qquad \text{where} \qquad T_0 \overset{\text{def}}{=} \max\Big\{2,\ \lceil T_{\mathcal{B}, \boldsymbol{M}, \nu}\big(1 + \ln(2/\delta)\big)\rceil\Big\}; \qquad (17)$$

therefore, with probability at least $1 - \delta/2$,

$$\forall t \in [T_0,\ T], \qquad \sum_{\tau=1}^{t-1} \left\| \boldsymbol{b}_\tau - \widehat{\boldsymbol{b}}_\tau \right\|_1 \leqslant 2(T_0-1) + \sum_{\tau=1}^{t-1} U_{\text{belief}}(t, \delta/2) \quad \text{and} \quad \varepsilon'_{t,a} \leqslant \varepsilon_{t,a} + \frac{2(T_0-1)}{\sqrt{\lambda}} \left\| \widehat{\boldsymbol{b}}_t \otimes \boldsymbol{\varphi}(a, \boldsymbol{x}_t) \right\|_{G_{t-1}^{-1}}. \tag{18}$$

Equations (6) and (13) ensure that

$$\forall t \in [T_0,\ T], \qquad \left\| \widehat{\boldsymbol{b}}_t \otimes \boldsymbol{\varphi}(a, \boldsymbol{x}_t) \right\|_{G_{t-1}^{-1}} \leqslant \frac{\left\| \widehat{\boldsymbol{b}}_t \otimes \boldsymbol{\varphi}(a, \boldsymbol{x}_t) \right\|_2}{\sqrt{\lambda}} \leqslant \frac{1}{\sqrt{\lambda}} \qquad \text{thus, finally,} \qquad \varepsilon'_{t,a} \leqslant \varepsilon_{t,a} + 2(T_0-1)/\lambda. \tag{19}$$

By a union bound, substituting these bounds in Equations (15) and (16) and using the definition of $a_t$ as the argument of some maximum, we get that with probability at least $1 - \delta$, for all $t \in [T_0, T]$,

$$\max_{a \in \mathcal{A}} \sum_{h \in [H]} \boldsymbol{b}_t(h)\, \boldsymbol{\varphi}(a, \boldsymbol{x}_t)^\top \boldsymbol{\theta}_h^\star \leqslant \frac{2(T_0-1)}{\lambda} + \max_{a \in \mathcal{A}} \left\{ \varepsilon_{t,a} + \sum_{h \in [H]} \widehat{\boldsymbol{b}}_t(h)\, \boldsymbol{\varphi}(a, \boldsymbol{x}_t)^\top \widehat{\boldsymbol{\theta}}_{t-1,h} \right\}$$

$$= \frac{2(T_0-1)}{\lambda} + \varepsilon_{t,a_t} + \sum_{h \in [H]} \widehat{\boldsymbol{b}}_t(h)\, \boldsymbol{\varphi}(a_t, \boldsymbol{x}_t)^\top \widehat{\boldsymbol{\theta}}_{t-1,h}$$

and

$$\sum_{h \in [H]} \widehat{\boldsymbol{b}}_t(h)\, \boldsymbol{\varphi}(a_t, \boldsymbol{x}_t)^\top \widehat{\boldsymbol{\theta}}_{t-1,h} \leqslant \frac{2(T_0-1)}{\lambda} + \varepsilon_{t,a_t} + \sum_{h \in [H]} \boldsymbol{b}_t(h)\, \boldsymbol{\varphi}(a_t, \boldsymbol{x}_t)^\top \boldsymbol{\theta}_h^\star,$$

thus

$$\max_{a \in \mathcal{A}} \sum_{h \in [H]} \boldsymbol{b}_t(h)\, \boldsymbol{\varphi}(a, \boldsymbol{x}_t)^\top \boldsymbol{\theta}_h^\star - \sum_{h \in [H]} \boldsymbol{b}_t(h)\, \boldsymbol{\varphi}(a_t, \boldsymbol{x}_t)^\top \boldsymbol{\theta}_h^\star \leqslant \frac{4(T_0-1)}{\lambda} + 2\varepsilon_{t,a_t}.$$

For $t \leqslant T_0 - 1$, the difference above, called the instantaneous pseudo-regret, is always bounded by 2 due to the boundedness stated in Assumption 2.1.

Therefore, the regret (5) is bounded, with probability at least $1 - \delta$, by

$$R_T = \sum_{t=1}^{T} \left( \max_{a \in \mathcal{A}} \sum_{h \in [H]} \boldsymbol{b}_t(h)\, \boldsymbol{\varphi}(a, \boldsymbol{x}_t)^\top \boldsymbol{\theta}_h^\star - \sum_{h \in [H]} \boldsymbol{b}_t(h)\, \boldsymbol{\varphi}(a_t, \boldsymbol{x}_t)^\top \boldsymbol{\theta}_h^\star \right) \leqslant 2(T_0-1) + \sum_{t=T_0}^{T} \big(2\varepsilon_{t,a_t} + 4(T_0-1)/\lambda\big)$$

$$\leqslant 2(T_0-1) + \frac{4T(T_0-1)}{\lambda} + 2\sum_{t=1}^{T} U_{\text{belief}}(t, \delta/2) + 2\sum_{t=1}^{T} \left\| \widehat{\boldsymbol{b}}_t \otimes \boldsymbol{\varphi}(a, \boldsymbol{x}_t) \right\|_{G_{t-1}^{-1}} \tag{20}$$

$$\times \left( \frac{1}{\sqrt{\lambda}} \sum_{\tau=1}^{t-1} U_{\text{belief}}(\tau, \delta/2) + \sqrt{\lambda H}\, C_{\boldsymbol{\theta}^\star} + v_\eta \sqrt{2\ln(2/\delta) + dH\ln\big(1 + t/(\lambda dH)\big)} \right),$$

where we obtained the second inequality by substituting the expression of $\varepsilon_{t,a_t}$ and by replacing $T_0$ by 1 in the summation indices. We further bound the expression above by upper bounding each $t$ by $T$ in the sum of three terms in parentheses and by using

$$\sum_{t=1}^{T} \left\| \widehat{\boldsymbol{b}}_t \otimes \boldsymbol{\varphi}(a, \boldsymbol{x}_t) \right\|_{G_{t-1}^{-1}} \leqslant \sqrt{2\, dHT\, \ln\big(1 + T/(dH\lambda)\big)},$$

which follows from Lemma A.4 applied to the vectors $\boldsymbol{y}_\tau = \widehat{\boldsymbol{b}}_\tau \otimes \boldsymbol{\varphi}(a_\tau, \boldsymbol{x}_\tau)$, of dimension $dH$ and with Euclidean norm

smaller than 1 as indicated in Equation (6). We get the following final bound: with probability at least $1 - \delta$,

$$R_T \leqslant 2(T_0 - 1) + \frac{4T(T_0 - 1)}{\lambda} + 2\sum_{t=1}^{T} U_{\text{belief}}(t, \delta/2) + 2\sqrt{2\,dHT\,\ln\left(1 + T/(dH\lambda)\right)} \tag{21}$$

$$\times \left( \frac{1}{\sqrt{\lambda}} \sum_{\tau=1}^{T} U_{\text{belief}}(\tau, \delta/2) + \sqrt{\lambda H}\, C_{\boldsymbol{\theta}^\star} + v_\eta \sqrt{2\ln(2/\delta) + dH\ln\left(1 + T/(\lambda dH)\right)} \right).$$

This upper bound is of order $\sqrt{T}$ up to logarithmic factors when no belief estimation is required (i.e., when $U_{\text{belief}}$ is null and $T_0 = 1$, and for $\lambda$ given by a constant).

Keeping in mind that $T_0$ is of order $\ln T$ as far as its dependency on $T$ is concerned, the second part of Assumption 3.1 ensures that the upper bound of Equation (21) is of order, up to poly-logarithmic factors,

$$\widetilde{\mathcal{O}}\Big(T/\lambda + \sqrt{T} + \sqrt{T}\left(\sqrt{T/\lambda} + \sqrt{\lambda}\right)\Big) = \widetilde{\mathcal{O}}(T^{3/4}),$$

where we exploited the choice $\lambda = T^{1/2}$, which is the optimal choice of the form $T^a$ as far as orders of magnitude in $T$ up to poly-logarithmic factors are concerned. $\square$

For the sake of self-completeness, we state the so-called elliptic potential lemma, which is extracted from Abbasi-Yadkori et al. (2011, Section C); we actually even re-prove it here because we extend it later to staged updates in Appendix E.2 and base our extension on the classic proof below.

**Lemma A.4.** *Consider vectors $\boldsymbol{y}_t \in \mathbb{R}^d$ with $\|\boldsymbol{y}_t\|_2 \leqslant 1$, a parameter $\lambda \geqslant 1$, and the Gram matrices $V_0 = \lambda \boldsymbol{I}_d$ and*

$$V_t = \lambda \boldsymbol{I}_d + \sum_{\tau=1}^{t} \boldsymbol{y}_\tau \boldsymbol{y}_\tau^\top \quad \text{for} \quad t \geqslant 1.$$

*For all integers $T \geqslant 1$, we have:*

$$\sum_{t=1}^{T} \big\| V_{t-1}^{-1/2}\, \boldsymbol{y}_t \big\|_2 \leqslant \sqrt{2dT\ln\left(1 + T/(d\lambda)\right)}.$$

*Proof.* The classic proof is extracted from Abbasi-Yadkori et al. (2011, Section C). By the Cauchy-Schwarz inequality,

$$\sum_{t=1}^{T} \big\| V_{t-1}^{-1/2}\, \boldsymbol{y}_t \big\|_2 \leqslant \sqrt{T\sum_{t=1}^{T} \big\| V_{t-1}^{-1/2}\, \boldsymbol{y}_t \big\|_2^2} = \sqrt{T\sum_{t=1}^{T} \boldsymbol{y}_t^\top V_{t-1}^{-1} \boldsymbol{y}_t}. \tag{22}$$

Now, for $t \geqslant 1$, given that $V_{t-1} \succeq \lambda \boldsymbol{I}_d$, where $\lambda \geqslant 1$, and given that $\|\boldsymbol{y}_t\|_2 \leqslant 1$ by assumption, we have

$$0 \leqslant \boldsymbol{y}_t^\top V_{t-1}^{-1} \boldsymbol{y}_t \leqslant \frac{1}{\lambda} \|\boldsymbol{y}_t\|_2^2 \leqslant 1.$$

We resort to the inequality $u \leqslant 2\ln(1+u)$ for $u \in [0, 1]$, and then to Lemma A.5, to get

$$\sum_{t=1}^{T} \boldsymbol{y}_t^\top V_{t-1}^{-1} \boldsymbol{y}_t \leqslant 2\sum_{t=1}^{T} \ln\left(1 + \boldsymbol{y}_t^\top V_{t-1}^{-1} \boldsymbol{y}_t\right) = 2\sum_{t=1}^{T} \ln\frac{\det(V_t)}{\det(V_{t-1})} = 2\ln\frac{\det(V_T)}{\det(V_0)}, \tag{23}$$

where $\det(V_0) = \lambda^d$ and an upper bound on $\det(V_T)$ is given by Lemma A.6. Collecting all inequalities concludes the proof. $\square$

**Lemma A.5** (Matrix determinant lemma, see, e.g., Ding & Zhou, 2007, Lemma 1.1). *For a $d \times d$ invertible matrix $A$ and vectors $\boldsymbol{y}, \boldsymbol{y}' \in \mathbb{R}^d$,*

$$\det\left(A + \boldsymbol{y}'\boldsymbol{y}^\top\right) = \left(1 + \boldsymbol{y}^\top A^{-1}\boldsymbol{y}'\right)\det(A).$$

**Lemma A.6** (Determinant–trace inequality, see, e.g., Abbasi-Yadkori et al., 2011, Lemma 10). *With the notation and assumptions of Lemma A.4,*

$$\det(V_t) \leqslant (\lambda + t/d)^d.$$

# B. Challenges Overcome and Details on the Complex Statistical Dependencies at Stake

The most complex reward model (1) considers a direct dependency on the hidden state in the reward, through the $\boldsymbol{\theta}^\star_{h_t}$ term. We first explain why and how this model introduces complex statistical dependencies between hidden states and actions taken, through the observed rewards. We then highlight some of the technical challenges that we faced, and we explain, at a high level, how we overcame them.

## B.1. The Complex Statistical Dependencies Induced by Model (1)

Consider any filtration $\mathcal{F}_t$ such that $\sigma(\boldsymbol{x}_{1:t}) \subseteq \mathcal{F}_t \subseteq \mathcal{F}^{\text{obs}}_t$. In model (1), we have, by the assumptions on the noise terms and by the tower rule: for all $a \in \mathcal{A}$,

$$\mathbb{E}\big[r_t(a) \mid \mathcal{F}_t\big] = \boldsymbol{\varphi}(a, \boldsymbol{x}_t)^\top \sum_{h \in [H]} \mathbb{P}(h_t = h \mid \mathcal{F}_t)\, \boldsymbol{\theta}^\star_h. \tag{24}$$

This equality extends to $r_t(a_t)$ provided that $a_t$ is $\mathcal{F}_t$–measurable, which imposes a first constraint on $\mathcal{F}_t$. When $a_t$ is not $\mathcal{F}_t$–measurable, it is actually difficult to relate $\mathbb{E}\big[r_t(a_t) \mid \mathcal{F}_t\big]$ to the quantities (24); in particular, in general,

$$\mathbb{E}\big[r_t(a_t) \mid \boldsymbol{x}_{1:t}\big] \qquad \text{and} \qquad \boldsymbol{\varphi}(a_t, \boldsymbol{x}_t)^\top \sum_{h \in [H]} \mathbb{P}(h_t = h \mid \boldsymbol{x}_{1:t})\, \boldsymbol{\theta}^\star_h = \boldsymbol{\varphi}(a_t, \boldsymbol{x}_t)^\top \sum_{h \in [H]} \boldsymbol{b}_t(h)\, \boldsymbol{\theta}^\star_h$$

are different quantities.

To mimic the reduction to linear contextual bandits that was possible for the simplified reward model (3), the posterior probabilities $\mathbb{P}(h_t = h \mid \mathcal{F}_t)$ read in (24) should be estimated. This is our second constraint.

There is a tension between the two constraints and the two extreme candidates $\sigma(\boldsymbol{x}_{1:t})$ and $\mathcal{F}^{\text{obs}}_t$.

First, the natural filtration to get measurability of $a_t$ is $\mathcal{F}^{\text{obs}}_t$; indeed, the actions picked $a_t$ depend on the rewards obtained in the past, not only on the contexts. However, note that

$$\mathbb{P}(h_t = h \mid \boldsymbol{x}_{1:t}, a_t) \quad \text{and} \quad \boldsymbol{b}_t(h) = \mathbb{P}(h_t = h \mid \boldsymbol{x}_{1:t})$$

are different in general: the action $a_t$ depends on past rewards, and therefore depends on $h_{t-1}$, which entails some dependency between $a_t$ and $h_t$. More generally,

$$\mathbb{P}(h_t = h \mid \mathcal{F}_t) \quad \text{and} \quad \boldsymbol{b}_t(h) = \mathbb{P}(h_t = h \mid \boldsymbol{x}_{1:t})$$

are different in general when $a_t$ is $\mathcal{F}_t$–measurable.

Second, the natural filtration to estimate easily the beliefs $\mathbb{P}(h_t = h \mid \mathcal{F}_t)$ is $\sigma(\boldsymbol{x}_{1:t})$: we discard reward and may then open the toolbox of HMM estimation. Also, the pseudo-regret is formulated in terms of beliefs, i.e., posterior distributions based on $\sigma(\boldsymbol{x}_{1:t})$.

Our solution basically consists of introducing a carefully constructed filtration in the middle of these two extremes; we denote it by $\mathcal{U}_t$ and try now to get an idea of some nice properties we expect from it.

## B.2. Caveats Encountered when Estimating Beliefs

**Difficulties.** As explained above, it is handy that the action $a_t$ be $\mathcal{U}_t$–measurable. Now, to make useful decisions, the filtration $\mathcal{U}_t$ cannot be reduced to just contexts and thus needs to include at least some (quantities based on the) rewards. However, two issues arise.

First, posterior distributions with respect to $\mathcal{U}_t$ are then difficult to compute even with the knowledge of the underlying HMM parameters, and thus are even more difficult to estimate. Indeed, beliefs are typically computed based on Bayes' update rules (see Appendix C), and such rules work should then take rewards into account; this is what Zhou et al. (2021) performed in the case of Bernoulli rewards, but finding a general and computationally efficient solution beyond that case seems challenging.

Second, posterior distributions with respect to $\mathcal{U}_t$ should be close enough to the beliefs (i.e., from posterior distributions with respect to $\boldsymbol{x}_{1:t}$).

**Ideas.** The first idea is estimate posteriori distributions independently, only based on contexts, and discard any additional information coming from rewards; the loss in efficiency should be acceptable given the definition (5) of the pseudo-regret, which relies on beliefs $\boldsymbol{b}_t$, i.e., on quantities only based on the contexts. It turns out that such belief estimation only based on contexts is standard (see Appendix C): it suffices to estimate the HMM parameters, for which standard procedures exist, and estimation errors on the beliefs can be obtained as functions of the estimation errors of the HMM parameters.

The second, and main, idea is to design the filtration $\mathcal{U}_t$ so that

$$\boldsymbol{b}_t(h) = \mathbb{P}(h_t = h \mid \boldsymbol{x}_{1:t}) \qquad \text{and} \qquad \overline{\boldsymbol{b}}_t(h) = \mathbb{P}(h_t = h \mid \mathcal{U}_t)$$

are close, while maintaining the $\mathcal{U}_t$–measurability of $a_t$, i.e., including enough information on past rewards in $\mathcal{U}_t$.

**Solution.** The solution is essentially based on a typical property of HMMs: that they forget exponentially fast the initial state. Put differently, after a sufficient time, HMMs not initialized with the same distributions over states are almost equal in distributions. This property suggests to work in stages of sufficient length: we update only periodically the $\mathcal{U}_t$ to include a new sufficient statistic based on the rewards of the stage just passed.

These sufficient statistics consist of estimates of the parameters $(\boldsymbol{\theta}_h^\star)_{h \in [H]}$ of the reward model. The actions $a_t$ are picked based on the latest such estimates available (and on the estimated beliefs), in an optimistic fashion, through the consideration of an upper confidence bound.

### B.3. Other Technical Tool from Nelson et al. (2022)

A final ingredient in our solution, discussed next, is to leverage an approach by Nelson et al. (2022) for proving deviation inequalities in cases where the LinUCB approach by Abbasi-Yadkori et al. (2011) is not applicable. This approach relies on an $\mathbb{L}^2$–Markov inequality (suited for random variables with second-order moments, that are not necessarily sub-Gaussian).

Note that this approach was somewhat unnecessary in the setting of Nelson et al. (2022): provided that they strengthen a bit their assumption on the noise terms from a second-order moment assumption (Assumption 2.3) to a well-accepted sub-Gaussian (Assumption 2.2), Nelson et al. (2022) could have resorted to the classic LinUCB tools, with absolutely *no need* of HMM forgetting properties (like Assumption 4.1), to get their regret bound: we showed this in Appendix A.

However, as we explain in this section, this approach is particularly convenient in the case of our more complex reward model (1). To do so, we first introduce some stylized quantities to be controlled in the proofs.

**Stylized quantities considered.** The proofs—see, in particular, Appendix E.1.1 for the most complex reward model (1) and Lemma A.2 for the simplified version (3) by Nelson et al. (2022)—indicate that they key quantities to be controlled are of the stylized form

$$\Delta_T = \sum_{t=1}^{T} \left( r_t(a_t) - \boldsymbol{\varphi}(a, \boldsymbol{x}_t)^\top \sum_{h \in [H]} \boldsymbol{b}_t(h) \, \boldsymbol{\theta}_h^\star \right).$$

The exact quantities appearing in the proofs of the regret bounds are slightly more complicated (in particular, they are vector-valued) but we capture here the essence of the arguments.

**Simplified reward model.** By definition of the simplified reward model (3),

$$\Delta_T = \sum_{t=1}^{T} \eta_t'(a_t) \,,$$

where the $\eta_t'(a_t)$ are martingale increments, e.g., with respect to $\mathcal{F}_{t+1}'^{\text{obs}}$. Indeed, on the one hand, $\eta_t'(a_t)$ is $\mathcal{F}_{t+1}'^{\text{obs}}$–measurable as the difference between $r_t'(a_t)$, which is one of the variables generating $\mathcal{F}_{t+1}'^{\text{obs}}$, and some quantity measurable with respect to $\sigma(\boldsymbol{x}_{1:t})$ and $a_t$, where $a_t$ itself is $\mathcal{F}_t'^{\text{obs}}$–measurable. On the other hand, the assumption that the noise term is independent from the present and the past entails that $\mathbb{E}[\eta_t'(a) \mid \mathcal{F}_t'^{\text{obs}}] = 0$ for all $a \in \mathcal{A}$, so that, using again that $a_t$ is $\mathcal{F}_t'^{\text{obs}}$–measurable, we also have

$$\mathbb{E}[\eta_t'(a_t) \mid \mathcal{F}_t'^{\text{obs}}] = 0 \,.$$

Because the $\eta_t'(a_t)$ form bounded martingale increments, with probability at least $1 - \delta$, the martingale $\Delta_T$ is smaller than something of the order of $\sqrt{T \ln(1/\delta)}$, e.g., by the Hoeffding-Azuma inequality.

**Most complex reward model: issue.** The main issue in the most complex reward model (1) is that we do not deal with martingale increments, due, in particular, to the direct dependencies of rewards on hidden states. More precisely, we have, by the definition (1), that

$$
\Delta_T = \sum_{t=1}^{T} \boldsymbol{\varphi}(a_t, \boldsymbol{x}_t)^\top \left( \boldsymbol{\theta}_{h_t}^\star - \sum_{h \in [H]} \boldsymbol{b}_t(h)\, \boldsymbol{\theta}_h^\star \right) + \sum_{t=1}^{T} \eta_t(a_t)\,.
$$

An argument similar to above shows that the $\eta_t(a_t)$ are martingale increments with respect to $\mathcal{F}_{t+1}^{\mathrm{all}}$, thus their sum is controlled.

The remaining question is thus to control

$$
\sum_{t=1}^{T} d_t\,, \qquad \text{where} \qquad d_t = \boldsymbol{\varphi}(a_t, \boldsymbol{x}_t)^\top \left( \boldsymbol{\theta}_{h_t}^\star - \sum_{h \in [H]} \boldsymbol{b}_t(h)\, \boldsymbol{\theta}_h^\star \right).
$$

However, the $d_t$ do not form martingale increments, nor are sufficiently close to martingale increments. The issue mostly lies in guaranteeing that $d_t$ is $\mathcal{F}_{t+1}$–measurable; it is straightforward to find filtrations such that $\mathbb{E}\big[d_t \mid \mathcal{F}_t\big]$ is close to 0, e.g., $\mathcal{F}_t = \mathcal{U}_t$:

$$
\mathbb{E}\big[d_t \mid \mathcal{U}_t\big] = \boldsymbol{\varphi}(a_t, \boldsymbol{x}_t)^\top \left( \sum_{h \in [H]} \big(\overline{\boldsymbol{b}}_t(h) - \boldsymbol{b}_t(h)\big)\boldsymbol{\theta}_h^\star \right),
$$

which is small as all terms $\overline{\boldsymbol{b}}_t(h) - \boldsymbol{b}_t(h)$ are small, see Appendix B.2.

Any natural filtration $\mathcal{F}_t$ we would consider (and $\mathcal{U}_t$ in particular) includes contexts $\boldsymbol{x}_{1:t}$ and is such that $a_t$ is $\mathcal{F}_t$–measurable. For such filtration, the requirement of $\mathcal{F}_{t+1}$–measurability of $d_t$ entails that $\boldsymbol{\theta}_{h_t}^\star$ should be $\mathcal{F}_{t+1}$–measurable. Say, for simplicity, that $h_t$ is $\mathcal{F}_{t+1}$–measurable.

With the same arguments as above, the constraint $\mathbb{E}\big[d_t \mid \mathcal{F}_t\big]$ close to 0 would then impose that

$$
\boldsymbol{b}_t(h) = \mathbb{P}(h_t = h \mid \boldsymbol{x}_{1:t}) \qquad \text{and} \qquad \mathbb{P}(h_t = h \mid \mathcal{F}_t)
$$

should be close, but $\mathbb{P}(h_t = h \mid \mathcal{F}_t)$ corresponds to a quantity of the form $\mathbb{P}(h_t = h \mid h_{t-1}, \boldsymbol{x}_t)$ and is likely to differ much from $\boldsymbol{b}_t(h)$. Indeed, the former is mostly a function of $h_{t-1}$, while the latter depends only weakly on the underlying states due to the HMM forgetting properties.

**Most complex reward model: solution.** It turns out that we do not need that the $d_t$ are sufficiently close to martingale increments to control their sum: via Markov's inequality, it suffices that their sum is small in $\mathbb{L}^2$–norm, e.g., of the order of $T^a$ for $a < 1$. Then, with high probability, the sum itself is small: with probability at least $1 - \delta$

$$
\sum_{t=1}^{T} d_t \leqslant \sqrt{\frac{\mathbb{E}\big[ (d_1^2 + \ldots + d_T^2) \big]}{\delta}} \leqslant \sqrt{\frac{\mathcal{O}(T^a)}{\delta}}\,.
$$

Nelson et al. (2022) illustrate this for the sum of noise terms $\eta_t'(a)$ in their simplified model, when relaxing the sub-Gaussian noise Assumption 2.2 to Assumption 2.3 on bounded conditional second-order moments. This relaxation seems unimportant and we provide a simple and straightforward-to-prove regret bound under Assumption 2.2 (see Appendix A). Yet, the analysis that Nelson et al. (2022) developed, while not fully useful for them, is powerful and may be mimicked (and extended) to handle the sum of $d_t$.

Indeed, we write

$$
\mathbb{E}\left[ \left( \sum_{t=1}^{T} d_t \right)^2 \right] = \underbrace{\sum_{t=1}^{T} \mathbb{E}\big[d_t^2\big]}_{\text{of order } T} + 2 \sum_{1 \leqslant t < t' \leqslant T} \mathbb{E}\big[d_t d_{t'}\big]\,,
$$

where we used Assumption 2.1, i.e., the boundedness of $d_t$, to see that the sum of expected square terms is of order $T$. We get, by the tower rule and by the fact that $\boldsymbol{b}_t(h)$ and $\boldsymbol{b}_{t'}(h)$ are $\mathcal{U}_{t'}$–measurable:

$$\mathbb{E}\big[d_t d_{t'}\big] = \mathbb{E}\Big[\mathbb{E}\big[d_t d_{t'} \mid \mathcal{U}_{t'}\big]\Big] = \mathbb{E}\left[\boldsymbol{\varphi}(a_t, \boldsymbol{x}_t)^\top \boldsymbol{\theta}^\star_{h_t} \boldsymbol{\varphi}(a_{t'}, \boldsymbol{x}_{t'})^\top \left(\boldsymbol{\theta}^\star_{h_{t'}} - \sum_{h \in [H]} \overline{\boldsymbol{b}}_{t'}(h)\, \boldsymbol{\theta}^\star_h\right)\right] + \psi\big(\overline{\boldsymbol{b}}_t - \boldsymbol{b}_{t'}\big),$$

where $\psi\big(\overline{\boldsymbol{b}}_t - \boldsymbol{b}_{t'}\big)$ is a term bounded by something of the order of $\|\overline{\boldsymbol{b}}_t - \boldsymbol{b}_{t'}\|_1$ and is therefore small, by the HMM forgetting property. The other term in the rewriting above involves differences between

$$\mathbb{P}\big(h_\tau = h \ \text{ and } \ h_{\tau'} = h' \mid \mathcal{U}_{\tau'}\big) \qquad \text{and} \qquad \overline{\boldsymbol{b}}_t,$$

which, again, are small when $t$ and $t'$ are sufficiently separated due to the HMM forgetting property.

**Formal proofs.** Formal proofs of all the vague statements of this appendix will be provided in the course of the proof of Theorem 4.2, in Appendix E.

## C. Belief Estimation

The aim of this section is to detail why the following assumption made on the belief estimation subroutine is reasonable.

**Assumption 3.1** (belief estimation error). The belief estimation procedure $\mathcal{B}$ is such that for all hidden Markov chains $(\boldsymbol{\pi}, \boldsymbol{M}, \nu)$ in a wide class, there exist

- a constant $T_{\mathcal{B}, \boldsymbol{M}, \nu}$ not necessarily known to the learner,
- a fully known belief error function $U_{\mathrm{belief}}$ on $\{1, 2, \ldots\} \times (0, 1)$, where $U_{\mathrm{belief}}(t, \delta)$ depends logarithmically on $\delta$ and, up to poly-log factors, for each $\delta \in (0, 1)$,

$$\sum_{t \in [T]} U_{\mathrm{belief}}(t, \delta) = \widetilde{\mathcal{O}}\big(T^{1/2}\big) \,,$$

such that for all $\delta \in (0, 1)$, with probability at least $1 - \delta$, the following statements hold for all $t \geqslant T_{\mathcal{B}, \boldsymbol{M}, \nu}\big(1 + \ln(1/\delta)\big)$:

- first, the labeling of hidden states is consistent over the rounds considered;
- second, $\big\| \widehat{\boldsymbol{b}}_t - \boldsymbol{b}_t \big\|_1 \leqslant U_{\mathrm{belief}}(t, \delta)$.

More precisely, we show that this assumption holds at least for the large class of HMMs satisfying Assumption 3.2 below. To state the latter, we let $\boldsymbol{E}$ denote the emission matrix, indexed by $\mathcal{X} \times [H]$ and obtained by concatenating the emission distributions $\nu_h$ seen as $X$–dimensional column vectors as $h \in [H]$. We will assume, among others, that $\boldsymbol{E}$ has full column rank: this imposes, in particular, that $H$ is smaller than the cardinality of $\mathcal{X}$. Recall the notion of singular value: $\sigma$ is a singular value of $\boldsymbol{E}$ if $\sigma^2$ is an eigenvalue of the square matrix $\boldsymbol{E}^\top \boldsymbol{E}$.

**Assumption 3.2.** The context set $\mathcal{X}$ is finite, with cardinality denoted by $|\mathcal{X}| = X$.

The emission matrix $\boldsymbol{E}$ has full column rank with smallest singular value $\sigma_{\min}(\boldsymbol{E}) > 0$, and its smallest element satisfies

$$e_{\nu, \min} \stackrel{\mathrm{def}}{=} \min_{h \in [H]} \min_{x \in \mathcal{X}} \nu_h(x) > 0 \,.$$

The transition matrix $\boldsymbol{M}$ is invertible, with smallest eigenvalue denoted by $\sigma_{\min}(\boldsymbol{M}) > 0$, and the smallest element of $\boldsymbol{M}$ is positive: $\varepsilon_{\boldsymbol{M}} = \min_{h, h'} M_{h, h'} > 0$.

Finally, the initial distribution $\boldsymbol{\pi}$ is the (unique) stationary distribution of $\boldsymbol{M}$.

To that end, we consider a combination of a so-called spectral method (see its algorithmic statement in Appendix C.4 and see references after the statement of Lemma 3.3) to estimate the HMM parameters $\boldsymbol{M}$ and $\boldsymbol{E}$, i.e., the $\nu_h$ for $h \in [H]$, together with a Bayes' update rule to deduce estimated belief from these estimated parameters. We also add an alignment step (see Appendix C.2) to keep track of the hidden states.

**The Bayes' update rule.** More precisely, for $t \geqslant 1$, we denote by $\widehat{\boldsymbol{M}}_t$ and $\widehat{\nu}_{t,h}$ the estimates of $\boldsymbol{M}$ and of the $\nu_h$ obtained based on the contexts $\boldsymbol{x}_1, \ldots, \boldsymbol{x}_t$; we also consider some estimation $\widehat{\boldsymbol{\pi}}$ of the distribution of $h_1$, for instance, the uniform distribution over $[H]$. The Bayes' update rule then works as follows. For $\tau = 1$, we compute, for all $h \in [H]$:

$$\widehat{\boldsymbol{b}}_{t,1}(h) = \frac{\widehat{\nu}_{t,h}(\boldsymbol{x}_1) \, \widehat{\boldsymbol{\pi}}(h)}{\displaystyle\sum_{h'' \in [H]} \widehat{\nu}_{t,h''}(\boldsymbol{x}_1) \, \widehat{\boldsymbol{\pi}}(h'')} \,. \tag{25}$$

We then compute successively, for all $2 \leqslant \tau \leqslant t$, for all $h \in [H]$,

$$\widehat{\boldsymbol{b}}_{t,\tau}(h) = \frac{\widehat{\nu}_{t,h}(\boldsymbol{x}_\tau) \displaystyle\sum_{h' \in [H]} \widehat{\boldsymbol{b}}_{t,\tau-1}(h') \, \widehat{M}_{t,h',h}}{\displaystyle\sum_{h'' \in [H]} \widehat{\nu}_{t,h''}(\boldsymbol{x}_\tau) \displaystyle\sum_{h' \in [H]} \widehat{\boldsymbol{b}}_{t,\tau-1}(h') \, \widehat{M}_{t,h',h''}} \,. \tag{26}$$

We finally issue

$$\widehat{\boldsymbol{b}}_t = \widehat{\boldsymbol{b}}_{t,t} \,. \tag{27}$$

We call the successive updates above the Bayes' update rule. When $M$, $\pi$, and the $\nu_h$ are used instead of their estimates in the formulas above, we obtain the true beliefs $b_t$.

We detail in the rest of this appendix how to obtain the following result, which shows that Assumption 3.1 is reasonable.

**Lemma 3.3.** *Assumption 3.1 is satisfied for all hidden Markov chains of Assumption 3.2, for the spectral method (followed by an alignment step) combined with the Bayes' update rule, with the known belief error function*

$$U_{\text{belief}}(t, \delta) = \ln(t) \left( H\sqrt{X} \sqrt{\frac{2\ln\left(6Xt(t+1)/\delta\right)}{t}} + e^{-\sqrt{t-1}} \right)$$

*and the unknown threshold $T_{\mathcal{B},M,\nu}$ whose closed-form expression is provided in Equation (31).*

**References for the spectral method.** The HMM parameter estimation via spectral method is a standard procedure commonly used for bandits with latent states or for partially observable Markov decision processes [POMDP] where underlying states follow an HMM. It was proposed by Hsu et al. (2012) and further developed by Anandkumar et al. (2012) and Anandkumar et al. (2014). For instance, Zhou et al. (2021) and Azizzadenesheli et al. (2016) apply this the method proposed by Anandkumar et al. (2012, Section 4.2), in combination with the power iteration method from Anandkumar et al., 2014. We restate the algorithm of Anandkumar et al. (2012, Section 4.2) in Appendix C.4.

### C.1. Proof of Lemma 3.3: Belief-Estimation Error

We explain how the belief-estimation error bound from Lemma 3.3 follows from the application of two known results on HMM parameter estimation (one for the estimation of the parameters themselves, one for the guarantees induced on the estimation of the beliefs), together with two simple additions: an alignment step to ensure coherence of the labeling of hidden states and a twist to get a fully known belief error function.

First, as detailed in Appendix C.3, Azizzadenesheli et al. (2016, Theorem 3) and Zhou et al. (2021, Proposition 1, Appendix B) offer some estimator error guarantees, which can be instantiated under Assumption 3.2 as follows (keeping the notation of the second reference), with constants

$$C_1 = \frac{21}{\sigma^2}C_3 \quad \text{and} \quad C_2 = \frac{4}{\sigma}\left(\sqrt{H} + \frac{21H}{\sigma^2}\right)C_3, \quad \text{where} \quad C_3 = \frac{16}{\varepsilon_M^{3/2}}\left(1 + \frac{12}{\varepsilon_M^2\sigma^3} + \frac{256}{\varepsilon_M^2\sigma^2}\right).$$

Based on Anandkumar et al. (2014, Equation 28), we also define

$$C_0 = \min\left\{(56 \cdot 9 \cdot 102)^{-1}, (100 \cdot 168)^{-1}, \Delta'\right\},$$

where $\Delta' > 0$ is a numerical constant defined in Anandkumar et al. (2014, Lemma B.5 with $\Delta = 1/50$), though not in closed-form. Compared to the mentioned references, we rather consider the Frobenius norm instead of the spectral norm, which introduces an additional $\sqrt{H}$ factor in the estimation bound for $M$.

**Proposition C.1** (Instantiation of Zhou et al., 2021, Proposition 1, itself based on Azizzadenesheli et al., 2016, Theorem 3)**.** *Under Assumption 3.2, the threshold*

$$\widetilde{T}_{\mathcal{B},M,\nu} = 2\left(\frac{12}{\varepsilon_M^2\sigma^2}\right)^2 \left(\ln(2X) + 1\right) \max\left\{\frac{16H^{1/3}}{C_0^{2/3}\varepsilon_M^{1/3}}, \frac{3H}{C_0^2\varepsilon_M\sigma^2}, 1\right\} \tag{28}$$

*is such that for all $t \geqslant \widetilde{T}_{\mathcal{B},M,\nu}\left(1 + \ln(1/\delta)\right)$, with probability $1 - \delta$, the estimates $\tilde{M}_t$ and $\tilde{\nu}_t$ from the spectral method can be well computed and satisfy, up to some permutation $\rho$ of $[H]$,*

$$\left\|\tilde{M}_t^{(\rho)} - M\right\|_2 \leqslant C_2\sqrt{\frac{2H\ln(6X/\delta)}{t}} \qquad \text{and} \qquad \forall h \in [H], \qquad \|\tilde{\nu}_{t,\rho(h)} - \nu_h\|_2 \leqslant C_1\sqrt{\frac{2\ln(6X/\delta)}{t}},$$

*where $\tilde{M}_t^{(\rho)} = \left[\tilde{M}_{t,\rho(h),\rho(h')}\right]_{h,h' \in [H]}$.*

The result above is "up to some permutation $\rho$ of $[H]$": this underlines that the spectral method does absolutely not guarantee that what was called "state $h$" in round $t$ will correspond to the same 'state $h$" in round $t + 1$. In terms of beliefs, this means

that the belief function obtained from the estimates of Proposition C.1 are good up to the labeling of the hidden states. Since the end result is about the $\ell_1$–error between the estimated beliefs and the true beliefs, the ordering of hidden labels does not matter as long as that ordering is constant over time. This is what the alignment procedure described in Appendix C.2 will guarantee.

For now, we move to controlling the $\ell_1$–error between the estimated beliefs and the true beliefs, which corresponds to the second statement of Lemma 3.3 and is independent of the ordering of hidden labels, as it corresponds to some global evaluation.

De Castro et al. (2017) developed the following bound linking the estimation errors of the HMM parameters to the estimation error on the beliefs (again, the result holds up to permutations).

**Proposition C.2** (De Castro et al., 2017, Proposition 2.1, see also Zhou et al., 2021, Proposition 3). *Under Assumption 3.2, letting*

$$L_0 = 4\,\frac{1 - \varepsilon_{\boldsymbol{M}}}{\varepsilon_{\boldsymbol{M}}^2}\,, \qquad L_1 = 4\left(\frac{1 - \varepsilon_{\boldsymbol{M}}}{\varepsilon_{\boldsymbol{M}}}\right)^2 \frac{1}{e_{\nu,\min}}\,, \qquad L_2 = 4\,\frac{(1 - \varepsilon_{\boldsymbol{M}})^2}{\varepsilon_{\boldsymbol{M}}^3}\,,$$

*the beliefs $\tilde{\boldsymbol{b}}_t$ obtained from (a suitable permutation of) the estimates $\tilde{\boldsymbol{\pi}}$, and $\tilde{\boldsymbol{M}}_t$ and $\tilde{\nu}_{t,h}$ through the Bayes' rule (25)–(27) satisfy*

$$\left\|\tilde{\boldsymbol{b}}_t - \boldsymbol{b}_t\right\|_1 \leqslant L_0 \left(1 - \frac{\varepsilon_{\boldsymbol{M}}}{1 - \varepsilon_{\boldsymbol{M}}}\right)^{t-1} \left\|\tilde{\boldsymbol{\pi}} - \boldsymbol{\pi}\right\|_2 + L_1 \sum_{h \in [H]} \left\|\tilde{\nu}_{t,h} - \nu_h\right\|_1 + L_2 \left\|\tilde{\boldsymbol{M}}_t - \boldsymbol{M}\right\|_2\,.$$

We now combine Propositions C.1 and C.2 and perform some simple upper boundings, where the second follows from the Cauchy-Schwarz inequality:

$$\left\|\boldsymbol{\pi} - \tilde{\boldsymbol{\pi}}\right\|_2 = \sqrt{\sum_{h \in [H]} \underbrace{\left(\boldsymbol{\pi}(h) - \tilde{\boldsymbol{\pi}}(h)\right)^2}_{\in [-1,1]}} \leqslant \sqrt{\left\|\boldsymbol{\pi} - \tilde{\boldsymbol{\pi}}\right\|_1} \leqslant \sqrt{2} \qquad \text{and} \qquad \left\|\tilde{\nu}_{t,h} - \nu_h\right\|_1 \leqslant \sqrt{X}\,\left\|\tilde{\nu}_{t,h} - \nu_h\right\|_2\,.$$

We also perform union bounds and use

$$\frac{\delta}{t(t+1)} \qquad \text{instead of} \qquad \delta \tag{29}$$

at each round $t \geqslant 1$: this is to ensure that the result of Proposition C.1 holds simultaneously for all $t \geqslant \widetilde{T}_{\mathcal{B},\boldsymbol{M},\nu}$ with probability at least $1 - \delta$.

We get, from Propositions C.1 and C.2 that with probability $1 - \delta$,

$$\forall t \geqslant \widetilde{T}_{\mathcal{B},\boldsymbol{M},\nu}\Big(1 + \ln\big(t(t+1)/\delta\big)\Big)\,, \qquad \text{up to identifying a correct permutation,}$$

$$\left\|\tilde{\boldsymbol{b}}_t - \boldsymbol{b}_t\right\|_1 \leqslant \sqrt{2}L_0 \left(1 - \frac{\varepsilon_{\boldsymbol{M}}}{1 - \varepsilon_{\boldsymbol{M}}}\right)^{t-1} + \left(L_1 C_1 H \sqrt{X} + L_2 C_2 \sqrt{H}\right)\sqrt{\frac{2\ln\big(6Xt(t+1)/\delta\big)}{t}}\,. \tag{30}$$

The right-hand side cannot be our $U_{\text{belief}}$ function, as it depends on unknown quantities $\varepsilon_{\boldsymbol{M}}, C_1, C_2, L_0, L_1, L_2$ (the latter depend on the unknown HMM parameters). We do not follow the mitigations alluded at in Zhou et al. (2021, Section 3.3) or Azizzadenesheli et al. (2016, Remark 3), consisting of estimating these quantities (this looks as difficult as estimating the HMM parameters) or replacing them by some hyperparameters tuned by hand; we rather bound them as functions of $t$ and increase the threshold $\widetilde{T}_{\mathcal{B},\boldsymbol{M},\nu}$ to compensate for that.

We note that

$$\left(1 - \frac{\varepsilon_{\boldsymbol{M}}}{1 - \varepsilon_{\boldsymbol{M}}}\right)^{t-1} \leqslant \mathrm{e}^{-\sqrt{t-1}} \qquad \text{as soon as} \qquad \varepsilon'_{\boldsymbol{M}}(t-1) \leqslant -\sqrt{t-1}\,,$$

$$\text{i.e.,} \qquad t \geqslant 1 + 1/\big(\varepsilon'_{\boldsymbol{M}}\big)^2\,, \qquad \text{where} \qquad \varepsilon'_{\boldsymbol{M}} = \ln\left(1 - \frac{\varepsilon_{\boldsymbol{M}}}{1 - \varepsilon_{\boldsymbol{M}}}\right)\,.$$

Thus, we let

$$T_{\mathcal{B},\boldsymbol{M},\nu} = \max\left\{ 6\,\widetilde{T}_{\mathcal{B},\boldsymbol{M},\nu}\left(1 + \ln\big(\widetilde{T}_{\mathcal{B},\boldsymbol{M},\nu}\big)\right), \;\; \exp(L_1 C_1 + L_2 C_2), \;\; \exp\big(\sqrt{2}L_0\big), \right. \tag{31}$$

$$\left. 1 + 1/\big(\varepsilon'_{\boldsymbol{M}}\big)^2, \;\; 6\,T_\Delta\big(1 + \ln(T_\Delta)\big) \right\},$$

where we recall that $\widetilde{T}_{\mathcal{B},\boldsymbol{M},\nu}$ is defined in Equation (28) and where $T_\Delta$ is defined in Equation (35) below. Then, for $t \geqslant T_{\mathcal{B},\boldsymbol{M},\nu}\big(1 + \ln(1/\delta)\big)$, we have

$$L_1 C_1 + L_2 C_2 \leqslant \ln\big(T_{\mathcal{B},\boldsymbol{M},\nu}\big) \leqslant \ln\Big(t/\big(1 + \ln(1/\delta)\big)\Big) \leqslant \ln(t), \qquad \sqrt{2}L_0 \leqslant \ln(t), \qquad \left(1 - \frac{\varepsilon_{\boldsymbol{M}}}{1 - \varepsilon_{\boldsymbol{M}}}\right)^{t-1} \leqslant e^{-\sqrt{t-1}},$$

so that the right-hand side of Equation (30) is indeed smaller than the quantity $U_{\text{belief}}(t,\delta)$ defined in Lemma 3.3 for these $t \geqslant T_{\mathcal{B},\boldsymbol{M},\nu}\big(1 + \ln(1/\delta)\big)$. We also need to make sure that these $t$ satisfy the condition of Equation (30): this is the case as (see proof right below)

$$t \geqslant T_{\mathcal{B},\boldsymbol{M},\nu}\big(1 + \ln(1/\delta)\big) \qquad \text{entails} \qquad t \geqslant \widetilde{T}_{\mathcal{B},\boldsymbol{M},\nu}\Big(1 + \ln\big(t(t+1)/\delta\big)\Big). \tag{32}$$

This concludes the proof of the belief-estimation error part of the lemma up to identifying the suitable permutations, a topic which we discuss below in Appendix C.2, and up to proving (32), which do next.

**Proof of Equation (32).** From the assumption $t \geqslant T_{\mathcal{B},\boldsymbol{M},\nu}\big(1 + \ln(1/\delta)\big)$ and from the definition of $T_{\mathcal{B},\boldsymbol{M},\nu}$, which guarantees that $T_{\mathcal{B},\boldsymbol{M},\nu} \geqslant 6\,\widetilde{T}_{\mathcal{B},\boldsymbol{M},\nu}\big(1 + \ln(\widetilde{T}_{\mathcal{B},\boldsymbol{M},\nu})\big)$, we get

$$\frac{t}{\widetilde{T}_{\mathcal{B},\boldsymbol{M},\nu}} \geqslant 6\Big(1 + \ln\big(\widetilde{T}_{\mathcal{B},\boldsymbol{M},\nu}\big)\Big)\big(1 + \ln(1/\delta)\big) \geqslant 6\big(1 + \ln(1/\delta)\big). \tag{33}$$

From the intermediate inequality above, we also show later that

$$\frac{t}{\widetilde{T}_{\mathcal{B},\boldsymbol{M},\nu}} \geqslant 3\ln(t) \geqslant 2\ln(t+1), \tag{34}$$

where the second inequality holds because $\ln(t+1) \leqslant 1.5\,\ln(t)$ for $t \geqslant 6$, a condition that is satisfied in particular here. The conclusion then follows from combining the three bounds established above:

$$1 + \ln\left(\frac{t(t+1)}{\delta}\right) = \big(1 + \ln(1/\delta)\big) + \ln(t) + \ln(t+1) \leqslant \left(\frac{1}{6} + \frac{1}{3} + \frac{1}{2}\right)\frac{t}{\widetilde{T}_{\mathcal{B},\boldsymbol{M},\nu}} = \frac{t}{\widetilde{T}_{\mathcal{B},\boldsymbol{M},\nu}}.$$

It only remain to prove the intermediate inequality of Equation (34).

For the first inequality below, we use that $u \mapsto u/3 - \ln(u)$ is increasing for $u \geqslant 3$ and apply this property to the left inequality of Equation (33), and for the second inequality below, we use that $6 < \exp(2)$ and $1 + \ln(x) \leqslant x$ for $x > 0$:

$$\frac{t}{3\widetilde{T}_{\mathcal{B},\boldsymbol{M},\nu}} - \ln\left(\frac{t}{\widetilde{T}_{\mathcal{B},\boldsymbol{M},\nu}}\right) \geqslant \frac{6\Big(1 + \ln\big(\widetilde{T}_{\mathcal{B},\boldsymbol{M},\nu}\big)\Big)}{3} - \ln\underbrace{\Big(6\big(1 + \ln(\widetilde{T}_{\mathcal{B},\boldsymbol{M},\nu})\big)\Big)}_{\leqslant e^2 \widetilde{T}_{\mathcal{B},\boldsymbol{M},\nu}} \geqslant \ln\big(\widetilde{T}_{\mathcal{B},\boldsymbol{M},\nu}\big),$$

which rewrites as

$$\ln(t) = \ln\big(\widetilde{T}_{\mathcal{B},\boldsymbol{M},\nu}\big) + \ln\left(\frac{t}{\widetilde{T}_{\mathcal{B},\boldsymbol{M},\nu}}\right) \leqslant \frac{t}{3\widetilde{T}_{\mathcal{B},\boldsymbol{M},\nu}},$$

which is exactly the intermediate inequality of Equation (34).

## C.2. Proof of Lemma 3.3: Coherence Statement

In this section, we introduce an alignment step to ensure that, after a sufficiently large number of rounds, the latent-state labels of the estimated HMM parameters are consistent over time and thus can be mapped with some constant ordering of states. This also implies that the corresponding estimated belief vectors are expressed in one common latent state coordinate system, so that the words "a suitable permutation of" are not required anymore in Proposition C.2.

What follows was already alluded at, but not described in this level detail, by Azizzadenesheli et al. (2016, Appendix C, proof of Theorem 3, step 3). The latter reference raises the issue that "the columns of estimated matrices are up to different permutations over states, i.e., these matrices have different columns ordering", and suggests an alignment procedure.

**Algorithmic statement.** More formally, let $\mathfrak{S}$ denote the set of all permutations $\rho$ of $[H]$. At step $t+1$, when obtaining the estimates $\tilde{M}_{t+1}$ and $\tilde{\nu}_{t+1,h}$ considered in Proposition C.1, we transform them into the estimates $\widehat{M}_{t+1}$ and $\widehat{\nu}_{t+1,h}$ by picking a permutation $\rho_{t+1}$ and considering

$$\forall h \in [H], \quad \widehat{\nu}_{t+1,h} = \tilde{\nu}_{t+1,\rho_{t+1}(h)} \qquad \text{and} \qquad \widehat{M}_{t+1} = \big[\tilde{\mathbb{M}}_{t+1,\rho_{t+1}(h),\rho_{t+1}(h')}\big]_{h,h' \in [H]},$$

where the permutation is picked as $\rho_{t+1} \in \underset{\pi \in \mathfrak{S}}{\operatorname{argmin}} \ \underset{h \in [H]}{\max} \big\| \widehat{\nu}_{t,h} - \tilde{\nu}_{t+1,\pi(h)} \big\|_2$.

At step $t=1$, we leave estimators unchanged, i.e., pick $\rho_1$ given by the identity.

**Analysis.** We consider the same union bound as the one performed in Equation (29), so that, in particular,

$$\mathbb{P}\Bigg( \forall t \geqslant \widetilde{T}_{\mathcal{B},M,\nu}\Big(1 + \ln\big(t(t+1)/\delta\big)\Big), \quad \exists \rho'_t \in \mathfrak{S} \quad \text{s.t.}$$
$$\forall h \in [H], \qquad \|\tilde{\nu}_{t,\rho'_t(h)} - \nu_h\|_2 \leqslant C_1 \sqrt{\frac{2\ln\big(6Xt(t+1)/\delta\big)}{t}} \Bigg) \geqslant 1 - \delta \,.$$

Denote by

$$\Delta = \min_{h \neq h'} \|\nu_h - \nu_{h'}\|_2 \,;$$

this quantity is positive as, by Assumption 3.2, $E$ is full rank and thus, the emission distributions $\nu_h$ are, in particular, all different. Now, consider a time $T_\Delta$ such that

$$\forall t \geqslant T_\Delta\Big(1 + \ln\big(t(t+1)/\delta\big)\Big), \qquad C_1 \sqrt{\frac{2\ln\big(6Xt(t+1)/\delta\big)}{t}} < \frac{\Delta}{4} \,;$$

for instance,

$$T_\Delta = 1 + \left\lceil \frac{32\,C_1^2\ln(6X)}{\Delta^2} \right\rceil \tag{35}$$

is a suitable value, as for $t \geqslant T_\Delta\Big(1 + \ln\big(t(t+1)/\delta\big)\Big)$, and using that $\ln(6X) \geqslant 1$,

$$C_1 \sqrt{\frac{2\ln\big(6Xt(t+1)/\delta\big)}{t}} \leqslant C_1 \sqrt{2\,\frac{\ln(6X)\big(1 + \ln\big(t(t+1)/\delta\big)\big)}{T_\Delta\big(1 + \ln\big(t(t+1)/\delta\big)\big)}} = C_1 \sqrt{\frac{2\ln(6X)}{T_\Delta}} < \frac{\Delta}{4} \,.$$

We consider the same threshold $T_{\mathcal{B},M,\nu}$ as in Equation (31); in particular, given the definition of $T_{\mathcal{B},M,\nu}$, the proof of Equation (32) with $T_\Delta$ instead of $\widetilde{T}_{\mathcal{B},M,\nu}$ guarantees that

$$t \geqslant T_{\mathcal{B},M,\nu}\big(1 + \ln(1/\delta)\big) \qquad \text{entails} \qquad t \geqslant T_\Delta\Big(1 + \ln\big(t(t+1)/\delta\big)\Big) \,.$$

Together with Equation (32), we thus have proved so far that

$$\mathbb{P}\Big( \forall t \geqslant T_{\mathcal{B},M,\nu}\big(1 + \ln(1/\delta)\big), \quad \exists \rho'_t \in \mathfrak{S} \quad \text{s.t.} \quad \forall h \in [H], \qquad \|\tilde{\nu}_{t,\rho'_t(h)} - \nu_h\|_2 < \Delta/4 \Big) \geqslant 1 - \delta \,. \tag{36}$$

The claimed coherence can now be formally stated as follows.

**Lemma C.3.** *Under the same $1 - \delta$ probability event considered in the end of Appendix C.1, which includes the event of Equation (36), we have that for all $t \geqslant T_{\mathcal{B},\mathbf{M},\nu}\big(1 + \ln(1/\delta)\big)$, the permutation $\rho'_t$ of Equation (36) is unique, and so is the permutation $\rho_{t+1}$ defined in the algorithmic statement above. In addition, there exists a permutation $\bar{\rho}$ such that*

$$\Delta/4 > \max_{h \in [H]} \big\| \widehat{\nu}_{t,h} - \nu_{\bar{\rho}(h)} \big\|_2 \to 0 \qquad while \qquad \forall h \in [H], \quad \forall h' \neq \bar{\rho}(h), \qquad \big\| \widehat{\nu}_{t,h} - \nu_{h'} \big\|_2 > 3\Delta/4 \,;$$

*thus, the algorithm keeps track of the latent states and uses a consistent hidden-state labeling after $T_{\mathcal{B},\mathbf{M},\nu}\big(1 + \ln(1/\delta)\big)$.*

*Proof.* Denote $t_0 = T_{\mathcal{B},\mathbf{M},\nu}\big(1 + \ln(1/\delta)\big)$ and fix $t \geqslant t_0$. Consider a suitable permutation $\rho'_t$. We note that under the event of interest, by a triangle inequality,

$$\forall m' \neq \rho'_t(h), \qquad \|\tilde{\nu}_{t,m'} - \nu_h\|_2 \geqslant \|\nu_m - \nu_h\|_2 - \|\tilde{\nu}_{t,m'} - \nu_m\|_2 > 3\Delta/4 \,,$$

where we introduced $m$ such that $\rho'_t(m) = m'$, so that $\|\tilde{\nu}_{t,m'} - \nu_m\|_2 < \Delta/4$; in particular, $m \neq h$, so that $\|\nu_m - \nu_h\|_2 \geqslant \Delta$. This shows that $\rho'_t$ is unique.

Now define $\bar{\rho} = (\rho'_{t_0})^{-1} \circ \rho_{t_0}$. We prove by induction on $t \geqslant t_0$ that

$$\max_{h \in [H]} \big\| \widehat{\nu}_{t,h} - \nu_{\bar{\rho}(h)} \big\|_2 < \Delta/4 \,,$$

together with the uniqueness of $\rho_{t+1}$. For $t = t_0$, by definition, $\widehat{\nu}_{t_0,h} = \tilde{\nu}_{t_0,\rho_{t_0}(h)}$, so that

$$\max_{h \in [H]} \big\| \widehat{\nu}_{t_0,h} - \nu_{\bar{\rho}(h)} \big\|_2 = \max_{h \in [H]} \big\| \tilde{\nu}_{t_0,\rho_{t_0}(h)} - \nu_{(\rho'_{t_0})^{-1}(\rho_{t_0}(h))} \big\|_2$$

$$= \max_{h' \in [H]} \big\| \tilde{\nu}_{t_0,h'} - \nu_{(\rho'_{t_0})^{-1}(h')} \big\|_2 = \max_{h' \in [H]} \big\| \tilde{\nu}_{t_0,\rho'_{t_0}(h')} - \nu_{h'} \big\|_2 < \Delta/4 \,,$$

where we performed various re-indexations of $[H]$ based on permutations and where we used the definition of $\rho'_{t_0}$ to get the final inequality. Assume that the induction property holds at some $t \geqslant t_0$, and define $\pi_{t+1} = \rho'_{t+1} \circ \bar{\rho}$. Then, by the triangle inequality,

$$\max_{h \in [H]} \big\| \widehat{\nu}_{t,h} - \tilde{\nu}_{t+1,\pi_{t+1}(h)} \big\|_2 \leqslant \max_{h \in [H]} \big\| \widehat{\nu}_{t,h} - \nu_{\bar{\rho}(h)} \big\|_2 + \max_{h \in [H]} \big\| \tilde{\nu}_{t+1,\rho'_{t+1}(\bar{\rho}(h))} - \nu_{\bar{\rho}(h)} \big\|_2$$

$$\leqslant \max_{h \in [H]} \big\| \widehat{\nu}_{t,h} - \nu_{\bar{\rho}(h)} \big\|_2 + \max_{h \in [H]} \big\| \tilde{\nu}_{t+1,\rho'_{t+1}(h)} - \nu_h \big\|_2 < \Delta/2 \,,$$

where we used that the first maximum is smaller than $\Delta/4$ by the induction hypothesis, and that the second maximum is also smaller than $\Delta/4$ by the definition of $\rho'_{t+1}$. On the other hand, if $\pi \neq \pi_{t+1}$, then there exists $h \in [H]$ such that $\pi(h) \neq \rho'_{t+1}\big(\bar{\rho}(h)\big)$, thus $(\rho'_{t+1})^{-1}\big(\pi(h)\big) \neq \bar{\rho}(h)$; using a triangle inequality, we have

$$\big\| \tilde{\nu}_{t+1,\pi(h)} - \nu_{\bar{\rho}(h)} \big\|_2 \geqslant \big\| \nu_{(\rho'_{t+1})^{-1}(\pi(h))} - \nu_{\bar{\rho}(h)} \big\|_2 - \big\| \tilde{\nu}_{t+1,\pi(h)} - \nu_{(\rho'_{t+1})^{-1}(\pi(h))} \big\|_2$$

$$= \underbrace{\big\| \nu_{(\rho'_{t+1})^{-1}(\pi(h))} - \nu_{\bar{\rho}(h)} \big\|_2}_{>\Delta} - \underbrace{\big\| \tilde{\nu}_{t+1,\rho'_{t+1}(h')} - \nu_{h'} \big\|_2}_{<\Delta/4} > 3\Delta/4 \,,$$

where the $< \Delta/4$ part comes from the definition and uniqueness of $\rho'_{t+1}$ used with $h' = (\rho'_{t+1})^{-1}(\pi(h))$, and the $> \Delta$ part from the very definition of $\Delta$. Therefore, by another triangle inequality and another use of the induction hypothesis,

$$\big\| \widehat{\nu}_{t,h} - \tilde{\nu}_{t+1,\pi(h)} \big\|_2 \geqslant \big\| \tilde{\nu}_{t+1,\pi(h)} - \nu_{\bar{\rho}(h)} \big\|_2 - \big\| \widehat{\nu}_{t,h} - \nu_{\bar{\rho}(h)} \big\|_2 > 3\Delta/4 - \Delta/4 = \Delta/2 \,.$$

This shows the uniqueness of $\rho_{t+1}$ and its closed-form expression $\rho_{t+1} = \pi_{t+1} = \rho'_{t+1} \circ \bar{\rho}$. Finally, substituting this expression and using the definition of $\rho'_{t+1}$,

$$\forall h \in [H], \qquad \big\| \widehat{\nu}_{t+1,h} - \nu_{\bar{\rho}(h)} \big\|_2 = \big\| \tilde{\nu}_{t+1,\rho_{t+1}(h)} - \nu_{\bar{\rho}(h)} \big\|_2 = \big\| \tilde{\nu}_{t+1,\rho'_{t+1}(\bar{\rho}(h))} - \nu_{\bar{\rho}(h)} \big\|_2 < \Delta/4 \,,$$

which closes the induction and completes the proof. $\square$

## C.3. Details on How Proposition C.1 Follows from Existing Results

Proposition C.1 follows from various results scattered throughout Hsu et al. (2012), which introduced the spectral method, Anandkumar et al. (2012) and Anandkumar et al. (2014). Azizzadenesheli et al. (2016) and Zhou et al. (2021) extended the method to more complex settings involving Markov decision processes (of which HMMs are special cases) and they offered a synthetical view of the constants involved in the estimation bound (though some of these constants are larger or depend on more complex quantities due to the consideration of Markov decision processes, which involve actions for the learner). More precisely, we follow below the exposition by Azizzadenesheli et al. (2016, Lemma 5, Lemma 8, Theorem 3, Theorem 16) and most importantly, Zhou et al. (2021, Proposition 1), with the needed modifications: the constants $C_1$, $C_2$, $C_3$ and the threshold $\widetilde{T}_{\mathcal{B}, \boldsymbol{M}, \nu}$ of Section C.1 are as in Zhou et al. (2021, Appendix B) except for removing an unnecessary term $|\mathcal{A}|$ from $C_1$ (it only arises due to their more complex setting) and up to substituting upper or lower bounds on some quantities, as detailed below.

**First series of quantities.** Assumption 3.2 entails that the hidden states form an ergodic Markov chain, which thus admits a unique stationary distribution $\pi$, is geometrically mixing in the following sense and with the following parameters (see Kontorovich & Weiss, 2014, Krishnamurthy, 2016, Theorems 2.7.2 and 2.7.4, Zhou et al., 2021, Appendix B):

$$\forall t \geqslant 2, \qquad \max_{h' \in [H]} \sum_{h \in [H]} \big| \mathbb{P}(h_t = h \mid h_1 = h') - \pi(h) \big| \leqslant 4(1 - \varepsilon_{\boldsymbol{M}})^{t-1} \,.$$

Further, due to the boundedness of $\boldsymbol{M}$ and the fact that $h_1$ follows $\pi$, for all $h \in [H]$ and all $t \geqslant 1$,

$$\mathbb{P}(h_t = h) = \pi(h) = \sum_{h' \in [H]} \pi(h') \underbrace{\boldsymbol{M}_{h',h}}_{\geqslant \varepsilon_{\boldsymbol{M}}} \geqslant \varepsilon_{\boldsymbol{M}} \,.$$

**Second series of quantities.** We define the following multi-view matrices $\boldsymbol{A}_1, \boldsymbol{A}_2, \boldsymbol{A}_3 \in [0, 1]^{X \times H}$ for $t \geqslant 2$,

$$\forall x \in \mathcal{X}, \ \forall h \in [H], \qquad \begin{aligned} \boldsymbol{A}_1(x, h) &= \mathbb{P}(\boldsymbol{x}_{t-1} = x \mid h_t = h) \,, \\ \boldsymbol{A}_2(x, h) &= \mathbb{P}(\boldsymbol{x}_t = x \mid h_t = h) \,, \\ \boldsymbol{A}_3(x, h) &= \mathbb{P}(\boldsymbol{x}_{t+1} = x \mid h_t = h) \,, \end{aligned}$$

and we are interested in $\min\{\sigma_{\min}(\boldsymbol{A}_1), \sigma_{\min}(\boldsymbol{A}_2), \sigma_{\min}(\boldsymbol{A}_3)\}$, where $\sigma_{\min}(\boldsymbol{A}_i)$ is the smallest singular value of the matrix $\boldsymbol{A}_i$, for $i \in \{1, 2, 3\}$. We have $\boldsymbol{A}_2 = \boldsymbol{E}$ and $\boldsymbol{A}_3 = \boldsymbol{E}\boldsymbol{M}^\top$. The closed-form expression for $\boldsymbol{A}_1$ is slightly more complex as we have to go backwards in the HMM:

$$\boldsymbol{A}_1(x, h) = \mathbb{P}(\boldsymbol{x}_{t-1} = x \mid h_t = h) = \sum_{h' \in [H]} \overbrace{\mathbb{P}(\boldsymbol{x}_{t-1} = x \mid h_{t-1} = h', \ h_t = h)}^{\text{no dep. on } h \text{ and } = \boldsymbol{E}_{x,h'}} \mathbb{P}(h_{t-1} = h' \mid h_t = h) \,,$$

$$\text{where} \qquad \mathbb{P}(h_{t-1} = h' \mid h_t = h) = \frac{\mathbb{P}(h_t = h \mid h_{t-1} = h') \, \mathbb{P}(h_{t-1} = h')}{\mathbb{P}(h_t = h)} = \frac{\pi(h') \, \boldsymbol{M}_{h',h}}{\pi(h)}$$

(we recall that the Marhov chain of hidden states is initialized with the stationary distribution $\pi$), so that

$$\boldsymbol{A}_1 = \boldsymbol{E} \operatorname{diag}(\pi) \boldsymbol{M} \operatorname{diag}(\pi)^{-1} \,,$$

where $\operatorname{diag}$ transforms a vector into a square diagonal matrix with diagonal coefficients given by the vector. We now use that for two matrices $\boldsymbol{E}, \boldsymbol{E}'$ of compatible sizes, we have $\sigma_{\min}(\boldsymbol{E}\boldsymbol{E}') \geqslant \sigma_{\min}(\boldsymbol{E}) \, \sigma_{\min}(\boldsymbol{E}')$. Also, given that all components of $\pi$ are in the interval $[\varepsilon_{\boldsymbol{M}}, 1]$,

$$\sigma_{\min}\big(\operatorname{diag}(\pi)\big) \geqslant \varepsilon_{\boldsymbol{M}} \qquad \text{and} \qquad \sigma_{\min}\big(\operatorname{diag}(\boldsymbol{p}_t)^{-1}\big) \geqslant 1 \,.$$

We obtain $\sigma_{\min}(\boldsymbol{A}_2) = \sigma_{\min}(\boldsymbol{E})$, as well as

$$\sigma_{\min}(\boldsymbol{A}_1) \geqslant \sigma_{\min}(\boldsymbol{E}) \, \sigma_{\min}(\boldsymbol{M}) \, \varepsilon_{\boldsymbol{M}} \qquad \text{and} \qquad \sigma_{\min}(\boldsymbol{A}_3) \geqslant \sigma_{\min}(\boldsymbol{E}) \, \sigma_{\min}(\boldsymbol{M}) \,.$$

Given that $\boldsymbol{M}$ is a stochastic matrix, we have $\varepsilon_{\boldsymbol{M}} < 1$ and $\sigma_{\min}(\boldsymbol{M}) \leqslant 1$, and thus

$$\min\{\sigma_{\min}(\boldsymbol{A}_1), \sigma_{\min}(\boldsymbol{A}_2), \sigma_{\min}(\boldsymbol{A}_3)\} \geqslant \sigma \stackrel{\text{def}}{=} \sigma_{\min}(\boldsymbol{E}) \, \sigma_{\min}(\boldsymbol{M}) \, \varepsilon_{\boldsymbol{M}} \,.$$

**Third series of quantities.** Finally, we also introduce the co-occurrence matrix $\boldsymbol{A}_4 \in [0,1]^{X \times X}$:

$$\forall (i,j) \in \mathcal{X} \times \mathcal{X}, \qquad \boldsymbol{A}_4(i,j) = \mathbb{E}\big[\mathbb{1}_{\{\boldsymbol{x}_{t+1}=i\}} \mathbb{1}_{\{\boldsymbol{x}_{t-1}=j\}}\big] = \mathbb{P}(\boldsymbol{x}_{t+1} = i, \boldsymbol{x}_{t-1} = j)$$

and are interested in $\sigma_{\min}(\boldsymbol{A}_4)$. In a HMM, $\boldsymbol{x}_{t+1}$ and $\boldsymbol{x}_{t-1}$ are conditionally independent given $h_t$, as both are drawn independently conditional on $h_{t+1}$ and $h_{t-1}$; therefore, for any $i,j \in \mathcal{X}$,

$$\mathbb{P}(\boldsymbol{x}_{t+1} = i, \boldsymbol{x}_{t-1} = j) = \sum_{h \in [H]} \mathbb{P}(h_t = h)\, \mathbb{P}(\boldsymbol{x}_{t+1} = i \mid h_t = h)\, \mathbb{P}(\boldsymbol{x}_{t-1} = j \mid h_t = h),$$

and thus, $\boldsymbol{A}_4 = \boldsymbol{A}_3 \operatorname{diag}(\pi) \boldsymbol{A}_1^\top$. Similarly as above, this rewriting entails

$$\sigma_{\min}(\boldsymbol{A}_4) \geqslant \sigma_{\min}(\boldsymbol{A}_3)\, \sigma_{\min}\big(\operatorname{diag}(\pi)\big)\, \sigma_{\min}\big(\boldsymbol{A}_1^\top\big) \geqslant \big(\sigma_{\min}(\boldsymbol{E})\, \sigma_{\min}(\boldsymbol{M})\, \varepsilon_{\boldsymbol{M}}\big)^2 = \sigma^2\,.$$

### C.4. Reminder on the Spectral Method for MHH Parameter Estimation

Finally, for the sake of self-completedness, we recall the spectral method for HMM parameter estimation, which is the method considered in Proposition C.1. The exposition follows closely Anandkumar et al. (2012, Section 4.2).

Remember that we assume that the context $\mathcal{X}$ space is finite, with cardinality $X$. With no loss of generality, we may therefore identify it with the canonical vectors in $\mathbb{R}^X$, i.e., up to numbering the elements in $\mathcal{X}$ and substituting the $i$–th element, where $i \in [X]$, by the column vector $\boldsymbol{e}_i = (0,\ldots,0,1,0\ldots,0)^\top \in \mathbb{R}^X$, where the unique element 1 is in $i$–th position.

We extend the tensor product notation to products of three elements: for all vectors $\boldsymbol{u}, \boldsymbol{v}, \boldsymbol{w} \in \mathbb{R}^X$, the two-dimensional matrix $\boldsymbol{u} \otimes \boldsymbol{v}$ and the three-dimensional matrix $\boldsymbol{u} \otimes \boldsymbol{v} \otimes \boldsymbol{w}$ are defined component wise by

$$\forall (i,j,k) \in [X]^3, \qquad (\boldsymbol{u} \otimes \boldsymbol{v})_{i,j} = \boldsymbol{u}_i \boldsymbol{v}_j \quad \text{and} \quad (\boldsymbol{u} \otimes \boldsymbol{v} \otimes \boldsymbol{w})_{i,j,k} = \boldsymbol{u}_i \boldsymbol{v}_j \boldsymbol{w}_k\,.$$

We may now restate the special case of the estimation of HMM parameters as[1] detailed in Anandkumar et al. (2012, Section 4.2).

**Remark on the practical implementation.** In our numerical experiments, we will post-process $\tilde{M}_t$ and $\tilde{E}_t$ by clipping small negative entries and performing the needed normalizations so that they define valid emission and transition matrices.

---

[1] The indexing conventions are slightly different here and in their article, so that extra transpositions appear here.

> **Box C: Spectral estimation of HMM parameters from observations $\boldsymbol{x}_{1:t}$**
>
> **Inputs:** known number $H$ of hidden states; observations $\boldsymbol{x}_1, \boldsymbol{x}_2, \ldots, \boldsymbol{x}_t$, where $t \geqslant 3$, in the form of canonical vectors
>
> **Parameter:** invertible matrix $\Gamma \in \mathbb{R}^{H \times H}$, e.g., a random rotation matrix; denote by $\Gamma_{h,\,\cdot} \in \mathbb{R}^H$ its $h$–th row transposed into a column-vector
>
> **Output:** estimates $\tilde{\boldsymbol{E}}_t$ and $\tilde{\boldsymbol{M}}_t$ of the emission and transition matrices $\boldsymbol{E}$ and $\boldsymbol{M}$
>
> 1. Compute the empirical moments
>
> $$\tilde{P}_{3,1}^{(t)} = \frac{1}{t-2} \sum_{s=2}^{t-1} \boldsymbol{x}_{s+1} \otimes \boldsymbol{x}_{s-1}, \quad \tilde{P}_{3,2}^{(t)} = \frac{1}{t-2} \sum_{s=2}^{t-1} \boldsymbol{x}_{s+1} \otimes \boldsymbol{x}_s, \quad \tilde{P}_{3,1,2}^{(t)} = \frac{1}{t-2} \sum_{s=2}^{t-1} \boldsymbol{x}_{s+1} \otimes \boldsymbol{x}_{s-1} \otimes \boldsymbol{x}_s.$$
>
> 2. Compute $\tilde{U}_3, \tilde{U}_1 \in \mathbb{R}^{X \times H}$, the matrices whose columns are, respectively, the left and right singular vectors of $\tilde{P}_{3,1}^{(t)}$ associated with its largest $H$ singular values.
>
>    Compute $\tilde{U}_2 \in \mathbb{R}^{X \times H}$, the matrix whose columns are the right singular vectors of $\tilde{P}_{3,2}^{(t)}$ associated with its largest $H$ singular values.
>
> 3. Define the contraction of the third-order tensor $\tilde{P}_{3,1,2}^{(t)}(\boldsymbol{z})$ along its third mode for each vector $\boldsymbol{z} \in \mathbb{R}^X$ by
>
> $$\left[\tilde{P}_{3,1,2}^{(t)}(\boldsymbol{z})\right]_{i,j} := \sum_{k=1}^{X} \left[\tilde{P}_{3,1,2}^{(t)}\right]_{i,j,k} \boldsymbol{z}_k,$$
>
>    and let
>
> $$\tilde{B}_{3,1,2}^{(t)}(\boldsymbol{z}) = \left(\tilde{U}_3^\top \tilde{P}_{3,1,2}^{(t)}(\boldsymbol{z}) \tilde{U}_1\right) \left(\tilde{U}_3^\top \tilde{P}_{3,1}^{(t)} \tilde{U}_1\right)^{-1};$$
>
>    then, compute a matrix $\tilde{R} \in \mathbb{R}^{H \times H}$, with columns of unit Euclidean norm, such that
>
> $$\tilde{R}^{-1} \tilde{B}_{3,1,2}^{(t)}(\tilde{U}_2 \Gamma_{1,\,\cdot}) \tilde{R} = \text{diag}\left(\tilde{\lambda}_{1,1}, \tilde{\lambda}_{1,2}, \ldots, \tilde{\lambda}_{1,H}\right);$$
>
>    if this is not possible, redraw $\Gamma$ and repeat this step.
>
> 4. For each $h \in [H]$, using the matrix $\tilde{R}$ computed in Step 3, define $\tilde{\lambda}_{h,1}, \tilde{\lambda}_{h,2}, \ldots, \tilde{\lambda}_{h,H}$ as the diagonal entries of
>
> $$\tilde{R}^{-1} \tilde{B}_{3,1,2}^{(t)}(\tilde{U}_2 \Gamma_{h,\,\cdot}) \tilde{R}$$
>
>    and form the matrix $\tilde{L} \in \mathbb{R}^{H \times H}$ with entries $\tilde{L}_{h,j} := \tilde{\lambda}_{h,j}$ for all $h, j \in [H]$.
>
> 5. Define $\tilde{O}_t = \tilde{U}_2 \Gamma^{-1} \tilde{L}$ and output
>
> $$\tilde{\boldsymbol{E}}_t = \tilde{O}_t \quad \text{and} \quad \tilde{\boldsymbol{M}}_t = \left(\left(\tilde{U}_3^\top \tilde{O}_t\right)^{-1} \tilde{R}\right)^\top.$$

## D. HMM Forgetting Properties and Related Reminders

This appendix justifies the exponentially fast forgetting of the initial distribution of the HMM stated in Assumption 4.1.

**Assumption 4.1** (exponentially fast forgetting of initial condition). There exists a constant $\gamma \in [0, 1)$ so that, for all $s \leqslant t$, for all pairs $h, h' \in [H]$ of hidden states,

$$\sum_{j \in [H]} \left| \mathbb{P}_{\{h_s = h\}}(h_t = j \mid \boldsymbol{x}_{s+1:t}) - \mathbb{P}_{\{h_s = h'\}}(h_t = j \mid \boldsymbol{x}_{s+1:t}) \right| \leqslant 2\gamma^{t-s} \,.$$

This assumption involves quantities of the form $\mathbb{P}_E(E' \mid \mathcal{G})$, where $E$ and $E'$ are events and $\mathcal{G}$ is a $\sigma$–algebra, all defined on the same underlying probability space $(\Omega, \mathcal{F})$: we provide some reminders on such quantities—including their definitions—in Appendix D.1 below.

See also Appendix D.2 for reminders on why we only condition by $\boldsymbol{x}_{s+1:t}$ in the probability distributions above.

For now, we compare the assumption above to the alternative forgetting property used by Nelson et al. (2022, see Corollary C.4.1 therein), which is of the form: there exists $\gamma \in [0, 1)$ and a constant $C$, both depending on the HMM, such that

$$\mathbb{E}\left[ \sum_{j \in [H]} \left| \mathbb{P}_{\{h_s = h\}}(h_t = j \mid \boldsymbol{x}_{s+1:t}) - \mathbb{P}(h_t = j \mid \boldsymbol{x}_{s+1:t}) \right| \right] \leqslant C \exp\bigl(-\gamma(t - s)\bigr) \,. \tag{37}$$

This constant $\gamma$ is given by $\mathrm{e}^{-\gamma'/2}$, where $\gamma'$ is the minimal mixing rate of the transition matrix $\boldsymbol{M}$:

$$\gamma' = \min_{h', h'' \in [H]} \sum_{h \in [H]} \min\{M_{h,h'}, M_{h,h''}\} \,.$$

Nelson et al. (2022) apply some inequalities for Markov processes established by Boyen & Koller, 1998, see, in particular, Theorem 3 to show the property stated in Equation (37).

The forgetting property used by Nelson et al. (2022) is (by far) less demanding as the condition is to be satisfied in expectation compared to the one of Assumption 4.1. However, this is perfectly consistent with the fact that Nelson et al. (2022) only provide bounds in expectation while the present article instead aims for high-probability bounds (see the discussion in Section 1).

### D.1. Bayes' Formula for Probabilities Conditional to $\sigma$–Algebras

We assume that $P(E) > 0$ and let $\mathbb{P}_E = \mathbb{P}(\,\cdot\mid E)$ denote the conditional probability with respect to event $E$. This is a probability distribution over $(\Omega, \mathcal{F})$ and its conditional probability $\mathbb{P}_E(\,\cdot\mid \mathcal{G})$ with respect to the $\sigma$–algebra $\mathcal{G} \subseteq \mathcal{F}$ is thus well defined.

We recall in the lemma below how to apply rigorously Bayes' theorem in this context.

**Lemma D.1.** *With the notation above and under the condition $P(E) > 0$, we have*

$$\mathbb{P}(E \cap E' \mid \mathcal{G}) = \mathbb{P}(E \mid \mathcal{G}) \times \mathbb{P}_E(E' \mid \mathcal{G}) \,.$$

*Proof.* A characterization of the conditional expectation $\mathbb{E}[Z \mid \mathcal{G}]$ of a nonnegative random variable $Z \geqslant 0$ is that it is a $\mathcal{G}$–measurable random variable satisfying

$$\forall A \in \mathcal{G}, \qquad \mathbb{E}[Z \mathbb{1}_A] = \mathbb{E}\bigl[E[Z \mid \mathcal{G}] \mathbb{1}_A\bigr] \,. \tag{38}$$

We thus should prove that for all events $A \in \mathcal{G}$,

$$\mathbb{E}\bigl[\mathbb{1}_A \mathbb{P}(E \cap E' \mid \mathcal{G})\bigr] = \mathbb{E}\bigl[\mathbb{1}_A \mathbb{P}(E \mid \mathcal{G}) \times \mathbb{P}_E(E' \mid \mathcal{G})\bigr] \,. \tag{39}$$

By Equation (38) for the second equality, the left-hand side of Equation (39) can be rewritten as

$$\mathbb{E}\bigl[\mathbb{1}_A \mathbb{P}(E \cap E' \mid \mathcal{G})\bigr] = \mathbb{E}\bigl[\mathbb{1}_A \mathbb{E}[\mathbb{1}_E \mathbb{1}_{E'} \mid \mathcal{G}]\bigr] = \mathbb{E}\bigl[\mathbb{1}_A \mathbb{1}_E \mathbb{1}_{E'}\bigr] \,.$$

By $\mathcal{G}$–measurability of $\mathbb{P}_E(E' \mid \mathcal{G})$ for the second equality and by Equation (38) for the third equality, the right-hand side of Equation (39) can first be rewritten as

$$
\begin{aligned}
\mathbb{E}\big[\mathbb{1}_A\, \mathbb{P}(E \mid \mathcal{G}) \times \mathbb{P}_E(E' \mid \mathcal{G})\big] &= \mathbb{E}\big[\mathbb{1}_A\, \mathbb{E}[\mathbb{1}_E \mid \mathcal{G}] \times \mathbb{P}_E(E' \mid \mathcal{G})\big]\\
&= \mathbb{E}\big[\mathbb{1}_A\, \mathbb{E}[\mathbb{1}_E\, \mathbb{P}_E(E' \mid \mathcal{G}) \mid \mathcal{G}]\big] = \mathbb{E}\big[\mathbb{1}_A\, \mathbb{1}_E\, \mathbb{P}_E(E' \mid \mathcal{G})\big]\,.
\end{aligned}
$$

We continue the calculation by noting, for the first and last equalities below, that, by definition, $\mathbb{E}[\,\cdot\,\mathbb{1}_E] = \mathbb{P}(E) \times \mathbb{E}_E[\,\cdot\,\mathbb{1}_E]$, where $\mathbb{E}_E$ denotes the conditional expectation with respect to the event $E$, and, for the third equality, by resorting again to Equation (38) with $\mathbb{E}_E$:

$$
\mathbb{E}\big[\mathbb{1}_A\, \mathbb{1}_E\, \mathbb{P}_E(E' \mid \mathcal{G})\big] = \mathbb{E}_E\big[\mathbb{1}_A\, \mathbb{P}_E(E' \mid \mathcal{G})\big] = \mathbb{E}_E\big[\mathbb{1}_A\, \mathbb{E}_E[\mathbb{1}_{E'} \mid \mathcal{G}]\big] = \mathbb{E}_E[\mathbb{1}_A\, \mathbb{1}_{E'}] = \mathbb{E}[\mathbb{1}_A\, \mathbb{1}_E\, \mathbb{1}_{E'}]\,.
$$

The proof is concluded by collecting all equalities. $\qquad\square$

### D.2. Consequences of the Hidden Markov Model Formulation

Before we discuss (and prove) that Assumption 4.1 is natural, it is useful to state a reminder on how some conditionings may be simplified.

Recall the definition of $\mathcal{F}_t^{\mathrm{all}}$ from Section 2:

$$
\mathcal{F}_t^{\mathrm{all}} = \sigma\Bigg(\Big(h_\tau,\, \boldsymbol{x}_\tau,\, \big(\eta_\tau(a)\big)_{a\in\mathcal{A}}\Big)_{\tau\leqslant t-1},\, h_t,\, \boldsymbol{x}_t\Bigg);
$$

this is the richer filtration we consider.

The HMM model implies that, for $\tau < \tau'$, conditionally on $h_\tau$, the distribution of $h_{\tau'}$ is independent of past and present information, that is, $\mathcal{F}_\tau^{\mathrm{all}}$ and $\eta_\tau(a)$, but of course, not from future information corresponding to rounds $\tau + 1$ till $\tau'$, like the contexts $\boldsymbol{x}_{\tau+1:\tau'}$.

More formally and for example, we have, for $\tau < \tau'$,

$$
\mathbb{P}(h_{\tau'} = h' \mid \mathcal{F}_\tau^{\mathrm{all}},\, \eta_\tau(a_\tau),\, \boldsymbol{x}_{\tau+1:\tau'}) = \mathbb{P}(h_{\tau'} = h' \mid h_\tau,\, \boldsymbol{x}_{\tau+1:\tau'})\,.
$$

Via the same tools as in Appendix D.1, this entails, in particular, that for all $h \in [H]$,

$$
\mathbb{P}_{\{h_\tau=h\}}(h_{\tau'} = h' \mid \mathcal{F}_\tau^{\mathrm{all}},\, \eta_\tau(a_\tau),\, \boldsymbol{x}_{\tau+1:\tau'}) = \mathbb{P}_{\{h_\tau=h\}}(h_{\tau'} = h' \mid \boldsymbol{x}_{\tau+1:\tau'})\,. \tag{40}
$$

### D.3. Some Classic Condition Leading to Assumption 4.1

In this appendix, we show how Assumption 4.1 follows from the assumption below, considered by Cappé et al. (2005, Chapter 3) and rewritten in our context. Remember that we consider an homogeneous HMM (the distributions of transitions and emissions do not depend on the round), which is why the assumption is only stated with hidden states $h_1$ and $h_2$.

**Assumption D.2** (Cappé et al., 2005, Assumption 59, "strong mixing condition"). There exists a transition kernel $K : \mathcal{X} \times [H] \to (0, 1)$ and measurable functions $\zeta^-, \zeta^+ : \mathcal{X} \to (0, +\infty)$, with $\zeta^- \leqslant \zeta^+$, such that for all Borel sets $E$ of $\mathcal{X}$, and all $h, h' \in [H]$,

$$
\int_E \zeta^-(\boldsymbol{x})\, K(\boldsymbol{x}, h')\, \mathrm{d}\boldsymbol{x} \leqslant \mathbb{P}(h_2 = h',\, \boldsymbol{x}_2 \in E \mid h_1 = h) \leqslant \int_E \zeta^+(\boldsymbol{x})\, K(\boldsymbol{x}, h')\, \mathrm{d}\boldsymbol{x}\,.
$$

Cappé et al. (2005, Proposition 61) almost immediately entails the following lemma.

**Lemma D.3.** *Assumption D.2 entails Assumption 4.1, whenever*

$$
\sup_{x\in\mathcal{X}} \left(1 - \frac{\zeta^-(\boldsymbol{x})}{\zeta^+(\boldsymbol{x})}\right) < 1\,.
$$

*This is the case at least when $\mathcal{X}$ is finite.*

*Proof.* By homogeneity, it suffices to prove Assumption 4.1 for $s = 1$ and $t \geqslant 2$. Cappé et al. (2005, Proposition 61, based on Assumption D.2 above) guarantees that for all $t \geqslant 2$, for all pairs $h, h' \in [H]$ of hidden states,

$$\sum_{j \in [H]} \left| \mathbb{P}_{\{h_1 = h\}}(h_t = j \mid \boldsymbol{x}_{1:t}) - \mathbb{P}_{\{h_1 = h'\}}(h_t = j \mid \boldsymbol{x}_{1:t}) \right| \leqslant 2 \prod_{\tau=2}^{t} \left( 1 - \frac{\zeta^-(\boldsymbol{x}_\tau)}{\zeta^+(\boldsymbol{x}_\tau)} \right). \tag{41}$$

Under the HMM property, $h_t$ is independent of $\boldsymbol{x}_1$ given $h_1$; thus $\boldsymbol{x}_1$ can be removed from the conditionings in the left-hand sides of the above inequality (see Appendix D.2). Each of the terms $1 - \zeta^-(\boldsymbol{x}_\tau)/\zeta^+(\boldsymbol{x}_\tau)$ is non-negative, since $\zeta^- \leqslant \zeta^+$ by assumption. We further bound the right-hand side of Equation (41) by $2\gamma^{t-1}$ (which is the upper bound claimed by Assumption 4.1), where

$$\gamma = \sup_{x \in \mathcal{X}} \left( 1 - \frac{\zeta^-(\boldsymbol{x})}{\zeta^+(\boldsymbol{x})} \right).$$

When $\mathcal{X}$ is finite, Assumption D.2 imposed that $\zeta^-(\boldsymbol{x}) > 0$ for all $\boldsymbol{x} \in \mathcal{X}$, thus each of the finitely many terms in the defining maximum of $\gamma$ are strictly smaller than 1, therefore, so is $\gamma$. $\qquad\square$

# E. Proof of Theorem 4.2

The aim of this section is to prove the main result, which we restate below.

**Theorem 4.2.** *Assume the horizon $T$ is known to the learner and fix $\delta \in (0,1)$. Consider the strategy of Box A with a belief estimation subroutine satisfying Assumption 3.1, with parameters $\lambda = T^{3/4}$ and $\ell = \lceil T^{3/4} \rceil$, as well as the confidence bonuses $\varepsilon_{t,a} = 1 + \sqrt{d}/\lambda$ for $t \in [1, \ell]$ and for $t \geqslant \ell + 1$,*

$$
\varepsilon_{t,a} = U_{\text{belief}}(t, \delta/2) + \left\| G_{(s_t-1)\ell}^{-1}\left(\widehat{\boldsymbol{b}}_t \otimes \boldsymbol{\varphi}(a, \boldsymbol{x}_t)\right) \right\|_2 \overbrace{\left( \lambda\sqrt{H}\, C_{\boldsymbol{\theta}^\star} + 4\sqrt{\frac{s_T(s_t-1)(1+s_t\gamma)\ell}{\delta(1-\gamma)}} + \sqrt{\frac{4s_T}{\delta} C_\eta(s_t-1)\ell} \right.}^{=f_t}
$$
$$
\left. + \frac{2(s_t-1)\gamma}{1-\gamma} + \sum_{\tau=1}^{(s_t-1)\ell} U_{\text{belief}}(\tau, \delta/2) \right),
$$

*where $s_t = \lceil t/\ell \rceil$ denotes the stage to which round $t$ belongs. Then, under Assumption 2.1 (boundedness of rewards), Assumption 2.3 (noise with bounded conditional second-order moments), Assumption 3.1 (controlled belief estimation error), and Assumption 4.1 (exponentially fast forgetting), with probability at least $1 - \delta$, up to poly-log factors,*

$$
R_T = \widetilde{\mathcal{O}}\big(T^{7/8}\big).
$$

*A closed-form expression of the regret bound may be found in the proof, see Equations (48) and (49).*

The analysis follows the same structure as the one in Appendix A for the simplified model. Therein, the main piece in establishing the regret bound of Theorem A.1 consisted of building confidence intervals in Lemma A.2: the total regret bound was basically given by 2 times the sum of the errors margins of these confidence intervals. The counterpart to Lemma A.2 is the following. The only difference is that union bounds must be performed with greater care, hence the consideration of stage-varying confidence levels $1 - \delta_s$.

**Lemma E.1.** *Fix errors levels $\delta_s \in (0,1)$ for each $s \geqslant 1$. Under Assumptions 2.1–2.3–4.1, for all $s \geqslant 1$, for all $t \in [(s-1)\ell + 1,\ s\ell]$, with probability at least $1 - 2\delta_s$,*

$$
\forall a \in \mathcal{A}, \quad \left| \sum_{h \in [H]} \boldsymbol{b}_t(h)\boldsymbol{\varphi}(a, \boldsymbol{x}_t)^\top \boldsymbol{\theta}_h^\star - \sum_{h \in [H]} \widehat{\boldsymbol{b}}_t(h)\boldsymbol{\varphi}(a, \boldsymbol{x}_t)^\top \widehat{\boldsymbol{\theta}}_{(s-1)\ell,h} \right| \leqslant \varepsilon'_{t,s,\lambda,\delta_s,a}, \tag{42}
$$

*where $\varepsilon'_{t,1,\lambda,\delta_1,a} = 1 + \sqrt{d}/\lambda$ for $s = 1$, and for $s \geqslant 2$,*

$$
\varepsilon'_{t,s,\lambda,\delta_s,a} = \left\| \boldsymbol{b}_t - \widehat{\boldsymbol{b}}_t \right\|_1 + \left\| G_{(s-1)\ell}^{-1}\left(\widehat{\boldsymbol{b}}_t \otimes \boldsymbol{\varphi}(a, \boldsymbol{x}_t)\right) \right\|_2 \left( \lambda\sqrt{H}\, C_{\boldsymbol{\theta}^\star} + \sqrt{\frac{4(s-1)(1+s\gamma)\ell}{\delta_s(1-\gamma)}} + \sqrt{\frac{1}{\delta_s} C_\eta(s-1)\ell} \right.
$$
$$
\left. + \frac{2(s-1)\gamma}{1-\gamma} + \sum_{\tau=1}^{(s-1)\ell} \left\| \boldsymbol{b}_\tau - \widehat{\boldsymbol{b}}_\tau \right\|_1 \right).
$$

Appendix E.1 provides the proof of Lemma E.1, while Appendix E.2 proves Theorem 4.2 based on Lemma E.1.

## E.1. Proof of Lemma E.1

The proof adapts the one of Lemma A.2: the very beginning is similar, up to considering $\widehat{\boldsymbol{\theta}}_{(s-1)\ell,h}$ instead of $\widehat{\boldsymbol{\theta}}_{t-1,h}$, but the core of the proof is significantly different, as the LinUCB approach by Abbasi-Yadkori et al. (2011) cannot be followed anymore; see details on the reasons for this non-applicability in Appendix B.

The proof below actually details the claims and proof structure presented in Appendix B, partly based on some $\mathbb{L}^2$–Markov-based deviation inequality by Nelson et al. (2022).

*Proof.* The deterministic bound $1 + \sqrt{d}/\lambda$ actually holds for all $s$ and $t$, see the comments after the statement of Lemma A.2. The rest of the proof thus only covers the case $s \geqslant 2$. By a triangle inequality and by leveraging again Assumption 2.1, the

target quantity can be bounded by

$$
\left| \sum_{h \in [H]} \boldsymbol{b}_t(h) \boldsymbol{\varphi}(a, \boldsymbol{x}_t)^\top \boldsymbol{\theta}_h^\star - \sum_{h \in [H]} \widehat{\boldsymbol{b}}_t(h) \boldsymbol{\varphi}(a, \boldsymbol{x}_t)^\top \widehat{\boldsymbol{\theta}}_{(s-1)\ell, h} \right|
$$

$$
\leqslant \left| \sum_{h \in [H]} \left( \boldsymbol{b}_t(h) - \widehat{\boldsymbol{b}}_t(h) \right) \overbrace{\boldsymbol{\varphi}(a, \boldsymbol{x}_t)^\top \boldsymbol{\theta}_h^\star}^{|\cdot| \leqslant 1} \right| + \left| \sum_{h \in [H]} \widehat{\boldsymbol{b}}_t(h) \boldsymbol{\varphi}(a, \boldsymbol{x}_t)^\top \left( \boldsymbol{\theta}_h^\star - \widehat{\boldsymbol{\theta}}_{(s-1)\ell, h} \right) \right|
$$

$$
\leqslant \underbrace{\sum_{h \in [H]} \left| \left( \boldsymbol{b}_t(h) - \widehat{\boldsymbol{b}}_t(h) \right) \right|}_{= \|\boldsymbol{b}_t - \widehat{\boldsymbol{b}}_t\|_1} + \underbrace{\left| \sum_{h \in [H]} \widehat{\boldsymbol{b}}_t(h) \boldsymbol{\varphi}(a, \boldsymbol{x}_t)^\top \left( \boldsymbol{\theta}_h^\star - \widehat{\boldsymbol{\theta}}_{(s-1)\ell, h} \right) \right|}_{= (\widehat{\boldsymbol{b}}_t \otimes \boldsymbol{\varphi}(a, \boldsymbol{x}_t))^\top (\widehat{\boldsymbol{\theta}}_{(s-1)\ell} - \boldsymbol{\theta}^\star)}. \tag{43}
$$

The rest of the proof bounds the second term of the upper bound above. We first rewrite the differences $\boldsymbol{\theta}^\star - \widehat{\boldsymbol{\theta}}_{(s-1)\ell}$ in terms of the payoffs, using the definitions around Equation (7):

$$
\boldsymbol{\theta}^\star - \widehat{\boldsymbol{\theta}}_{(s-1)\ell} = G_{(s-1)\ell}^{-1} \left( G_{(s-1)\ell} \boldsymbol{\theta}^\star - \sum_{\tau=1}^{(s-1)\ell} \left( \widehat{\boldsymbol{b}}_\tau \otimes \boldsymbol{\varphi}(a_\tau, \boldsymbol{x}_\tau) \right) r_\tau(a_\tau) \right)
$$

$$
= G_{(s-1)\ell}^{-1} \left( \lambda \boldsymbol{I}_{dH} \boldsymbol{\theta}^\star - \sum_{\tau=1}^{(s-1)\ell} \left( \widehat{\boldsymbol{b}}_\tau \otimes \boldsymbol{\varphi}(a_\tau, \boldsymbol{x}_\tau) \right) \left( r_\tau(a_\tau) - \left( \widehat{\boldsymbol{b}}_\tau \otimes \boldsymbol{\varphi}(a_\tau, \boldsymbol{x}_\tau) \right)^\top \boldsymbol{\theta}^\star \right) \right).
$$

We substitute $r_\tau(a_\tau)$ in the expression above, but first rewrite it :

$$
r_\tau(a_\tau) \stackrel{\text{def}}{=} \boldsymbol{\varphi}(a_\tau, \boldsymbol{x}_\tau)^\top \boldsymbol{\theta}_{h_\tau}^\star + \eta_\tau(a_\tau)
$$

$$
= \boldsymbol{\varphi}(a_\tau, \boldsymbol{x}_\tau)^\top \left( \boldsymbol{\theta}_{h_\tau}^\star - \sum_{h' \in [H]} \bar{\boldsymbol{b}}_\tau(h') \boldsymbol{\theta}_{h'}^\star \right) + \sum_{h' \in [H]} \boldsymbol{\varphi}(a_\tau, \boldsymbol{x}_\tau)^\top \boldsymbol{\theta}_{h'}^\star \left( \bar{\boldsymbol{b}}_\tau(h') - \widehat{\boldsymbol{b}}_\tau(h') \right) + \left( \widehat{\boldsymbol{b}}_\tau \otimes \boldsymbol{\varphi}(a_\tau, \boldsymbol{x}_\tau) \right)^\top \boldsymbol{\theta}^\star + \eta_\tau(a_\tau).
$$

The second term in Equation (43) may therefore be rewritten as

$$
\sum_{h \in [H]} \widehat{\boldsymbol{b}}_t(h) \boldsymbol{\varphi}(a, \boldsymbol{x}_t)^\top \left( \boldsymbol{\theta}_h^\star - \widehat{\boldsymbol{\theta}}_{(s-1)\ell, h} \right) = \left( \widehat{\boldsymbol{b}}_t \otimes \boldsymbol{\varphi}(a, \boldsymbol{x}_t) \right)^\top \left( \widehat{\boldsymbol{\theta}}_{(s-1)\ell} - \boldsymbol{\theta}^\star \right)
$$

$$
= \left( \widehat{\boldsymbol{b}}_t \otimes \boldsymbol{\varphi}(a, \boldsymbol{x}_t) \right)^\top G_{(s-1)\ell}^{-1} \left( S'_{\text{diff},(s-1)\ell} + S'_{\text{belief},(s-1)\ell} + S'_{\text{eta},(s-1)\ell} - \lambda \boldsymbol{I}_{dH} \boldsymbol{\theta}^\star \right),
$$

where

$$
S'_{\text{diff},(s-1)\ell} = \sum_{\tau=1}^{(s-1)\ell} \boldsymbol{\varphi}(a_\tau, \boldsymbol{x}_\tau)^\top \left( \boldsymbol{\theta}_{h_\tau}^\star - \sum_{h' \in [H]} \bar{\boldsymbol{b}}_\tau(h') \boldsymbol{\theta}_{h'}^\star \right) \widehat{\boldsymbol{b}}_\tau \otimes \boldsymbol{\varphi}(a_\tau, \boldsymbol{x}_\tau),
$$

$$
S'_{\text{belief},(s-1)\ell} = \sum_{\tau=1}^{(s-1)\ell} \sum_{h' \in [H]} \boldsymbol{\varphi}(a_\tau, \boldsymbol{x}_\tau)^\top \boldsymbol{\theta}_{h'}^\star \left( \bar{\boldsymbol{b}}_\tau(h') - \widehat{\boldsymbol{b}}_\tau(h') \right) \widehat{\boldsymbol{b}}_\tau \otimes \boldsymbol{\varphi}(a_\tau, \boldsymbol{x}_\tau),
$$

$$
S'_{\text{eta},(s-1)\ell} = \sum_{\tau=1}^{(s-1)\ell} \eta_\tau(a_\tau) \widehat{\boldsymbol{b}}_\tau \otimes \boldsymbol{\varphi}(a_\tau, \boldsymbol{x}_\tau).
$$

The Euclidean norm of each of these three term is bounded in a series of lemmas below: $S'_{\text{diff},(s-1)\ell}$ in Lemma E.2, $S'_{\text{eta},(s-1)\ell}$ in Lemma E.3, and $S'_{\text{belief},(s-1)\ell}$ in Lemma E.4.

More precisely, we bound the second term in [Equation (43)](#) by a Cauchy-Schwarz inequality and a triangle inequality:

$$
\left| \sum_{h \in [H]} \widehat{\boldsymbol{b}}_t(h) \boldsymbol{\varphi}(a, \boldsymbol{x}_t)^\top \left( \boldsymbol{\theta}_h^\star - \widehat{\boldsymbol{\theta}}_{(s-1)\ell,h} \right) \right|
$$

$$
= \left| \left( \widehat{\boldsymbol{b}}_t \otimes \boldsymbol{\varphi}(a, \boldsymbol{x}_t) \right)^\top G_{(s-1)\ell}^{-1} \left( S'_{\mathrm{diff},(s-1)\ell} + S'_{\mathrm{belief},(s-1)\ell} + S'_{\mathrm{eta},(s-1)\ell} - \lambda \boldsymbol{I}_{dH} \boldsymbol{\theta}^\star \right) \right|
$$

$$
\leqslant \left\| G_{(s-1)\ell}^{-1} \left( \widehat{\boldsymbol{b}}_t \otimes \boldsymbol{\varphi}(a, \boldsymbol{x}_t) \right) \right\|_2 \left( \left\| S'_{\mathrm{diff},(s-1)\ell} \right\|_2 + \left\| S'_{\mathrm{eta},(s-1)\ell} \right\|_2 + \left\| S'_{\mathrm{belief},(s-1)\ell} \right\|_2 + \left\| \lambda \boldsymbol{\theta}^\star \right\|_2 \right),
$$

where $\| \lambda \boldsymbol{\theta}^\star \|_2 \leqslant \lambda \sqrt{H} \, C_{\boldsymbol{\theta}^\star}$ by [Assumption 2.1](#). [Lemma E.2](#) ensures that with probability at least $1 - \delta_s$,

$$
\left\| S'_{\mathrm{diff},(s-1)\ell} \right\|_2 = \left\| \sum_{\tau=1}^{(s-1)\ell} \boldsymbol{\varphi}(a_\tau, \boldsymbol{x}_\tau)^\top \left( \boldsymbol{\theta}_{h_\tau}^\star - \sum_{h' \in [H]} \bar{\boldsymbol{b}}_\tau(h') \boldsymbol{\theta}_{h'}^\star \right) \widehat{\boldsymbol{b}}_\tau \otimes \boldsymbol{\varphi}(a_\tau, \boldsymbol{x}_\tau) \right\|_2 \leqslant \sqrt{\frac{4(s-1)(1+s\gamma)\ell}{\delta_s(1-\gamma)}},
$$

where we performed some bounding to get a more compact bound. [Lemma E.3](#) ensures that with probability at least $1 - \delta_s$,

$$
\left\| S'_{\mathrm{eta},(s-1)\ell} \right\|_2 = \left\| \sum_{\tau=1}^{(s-1)\ell} \eta_\tau(a_\tau) \, \widehat{\boldsymbol{b}}_\tau \otimes \boldsymbol{\varphi}(a_\tau, \boldsymbol{x}_\tau) \right\|_2 \leqslant \sqrt{\frac{1}{\delta_s}} \, C_\eta (s-1)\ell .
$$

Finally, [Lemma E.4](#) guarantees that with probability 1,

$$
\left\| S'_{\mathrm{belief},(s-1)\ell} \right\|_2 = \left\| \sum_{\tau=1}^{(s-1)\ell} \sum_{h' \in [H]} \boldsymbol{\varphi}(a_\tau, \boldsymbol{x}_\tau)^\top \boldsymbol{\theta}_{h'}^\star \left( \bar{\boldsymbol{b}}_\tau(h') - \widehat{\boldsymbol{b}}_\tau(h') \right) \widehat{\boldsymbol{b}}_\tau \otimes \boldsymbol{\varphi}(a_\tau, \boldsymbol{x}_\tau) \right\|_2 \leqslant \frac{2(s-1)\gamma}{1-\gamma} + \sum_{\tau=1}^{(s-1)\ell} \left\| \boldsymbol{b}_\tau - \widehat{\boldsymbol{b}}_\tau \right\|_1 .
$$

The proof is concluded by collecting all the bounds above and by applying a union bound. $\square$

### E.1.1. BOUND ON $\left\| S'_{\mathrm{diff},s\ell} \right\|_2$

This is the most difficult term to bound and we follow the approach described at a high level in Appendix B: this approach constitutes the key technical contribution by [Nelson et al. (2022)](#). In particular, we apply Markov's inequality in $\mathbb{L}^2$–norm; this has the drawback that the associated high-probability bound depends on the risk $\delta$ through $\sqrt{1/\delta}$ instead of $\sqrt{\ln(1/\delta)}$ in the LinUCB approach (see Appendix A).

**Lemma E.2.** *Under Assumptions [2.1](#) and [4.1](#), for all $s \geqslant 1$,*

$$
\mathbb{E}\left[ \left\| S'_{\mathrm{diff},s\ell} \right\|_2^2 \right] = \mathbb{E}\left[ \left\| \sum_{\tau=1}^{s\ell} \boldsymbol{\varphi}(a_\tau, \boldsymbol{x}_\tau)^\top \left( \boldsymbol{\theta}_{h_\tau}^\star - \sum_{h \in [H]} \bar{\boldsymbol{b}}_\tau(h) \boldsymbol{\theta}_h^\star \right) \widehat{\boldsymbol{b}}_\tau \otimes \boldsymbol{\varphi}(a_\tau, \boldsymbol{x}_\tau) \right\|_2^2 \right] \leqslant 4s\ell + \frac{2s(s+1)\ell\gamma}{1-\gamma} .
$$

*Thus, for all $\delta \in (0,1)$, for each $s \geqslant 1$, with probability at least $1 - \delta$,*

$$
\left\| S'_{\mathrm{diff},s\ell} \right\|_2 = \left\| \sum_{\tau=1}^{s\ell} \boldsymbol{\varphi}(a_\tau, \boldsymbol{x}_\tau)^\top \left( \boldsymbol{\theta}_{h_\tau}^\star - \sum_{h \in [H]} \bar{\boldsymbol{b}}_\tau(h) \boldsymbol{\theta}_h^\star \right) \widehat{\boldsymbol{b}}_\tau \otimes \boldsymbol{\varphi}(a_\tau, \boldsymbol{x}_\tau) \right\|_2 \leqslant \sqrt{\frac{1}{\delta} \left( 4s\ell + \frac{2s(s+1)\ell\gamma}{1-\gamma} \right)} .
$$

*Proof.* The second inequality follows from the first one via Markov's inequality. We thus only prove the first inequality below.

*Step 1: Preparation.* Introduce the scalar-valued random variables

$$
z_\tau = \boldsymbol{\varphi}(a_\tau, \boldsymbol{x}_\tau)^\top \left( \boldsymbol{\theta}_{h_\tau}^\star - \sum_{h \in [H]} \bar{\boldsymbol{b}}_\tau(h) \boldsymbol{\theta}_h^\star \right), \qquad \text{where} \quad |z_\tau| \leqslant 2
$$

by Assumption 2.1 and the fact that $\bar{\boldsymbol{b}}_\tau$ is a probability distribution. Therefore, by developing the squared norm and by applying the inequalities above, as well as the bound of Equation (6), to the diagonal terms only, the target quantity may be rewritten as and bounded by

$$
\begin{aligned}
&\mathbb{E}\left[\left\|\sum_{\tau=1}^{s\ell} z_\tau\, \widehat{\boldsymbol{b}}_\tau \otimes \boldsymbol{\varphi}(a_\tau, \boldsymbol{x}_\tau)\right\|_2^2\right] \\
&= \sum_{\tau=1}^{s\ell}\sum_{\tau'=1}^{s\ell} \mathbb{E}\left[z_\tau z_{\tau'} \left(\widehat{\boldsymbol{b}}_\tau \otimes \boldsymbol{\varphi}(a_\tau, \boldsymbol{x}_\tau)\right)^\top \left(\widehat{\boldsymbol{b}}_{\tau'} \otimes \boldsymbol{\varphi}(a_{\tau'}, \boldsymbol{x}_{\tau'})\right)\right] \\
&\leqslant \underbrace{\sum_{\tau=1}^{s\ell} \mathbb{E}\left[z_\tau^2\right]}_{\leqslant 4s\ell} + 2\sum_{1\leqslant\tau<\tau'\leqslant s\ell} \mathbb{E}\left[z_\tau z_{\tau'} \underbrace{\left(\widehat{\boldsymbol{b}}_\tau \otimes \boldsymbol{\varphi}(a_\tau, \boldsymbol{x}_\tau)\right)^\top \left(\widehat{\boldsymbol{b}}_{\tau'} \otimes \boldsymbol{\varphi}(a_{\tau'}, \boldsymbol{x}_{\tau'})\right)}_{\text{to be dealt with}}\right].
\end{aligned}
\tag{44}
$$

We recall that we introduced $\mathcal{U}_t = \sigma\big(\boldsymbol{x}_{1:t}, \big(\widehat{\boldsymbol{\theta}}_{s\ell}\big)_{s\leqslant s_t-1}\big)$ for $t \geqslant 1$. Now, we note that the inner product in the cross terms above (marked as "to be dealt with") is $\sigma(\mathcal{U}_{\tau'})$–measurable, as, in particular, actions $a_\tau$ and $a_{\tau'}$ (since the algorithm proceeds in stages) and estimated beliefs $\widehat{\boldsymbol{b}}_{\tau'}$ and $\widehat{\boldsymbol{b}}_{\tau'}$ are so. By the Cauchy-Schwarz inequality and the bound of Equation (6), it is also seen to be smaller than 1. Therefore, by the tower rule, we further bound the $\tau < \tau'$ cross term above by

$$
\begin{aligned}
&\mathbb{E}\left[z_\tau z_{\tau'} \overbrace{\left(\widehat{\boldsymbol{b}}_\tau \otimes \boldsymbol{\varphi}(a_\tau, \boldsymbol{x}_\tau)\right)^\top \left(\widehat{\boldsymbol{b}}_{\tau'} \otimes \boldsymbol{\varphi}(a_{\tau'}, \boldsymbol{x}_{\tau'})\right)}^{\text{is } \sigma(\mathcal{U}_{\tau'})\text{–measurable}}\right] \\
&= \mathbb{E}\left[\mathbb{E}\left[z_\tau z_{\tau'} \mid \mathcal{U}_{\tau'}\right] \underbrace{\left(\widehat{\boldsymbol{b}}_\tau \otimes \boldsymbol{\varphi}(a_\tau, \boldsymbol{x}_\tau)\right)^\top \left(\widehat{\boldsymbol{b}}_{\tau'} \otimes \boldsymbol{\varphi}(a_{\tau'}, \boldsymbol{x}_{\tau'})\right)}_{\leqslant 1}\right] \leqslant \mathbb{E}\left[\left|\mathbb{E}\left[z_\tau z_{\tau'} \mid \mathcal{U}_{\tau'}\right]\right|\right].
\end{aligned}
$$

To get the claimed bound, we prove that for each $1 \leqslant \tau < \tau' \leqslant s\ell$,

$$
\left|\mathbb{E}\left[z_\tau z_{\tau'} \mid \mathcal{U}_{\tau'}\right]\right| \leqslant
\begin{cases}
2\gamma^{\tau'-\tau} & \text{if } \tau \geqslant (s_{\tau'}-1)\ell+1, \\
& \text{i.e., if } \tau \text{ belongs to the same stage as } \tau', \\
2\gamma^{\tau'-(s_{\tau'}-1)\ell} & \text{if } \tau \leqslant (s_{\tau'}-1)\ell, \\
& \text{i.e., if } \tau \text{ and } \tau' \text{ belong to different stages,}
\end{cases}
\tag{45}
$$

which we do next in the subsequent steps of the proof. Then, based on Equation (45), we obtain

$$
\begin{aligned}
&\sum_{1\leqslant\tau<\tau'\leqslant s\ell} \left|\mathbb{E}\left[z_\tau z_{\tau'} \mid \mathcal{U}_{\tau'}\right]\right| \\
&\leqslant 2\sum_{s'=1}^{s}\sum_{\tau'=(s'-1)\ell+1}^{s'\ell} \left(\overbrace{\sum_{\tau=1}^{(s_{\tau'}-1)\ell} \gamma^{\tau'-(s_{\tau'}-1)\ell}}^{=(s'-1)\ell\,\gamma^{\tau'-(s'-1)\ell}} + \overbrace{\sum_{\tau=(s_{\tau'}-1)\ell+1}^{\tau'-1} \gamma^{\tau'-\tau}}^{\leqslant \gamma/(1-\gamma)}\right) \\
&\leqslant 2\sum_{s'=1}^{s}\sum_{\tau'=(s'-1)\ell+1}^{s'\ell} \left((s'-1)\ell\,\gamma^{\tau'-(s'-1)\ell} + \frac{\gamma}{1-\gamma}\right) \leqslant \frac{2\gamma}{1-\gamma}\sum_{s'=1}^{s} s'\ell = \frac{s(s+1)\ell\gamma}{1-\gamma},
\end{aligned}
$$

from which the first inequality of the lemma follows by Equation (44). It only remains to show Equation (45).

*Step 2: Proof of Equation (45), part 1.* In this step, we show that

$$
\left|\mathbb{E}\left[z_\tau z_{\tau'} \mid \mathcal{U}_{\tau'}\right]\right| \leqslant \sum_{h\in[H]} \mathbb{P}(h_\tau = h \mid \mathcal{U}_{\tau'}) \sum_{h'\in[H]} \left|\mathbb{P}_{\{h_\tau=h\}}(h_{\tau'} = h' \mid \mathcal{U}_{\tau'}) - \mathbb{P}(h_{\tau'} = h' \mid \mathcal{U}_{\tau'})\right|.
\tag{46}
$$

In the closed-form expression for $z_\tau z_{\tau'}$, the only quantities that are not $\mathcal{U}_{\tau'}$–measurable are $\boldsymbol{\theta}_{h_\tau}^\star$ and $\boldsymbol{\theta}_{h_{\tau'}}^\star$; the other terms are $\mathcal{U}_{\tau'}$–measurable: $\boldsymbol{\varphi}(a_\tau, \boldsymbol{x}_\tau)$ and $\boldsymbol{\varphi}(a_{\tau'}, \boldsymbol{x}_{\tau'})$, as well as $\bar{\boldsymbol{b}}_\tau(h)$ and $\bar{\boldsymbol{b}}_{\tau'}(h)$. Also, by definition of $\bar{\boldsymbol{b}}_{\tau'}$,

$$
\mathbb{E}\left[\boldsymbol{\theta}_{h_{\tau'}}^\star \mid \mathcal{U}_{\tau'}\right] = \sum_{h\in[H]} \bar{\boldsymbol{b}}_{\tau'}(h)\,\boldsymbol{\theta}_h^\star.
$$

Therefore,

$$
\mathbb{E}\big[z_\tau z_{\tau'} \mid \mathcal{U}_{\tau'}\big]
$$

$$
= \mathbb{E}\left[ \boldsymbol{\varphi}(a_\tau, \boldsymbol{x}_\tau)^\top \left( \boldsymbol{\theta}^\star_{h_\tau} - \sum_{h \in [H]} \bar{\boldsymbol{b}}_\tau(h) \boldsymbol{\theta}^\star_h \right) \boldsymbol{\varphi}(a_{\tau'}, \boldsymbol{x}_{\tau'})^\top \left( \boldsymbol{\theta}^\star_{h_{\tau'}} - \sum_{h \in [H]} \bar{\boldsymbol{b}}_{\tau'}(h) \boldsymbol{\theta}^\star_h \right) \Bigg| \mathcal{U}_{\tau'} \right]
$$

$$
= \mathbb{E}\left[ \boldsymbol{\varphi}(a_\tau, \boldsymbol{x}_\tau)^\top \boldsymbol{\theta}^\star_{h_\tau} \, \boldsymbol{\varphi}(a_{\tau'}, \boldsymbol{x}_{\tau'})^\top \left( \boldsymbol{\theta}^\star_{h_{\tau'}} - \sum_{h \in [H]} \bar{\boldsymbol{b}}_{\tau'}(h) \boldsymbol{\theta}^\star_h \right) \Bigg| \mathcal{U}_{\tau'} \right].
$$

We continue the calculation by applying formulas of the form: for all $\mathcal{U}_{\tau'}$–measurable functions $F$,

$$
\mathbb{E}\Big[ F\big(\boldsymbol{\theta}^\star_{h_\tau}, \boldsymbol{\theta}^\star_{h_{\tau'}}\big) \,\Big|\, \mathcal{U}_{\tau'} \Big] = \sum_{h,h' \in [H]} \mathbb{P}\big(h_\tau = h \ \text{and} \ h_{\tau'} = h' \mid \mathcal{U}_{\tau'}\big) \, F\big(\boldsymbol{\theta}^\star_h, \boldsymbol{\theta}^\star_{h'}\big).
$$

To do so, we consider the functions

$$
G : \big(\boldsymbol{\theta}^\star_h, \boldsymbol{\theta}^\star_{h'}\big) \longmapsto \boldsymbol{\varphi}(a_\tau, \boldsymbol{x}_\tau)^\top \boldsymbol{\theta}^\star_h \, \boldsymbol{\varphi}(a_{\tau'}, \boldsymbol{x}_{\tau'})^\top \boldsymbol{\theta}^\star_{h'}
$$

$$
\text{and} \qquad \boldsymbol{\theta}^\star_h \longmapsto -\boldsymbol{\varphi}(a_\tau, \boldsymbol{x}_\tau)^\top \boldsymbol{\theta}^\star_h \, \boldsymbol{\varphi}(a_{\tau'}, \boldsymbol{x}_{\tau'})^\top \sum_{j \in [H]} \bar{\boldsymbol{b}}_{\tau'}(j) \boldsymbol{\theta}^\star_j,
$$

and get the rewriting

$$
\mathbb{E}\big[ z_\tau z_{\tau'} \mid \mathcal{U}_{\tau'} \big] = \sum_{h,h' \in [H]} \mathbb{P}\big(h_\tau = h \ \text{and} \ h_{\tau'} = h' \mid \mathcal{U}_{\tau'}\big) \, G\big(\boldsymbol{\theta}^\star_h, \boldsymbol{\theta}^\star_{h'}\big)
$$

$$
- \sum_{h \in [H]} \mathbb{P}\big(h_\tau = h \mid \mathcal{U}_{\tau'}\big) \sum_{j \in [H]} \bar{\boldsymbol{b}}_{\tau'}(j) \, G\big(\boldsymbol{\theta}^\star_h, \boldsymbol{\theta}^\star_j\big).
$$

The inequality claimed in Equation (46) follows by noting that $\big| G\big(\boldsymbol{\theta}^\star_h, \boldsymbol{\theta}^\star_{h'}\big) \big| \leqslant 1$ (by Assumption 2.1) and by applying Lemma D.1 (Bayes' formula with expectations conditional to $\sigma$–algebras).

*Step 3: Proof of Equation (45), part 2.* Given the bound of Equation (46), it suffices to show that for all $h \in [H]$,

$$
\sum_{h' \in [H]} \left| \mathbb{P}_{\{h_\tau = h\}}(h_{\tau'} = h' \mid \mathcal{U}_{\tau'}) - \mathbb{P}\big(h_{\tau'} = h' \mid \mathcal{U}_{\tau'}\big) \right|
$$

$$
\leqslant \begin{cases} 2\gamma^{\tau' - \tau} & \text{if } \tau \geqslant (s_{\tau'} - 1)\ell + 1, \text{ i.e., if } \tau \text{ belongs to the same stage as } \tau', \\ 2\gamma^{\tau' - (s_{\tau'} - 1)\ell} & \text{if } \tau \leqslant (s_{\tau'} - 1)\ell, \text{ i.e., if } \tau \text{ and } \tau' \text{ belong to different stages.} \end{cases} \tag{47}
$$

In the case when $\tau \geqslant (s_{\tau'} - 1)\ell + 1$, i.e., when $s_\tau = s_{\tau'}$, we combine a law of total probability with Lemma D.1 to get the decomposition

$$
\mathbb{P}\big(h_{\tau'} = h' \mid \mathcal{U}_{\tau'}\big) = \sum_{j \in [H]} \mathbb{P}\big(h_\tau = j \mid \mathcal{U}_{\tau'}\big) \, \mathbb{P}_{\{h_\tau = j\}}\big(h_{\tau'} = h' \mid \mathcal{U}_{\tau'}\big).
$$

Also, by the HMM conditional independence discussed around Equation (40), since $\mathcal{U}_{\tau'}$ is generated by estimates $\widehat{\boldsymbol{\theta}}_{s\ell}$ with $s \leqslant s_{\tau'} - 1 = s_{\tau-1}$, which are therefore more in the past than $h_\tau$, and by the contexts $\boldsymbol{x}_{1:\tau'}$, we have

$$
\mathbb{P}_{\{h_\tau = h\}}(h_{\tau'} = h' \mid \mathcal{U}_{\tau'}) = \mathbb{P}_{\{h_\tau = h\}}(h_{\tau'} = h' \mid \boldsymbol{x}_{\tau+1:\tau'}).
$$

Using successively these equalities (together with a triangle inequality), the quantity of interest may be upper bounded by

$$
\sum_{h' \in [H]} \left| \mathbb{P}_{\{h_\tau = h\}}(h_{\tau'} = h' \mid \mathcal{U}_{\tau'}) - \mathbb{P}\big(h_{\tau'} = h' \mid \mathcal{U}_{\tau'}\big) \right|
$$

$$
\leqslant \sum_{j \in [H]} \mathbb{P}\big(h_\tau = j \mid \mathcal{U}_{\tau'}\big) \sum_{h' \in [H]} \left| \mathbb{P}_{\{h_\tau = h\}}(h_{\tau'} = h' \mid \mathcal{U}_{\tau'}) - \mathbb{P}_{\{h_\tau = j\}}\big(h_{\tau'} = h' \mid \mathcal{U}_{\tau'}\big) \right|
$$

$$
= \sum_{j \in [H]} \mathbb{P}\big(h_\tau = j \mid \mathcal{U}_{\tau'}\big) \underbrace{\sum_{h' \in [H]} \left| \mathbb{P}_{\{h_\tau = h\}}(h_{\tau'} = h' \mid \boldsymbol{x}_{\tau+1:\tau'}) - \mathbb{P}_{\{h_\tau = j\}}\big(h_{\tau'} = h' \mid \boldsymbol{x}_{\tau+1:\tau'}\big) \right|}_{\leqslant 2\gamma^{\tau' - \tau}},
$$

where the $\leqslant 2\gamma^{\tau'-\tau}$ bound follows from Assumption 4.1. This proves Equation (47) in the first case, when $\tau$ belongs to the same stage as $\tau'$.

For the second case, when $\tau \leqslant (s_{\tau'} - 1)\ell$, i.e., $\tau$ belongs to an stage earlier than the one of $\tau'$, we adapt the argument above by also introducing $h_{(s_{\tau'}-1)\ell}$. Two combinations of a law of total probability together with Lemma D.1 and a triangle inequality entail the following bound on the quantity of interest:

$$\sum_{h' \in [H]} \left| \mathbb{P}_{\{h_\tau = h\}}(h_{\tau'} = h' \mid \mathcal{U}_{\tau'}) - \mathbb{P}(h_{\tau'} = h' \mid \mathcal{U}_{\tau'}) \right|$$

$$\leqslant \sum_{h' \in [H]} \sum_{i \in [H]} \sum_{j \in [H]} \mathbb{P}(h_{(s_{\tau'}-1)\ell} = i \mid \mathcal{U}_{\tau'}) \, \mathbb{P}(h_{(s_{\tau'}-1)\ell} = j \mid \mathcal{U}_{\tau'})$$

$$\times \left| \mathbb{P}_{\{h_\tau = h \text{ and } h_{(s_{\tau'}-1)\ell} = i\}}(h_{\tau'} = h' \mid \mathcal{U}_{\tau'}) - \mathbb{P}_{\{h_{(s_{\tau'}-1)\ell} = j\}}(h_{\tau'} = h' \mid \mathcal{U}_{\tau'}) \right|.$$

Given that $\mathcal{U}_{\tau'}$ is generated by estimates $\widehat{\boldsymbol{\theta}}_{s\ell}$ with $s \leqslant s_{\tau'} - 1$ which only depend on information till round $(s_{\tau'} - 1)\ell$, and by the contexts $\boldsymbol{x}_{1:\tau'}$, we have, by the HMM conditional independence discussed around Equation (40), that for all $j, h' \in [H]$,

$$\mathbb{P}_{\{h_{(s_{\tau'}-1)\ell} = j\}}(h_{\tau'} = h' \mid \mathcal{U}_{\tau'}) = \mathbb{P}_{\{h_{(s_{\tau'}-1)\ell} = j\}}(h_{\tau'} = h' \mid \boldsymbol{x}_{(s_{\tau'}-1)\ell+1:\tau'}).$$

Actually, since $\tau \leqslant (s_{\tau'} - 1)\ell$, we even have, with the same arguments, for all $j, h, h' \in [H]$,

$$\mathbb{P}_{\{h_\tau = h \text{ and } h_{(s_{\tau'}-1)\ell} = j\}}(h_{\tau'} = h' \mid \mathcal{U}_{\tau'}) = \mathbb{P}_{\{h_{(s_{\tau'}-1)\ell} = j\}}(h_{\tau'} = h' \mid \boldsymbol{x}_{(s_{\tau'}-1)\ell+1:\tau'}).$$

Substituting these equalities in the bound established above, and resorting to Assumption 4.1 entails

$$\sum_{h' \in [H]} \left| \mathbb{P}_{\{h_\tau = h\}}(h_{\tau'} = h' \mid \mathcal{U}_{\tau'}) - \mathbb{P}(h_{\tau'} = h' \mid \mathcal{U}_{\tau'}) \right|$$

$$\leqslant \sum_{j \in [H]} \sum_{i \in [H]} \mathbb{P}(h_{(s_{\tau'}-1)\ell} = i \mid \mathcal{U}_{\tau'}) \, \mathbb{P}(h_{(s_{\tau'}-1)\ell} = j \mid \mathcal{U}_{\tau'})$$

$$\times \underbrace{\sum_{h' \in [H]} \left| \mathbb{P}_{\{h_{(s_{\tau'}-1)\ell} = i\}}(h_{\tau'} = h' \mid \boldsymbol{x}_{(s_{\tau'}-1)\ell+1:\tau'}) - \mathbb{P}_{\{h_{(s_{\tau'}-1)\ell} = j\}}(h_{\tau'} = h' \mid \boldsymbol{x}_{(s_{\tau'}-1)\ell+1:\tau'}) \right|}_{\leqslant 2\gamma^{\tau'-(s_{\tau'}-1)\ell}}.$$

This proves Equation (47) in the second case, and concludes the proof of the lemma. $\qquad\square$

### E.1.2. BOUND ON $\left\| S'_{\text{eta},s\ell} \right\|_2$

To bound the term $\left\| S'_{\text{eta},s\ell} \right\|_2$, we mimic, and simplify, the proof conducted right before for Lemma E.2: we adapt its Step 1 (and do not need Steps 2 and 3). Actually, under the stronger noise Assumption 2.2, a LinUCB-type approach as in Appendix A could have been followed (i.e., Lemma A.3 could have been applied). We however prefer to mimic and simplify the proof of Lemma E.2.

**Lemma E.3.** *Under the Assumptions 2.1 and 2.3, for all $s \geqslant 1$,*

$$\mathbb{E}\left[ \left\| S'_{\text{eta},s\ell} \right\|_2^2 \right] = \mathbb{E}\left[ \left\| \sum_{\tau=1}^{s\ell} \eta_\tau(a_\tau) \, \widehat{\boldsymbol{b}}_\tau \otimes \boldsymbol{\varphi}(a_\tau, \boldsymbol{x}_\tau) \right\|_2^2 \right] \leqslant C_\eta s\ell.$$

*Thus, for all $\delta \in (0, 1)$, for each $s \geqslant 1$, with probability at least $1 - \delta$,*

$$\left\| S'_{\text{eta},s\ell} \right\|_2 = \left\| \sum_{\tau=1}^{s\ell} \eta_\tau(a_\tau) \, \widehat{\boldsymbol{b}}_\tau \otimes \boldsymbol{\varphi}(a_\tau, \boldsymbol{x}_\tau) \right\|_2 \leqslant \sqrt{\frac{1}{\delta} C_\eta s\ell}.$$

*Proof.* The second inequality follows from the first one via Markov's inequality. For the first inequality, we develop the squared norm and apply the bound of Equation (6):

$$\mathbb{E}\left[\left\|\sum_{\tau=1}^{s\ell}\eta_\tau(a_\tau)\,\widehat{\boldsymbol{b}}_\tau\otimes\boldsymbol{\varphi}(a_\tau,\boldsymbol{x}_\tau)\right\|_2^2\right]$$
$$\leqslant\sum_{\tau=1}^{s\ell}\mathbb{E}\big[\eta_\tau(a_\tau)^2\big]+2\sum_{1\leqslant\tau<\tau'\leqslant s\ell}\mathbb{E}\left[\eta_\tau(a_\tau)\eta_{\tau'}(a'_\tau)\big(\widehat{\boldsymbol{b}}_\tau\otimes\boldsymbol{\varphi}(a_\tau,\boldsymbol{x}_\tau)\big)^\top\big(\widehat{\boldsymbol{b}}_{\tau'}\otimes\boldsymbol{\varphi}(a_{\tau'},\boldsymbol{x}_{\tau'})\big)\right].$$

For the first component of above inequality, by Assumption 2.3 and using that the selected action $a_t$ is $\mathcal{F}_t^{\text{all}}$–measurable, we have, for all $\tau\geqslant 1$,

$$\mathbb{E}\Big[\eta_\tau(a_\tau)^2\,\Big|\,\mathcal{F}_\tau^{\text{all}}\Big]=\sum_{a\in\mathcal{A}}\mathbb{1}_{\{a_t=a\}}\,\mathbb{E}\Big[\eta_\tau(a)^2\,\Big|\,\mathcal{F}_\tau^{\text{all}}\Big]\leqslant C_\eta\,,$$

so that, by the tower rule,

$$\sum_{\tau=1}^{s\ell}\mathbb{E}\Big[\eta_\tau(a_\tau)^2\Big]=\sum_{\tau=1}^{s\ell}\mathbb{E}\Big[\mathbb{E}\Big[\eta_\tau(a_\tau)^2\,\Big|\,\mathcal{F}_\tau^{\text{all}}\Big]\Big]\leqslant s\ell C_\eta\,.$$

For the second sum, fix a pair $1\leqslant\tau<\tau'\leqslant s\ell$. We use that $a_\tau$ is measurable w.r.t. $\mathcal{F}_\tau^{\text{all}}$, and that $\mathcal{F}_{\tau'}^{\text{all}}$ is generated by $\boldsymbol{x}_\tau$, the $\eta_\tau(a)$, and other variables, to show that the random variables $\eta_\tau(a_\tau)$ and $\boldsymbol{\varphi}(a_\tau,\boldsymbol{x}_\tau)$ are all $\mathcal{F}_{\tau'}^{\text{all}}$–measurable. We also have that $\widehat{\boldsymbol{b}}_\tau$ and $\widehat{\boldsymbol{b}}_{\tau'}$ are measurable w.r.t. $\boldsymbol{x}_1,\ldots,\boldsymbol{x}_{\tau'}$, thus w.r.t. $\mathcal{F}_{\tau'}^{\text{all}}$, and by similar arguments, $a_{\tau'}$ and $\boldsymbol{\varphi}(a_{\tau'},\boldsymbol{x}_{\tau'})$ are also $\mathcal{F}_{\tau'}^{\text{all}}$–measurable. Therefore, by the tower rule and by Assumption 2.3,

$$\mathbb{E}\left[\eta_\tau(a_\tau)\eta_{\tau'}(a'_\tau)\big(\widehat{\boldsymbol{b}}_\tau\otimes\boldsymbol{\varphi}(a_\tau,\boldsymbol{x}_\tau)\big)^\top\big(\widehat{\boldsymbol{b}}_{\tau'}\otimes\boldsymbol{\varphi}(a_{\tau'},\boldsymbol{x}_{\tau'})\big)\right]$$
$$=\mathbb{E}\left[\eta_\tau(a_\tau)\big(\widehat{\boldsymbol{b}}_\tau\otimes\boldsymbol{\varphi}(a_\tau,\boldsymbol{x}_\tau)\big)^\top\big(\widehat{\boldsymbol{b}}_{\tau'}\otimes\boldsymbol{\varphi}(a_{\tau'},\boldsymbol{x}_{\tau'})\big)\sum_{a\in\mathcal{A}}\mathbb{1}_{\{a'_\tau=a\}}\underbrace{\mathbb{E}\Big[\eta_{\tau'}(a)\mid\mathcal{F}_{\tau'}^{\text{all}}\Big]}_{=0}\right]=0\,.$$

The proof is concluded by collecting all (in)equalities. $\qquad\square$

### E.1.3. BOUND ON $\big\|S'_{\text{belief},s\ell}\big\|_2$

To bound the term $\big\|S'_{\text{belief},s\ell}\big\|_2$, we also mimic, and simplify, the proof of Lemma E.2 conducted in Appendix E.1.1: we do not need its Steps 1 and 2 and we adapt its Step 3.

**Lemma E.4.** *Under Assumptions 2.1 and 4.1, for all $s\geqslant 1$, with probability 1,*

$$\big\|S'_{\text{belief},s\ell}\big\|_2=\left\|\sum_{\tau=1}^{s\ell}\sum_{h\in[H]}\boldsymbol{\varphi}(a_\tau,\boldsymbol{x}_\tau)^\top\boldsymbol{\theta}_h^\star\big(\bar{\boldsymbol{b}}_\tau(h)-\widehat{\boldsymbol{b}}_\tau(h)\big)\widehat{\boldsymbol{b}}_\tau\otimes\boldsymbol{\varphi}(a_\tau,\boldsymbol{x}_\tau)\right\|_2\leqslant\frac{2s\gamma}{1-\gamma}+\sum_{\tau=1}^{s\ell}\big\|\boldsymbol{b}_\tau-\widehat{\boldsymbol{b}}_\tau\big\|_1\,.$$

*Proof.* By the triangle inequality and the bounds indicated by Assumption 2.1 and Equation (6),

$$\left\|\sum_{\tau=1}^{s\ell}\sum_{h\in[H]}\overbrace{\boldsymbol{\varphi}(a_\tau,\boldsymbol{x}_\tau)^\top\boldsymbol{\theta}_h^\star}^{|\cdot|\leqslant 1}\big(\bar{\boldsymbol{b}}_\tau(h)-\widehat{\boldsymbol{b}}_\tau(h)\big)\overbrace{\widehat{\boldsymbol{b}}_\tau\otimes\boldsymbol{\varphi}(a_\tau,\boldsymbol{x}_\tau)}^{\|\cdot\|_2\leqslant 1}\right\|_2\leqslant\sum_{\tau=1}^{s\ell}\sum_{h\in[H]}\big|\bar{\boldsymbol{b}}_\tau(h)-\widehat{\boldsymbol{b}}_\tau(h)\big|=\sum_{\tau=1}^{s\ell}\big\|\bar{\boldsymbol{b}}_\tau-\widehat{\boldsymbol{b}}_\tau\big\|_1\,.$$

The claimed bound is obtained by another triangle inequality and by Lemma E.5 below: for all $s\geqslant 1$,

$$\sum_{\tau=1}^{s\ell}\big\|\bar{\boldsymbol{b}}_\tau-\widehat{\boldsymbol{b}}_\tau\big\|_1\leqslant\sum_{\tau=1}^{s\ell}\Big(\big\|\bar{\boldsymbol{b}}_\tau-\boldsymbol{b}_\tau\big\|_1+\big\|\boldsymbol{b}_\tau-\widehat{\boldsymbol{b}}_\tau\big\|_1\Big)\leqslant\frac{2s\gamma}{1-\gamma}+\sum_{\tau=1}^{s\ell}\big\|\boldsymbol{b}_\tau-\widehat{\boldsymbol{b}}_\tau\big\|_1\,.\qquad\square$$

**Lemma E.5.** *Under Assumption 4.1 (exponentially fast forgetting of initial condition), for each $s\geqslant 1$, with probability 1,*

$$\sum_{\tau=(s-1)\ell+1}^{s\ell}\big\|\boldsymbol{b}_\tau-\bar{\boldsymbol{b}}_\tau\big\|_1\leqslant\frac{2\gamma}{1-\gamma}\,.$$

*Proof.* The proof is a mere adaptation of Step 3 of the proof of Lemma E.2 located in Appendix E.1.1. By two applications of the law of total probability and Lemma D.1 for the second equality, by the conditional independence discussed around Equation (40) for the third equality, and by a triangle inequality together with Assumption 4.1 for the final inequality,

$$
\sum_{\tau=(s-1)\ell+1}^{s\ell} \left\| \boldsymbol{b}_\tau - \bar{\boldsymbol{b}}_\tau \right\|_1
$$

$$
= \sum_{\tau=(s-1)\ell+1}^{s\ell} \sum_{h\in[H]} \left| \mathbb{P}(h_\tau = h \mid \boldsymbol{x}_{1:\tau}) - \mathbb{P}(h_\tau = h \mid \mathcal{U}_\tau) \right|
$$

$$
= \sum_{\tau=(s-1)\ell+1}^{s\ell} \sum_{h\in[H]} \left| \sum_{i\in[H]} \mathbb{P}\big(h_{(s-1)\ell} = i \mid \boldsymbol{x}_{1:\tau}\big)\, \mathbb{P}_{\{h_{(s-1)\ell}=i\}}(h_\tau = h \mid \boldsymbol{x}_{1:\tau}) \right.
$$
$$
\left. - \sum_{j\in[H]} \mathbb{P}\big(h_{(s-1)\ell} = j \mid \mathcal{U}_\tau\big)\, \mathbb{P}_{\{h_{(s-1)\ell}=j\}}(h_\tau = h \mid \mathcal{U}_\tau) \right|
$$

$$
= \sum_{\tau=(s-1)\ell+1}^{s\ell} \sum_{h\in[H]} \left| \sum_{i\in[H]} \mathbb{P}\big(h_{(s-1)\ell} = i \mid \boldsymbol{x}_{1:\tau}\big)\, \mathbb{P}_{\{h_{(s-1)\ell}=i\}}\big(h_\tau = h \mid \boldsymbol{x}_{(s-1)\ell+1:\tau}\big) \right.
$$
$$
\left. - \sum_{j\in[H]} \mathbb{P}\big(h_{(s-1)\ell} = j \mid \mathcal{U}_\tau\big)\, \mathbb{P}_{\{h_{(s-1)\ell}=j\}}\big(h_\tau = h \mid \boldsymbol{x}_{(s-1)\ell+1:\tau}\big) \right|
$$

$$
\leqslant \sum_{\tau=(s-1)\ell+1}^{s\ell} \sum_{i\in[H]} \sum_{j\in[H]} \mathbb{P}\big(h_{(s-1)\ell} = i \mid \boldsymbol{x}_{1:\tau}\big)\, \mathbb{P}\big(h_{(s-1)\ell} = j \mid \mathcal{U}_\tau\big)
$$
$$
\times \underbrace{\sum_{h\in[H]} \left| \mathbb{P}_{\{h_{(s-1)\ell}=i\}}\big(h_\tau = h \mid \boldsymbol{x}_{(s-1)\ell+1:\tau}\big) - \mathbb{P}_{\{h_{(s-1)\ell}=j\}}\big(h_\tau = h \mid \boldsymbol{x}_{(s-1)\ell+1:\tau}\big) \right|}_{\leqslant 2\gamma^{\tau-(s-1)\ell}},
$$

from which the stated bound follows, by the formula for geometric sums. □

## E.2. Proof of Theorem 4.2

This section now proves Theorem 4.2 based on Lemma E.1: as in Appendix A—namely, the proof of Theorem A.1 based on Lemma A.2—, the final regret bound is basically given by 2 times the sum of the upper confidence bounds stated in Lemma E.1. We adapt proof of Theorem A.1 first, to take into account the staged nature of the strategy of Box A, and second, to carefully take care of unions bounds. Indeed, Lemma A.2 offered a deviation bound uniform over time rounds $t \geqslant 1$ and with a low $\sqrt{\ln(1/\delta)}$ dependency on the risk level $\delta \in (0, 1)$. On the contrary, Lemma E.1 only provides deviation bounds for each stage $s \geqslant 1$ with a $1/\sqrt{\delta_s}$ dependency on the risk level $\delta_s$ used for that stage.

The confidence bonuses $\varepsilon_{t,a}$ considered in Theorem 4.2 correspond to the upper bounds of Lemma E.1 up to the replacements of $\delta_s$ by $\delta/(4s_T)$ and of the unknown $\|\boldsymbol{b}_\tau - \widehat{\boldsymbol{b}}_\tau\|_1$ by their high-probability bounds $U_{\text{belief}}(\tau, \delta/2)$.

*Proof.* We do not substitute yet the specific values of $\lambda$ and $\ell$ considered and recall that $T$ is assumed to be known. We denote by $\varepsilon_{t,a}^\dagger$ the upper bound read in Lemma E.1 for the risk $\delta/(4s_T)$ and by substituting the stage $s_t$ to which a round $t \geqslant 1$ belongs, i.e., $\varepsilon_{t,a}^\dagger = 1 + \sqrt{d}/\lambda$ for $1 \leqslant t \leqslant \ell$, and for $\ell + 1 \leqslant t \leqslant T$,

$$
\varepsilon_{t,a}^\dagger = \left\| \boldsymbol{b}_t - \widehat{\boldsymbol{b}}_t \right\|_1 + \left\| G_{(s_t-1)\ell}^{-1}\big(\widehat{\boldsymbol{b}}_t \otimes \boldsymbol{\varphi}(a, \boldsymbol{x}_t)\big) \right\|_2 \left( \lambda\sqrt{H}\, C_{\boldsymbol{\theta}^\star} + 4\sqrt{\frac{s_T(s_t-1)(1+s_t\gamma)\ell}{\delta(1-\gamma)}} + \sqrt{\frac{4s_T}{\delta} C_\eta (s_t-1)\ell} \right.
$$
$$
\left. + \frac{2(s_t-1)\gamma}{1-\gamma} + \sum_{\tau=1}^{(s_t-1)\ell} \left\| \boldsymbol{b}_\tau - \widehat{\boldsymbol{b}}_\tau \right\|_1 \right).
$$

By Lemma E.1 and a union bound, we have the following high-probability uniform deviation bound: with probability at

least $1 - \delta/2$,

$$\forall 1 \leqslant t \leqslant T, \qquad \forall a \in \mathcal{A}, \qquad \left| \sum_{h \in [H]} \boldsymbol{b}_t(h) \boldsymbol{\varphi}(a, \boldsymbol{x}_t)^\top \boldsymbol{\theta}_h^\star - \sum_{h \in [H]} \widehat{\boldsymbol{b}}_t(h) \boldsymbol{\varphi}(a, \boldsymbol{x}_t)^\top \widehat{\boldsymbol{\theta}}_{(s_t-1)\ell, h} \right| \leqslant \varepsilon_{t,a}^\dagger .$$

From the guarantee above, we get similar guarantees as in Equations (15)–(16): with probability at least $1 - \delta/2$,

$$\max_{a \in \mathcal{A}} \sum_{h \in [H]} \boldsymbol{b}_t(h) \, \boldsymbol{\varphi}(a, \boldsymbol{x}_t)^\top \boldsymbol{\theta}_h^\star \leqslant \max_{a \in \mathcal{A}} \left\{ \varepsilon_{t,a}^\dagger + \sum_{h \in [H]} \widehat{\boldsymbol{b}}_t(h) \, \boldsymbol{\varphi}(a, \boldsymbol{x}_t)^\top \widehat{\boldsymbol{\theta}}_{(s_t-1)\ell, h} \right\}$$

and

$$\sum_{h \in [H]} \widehat{\boldsymbol{b}}_t(h) \, \boldsymbol{\varphi}(a_t, \boldsymbol{x}_t)^\top \widehat{\boldsymbol{\theta}}_{(s_t-1)\ell, h} \leqslant \varepsilon_{t,a_t}^\dagger + \sum_{h \in [H]} \boldsymbol{b}_t(h) \, \boldsymbol{\varphi}(a_t, \boldsymbol{x}_t)^\top \boldsymbol{\theta}_h^\star .$$

We now want to replace the terms $\|\boldsymbol{b}_\tau - \widehat{\boldsymbol{b}}_\tau\|_1$ by $U_{\text{belief}}(\tau, \delta/2)$ and to do so, we adapt the results developed in Equations (17) to (19), which only depend on the belief estimation subroutine and only require Assumption 3.1. More precisely, Equation (17) remains valid: with probability at least $1 - \delta/2$,

$$\forall t \in [T_0, \ T], \qquad \left\| \boldsymbol{b}_t - \widehat{\boldsymbol{b}}_t \right\|_1 \leqslant U_{\text{belief}}(t, \delta/2), \qquad \text{where} \qquad T_0 \stackrel{\text{def}}{=} \max\left\{ 2, \ \lceil T_{\mathcal{B}, \boldsymbol{M}, \nu} \left( 1 + \ln(2/\delta) \right) \rceil \right\} .$$

Thus, given that $G_{(s_t-1)\ell} \succeq \lambda \boldsymbol{I}_{dH}$ for $t \geqslant \ell + 1$ and by Equation (6), we also have, $t \geqslant \ell + 1$,

$$\left\| G_{(s_t-1)\ell}^{-1} \left( \widehat{\boldsymbol{b}}_t \otimes \boldsymbol{\varphi}(a, \boldsymbol{x}_t) \right) \right\|_2 \leqslant \frac{1}{\lambda} \left\| \widehat{\boldsymbol{b}}_t \otimes \boldsymbol{\varphi}(a, \boldsymbol{x}_t) \right\|_2 \leqslant \frac{1}{\lambda} .$$

Finally, taking into account that $\varepsilon_{t,a}^\dagger = 1 + \sqrt{d}/\lambda = \varepsilon_{t,a}$ for $1 \leqslant t \leqslant \ell$, we have (whether $T_0$ is larger or smaller than $\ell$) that with probability at least $1 - \delta/2$,

$$\forall t \in [T_0, \ T], \qquad \varepsilon_{t,a}^\dagger \leqslant \varepsilon_{t,a} + 2(T_0 - 1)/\lambda ,$$

where the $\varepsilon_{t,a}$ are the confidence bonuses considered in the statement of Theorem 4.2.

The bounds above, together with the same arguments as in Equations (19) and (20) and the definition of the Box B algorithm as picking arms $a_t$ maximizing some empirical upper confidence bounds, entail that with probability at least $1 - \delta$,

$$R_T = \sum_{t=1}^T \left( \max_{a \in \mathcal{A}} \sum_{h \in [H]} \boldsymbol{b}_t(h) \, \boldsymbol{\varphi}(a, \boldsymbol{x}_t)^\top \boldsymbol{\theta}_h^\star - \sum_{h \in [H]} \boldsymbol{b}_t(h) \, \boldsymbol{\varphi}(a_t, \boldsymbol{x}_t)^\top \boldsymbol{\theta}_h^\star \right) \leqslant 2(T_0 - 1) + \sum_{t=T_0}^T \left( 2\varepsilon_{t,a_t} + 4(T_0 - 1)/\lambda \right) .$$

We now substitute bounds on the $\varepsilon_{t,a_t}$, by replacing $f_t$ in its definition by the upper bound $f_T$ and by bounding $(s_T - 1)\ell < T$ therein, and also substitute the closed-form expression for $T_0$: with probability at least $1 - \delta$,

$$R_T \leqslant \left( 2 + 4T/\lambda \right) T_{\mathcal{B}, \boldsymbol{M}, \nu} \left( 1 + \ln(2/\delta) \right) + 2 \sum_{t=1}^T U_{\text{belief}}(t, \delta/2)$$

$$+ 2 G_{\text{sum}} \left( \lambda \sqrt{H} \, C_{\boldsymbol{\theta}^\star} + 4 \sqrt{\frac{T s_T (1 + s_T \gamma)}{\delta (1 - \gamma)}} + \sqrt{\frac{4 s_T}{\delta} T C_\eta} + \frac{2 s_T \gamma}{1 - \gamma} + \sum_{\tau=1}^T U_{\text{belief}}(\tau, \delta/2) \right), \quad (48)$$

where

$$G_{\text{sum}} = \sum_{s=1}^{s_T} \sum_{t=(s-1)\ell+1}^{\min\{s\ell, T\}} \left\| G_{(s-1)\ell}^{-1} \left( \widehat{\boldsymbol{b}}_t \otimes \boldsymbol{\varphi}(a_t, \boldsymbol{x}_t) \right) \right\|_2 .$$

We bound $G_{\text{sum}}$ by applying Lemma E.6 below to vectors $\boldsymbol{y}_\tau = \widehat{\boldsymbol{b}}_\tau \otimes \boldsymbol{\varphi}(a_\tau, \boldsymbol{x}_\tau)$, of dimension $dH$ and with Euclidean norm smaller than 1 as indicated in Equation (6), till stage $S = s_T$: we get the deterministic upper bound

$$G_{\text{sum}} \leqslant \frac{1}{\sqrt{\lambda}} \sqrt{2 \, dH s_T \ell \left( 1 + \ell/\lambda \right) \ln\left( 1 + s_T \ell/(dH\lambda) \right)} . \quad (49)$$

Equations (48) and (49) provide the closed-form regret bound claimed in the statement of Theorem 4.2.

It now suffices to show that it is of order $T^{7/8}$ up to logarithmic factors for the choices $\ell = \lceil T^{3/4} \rceil$ and $\lambda = T^{3/4}$. Actually, taking $\ell = \lceil T^a \rceil$ (thus $s_T = T/\ell$ is of order $T^{1-a}$) and $\lambda = T^b$, recalling that $T_{\mathcal{B},\boldsymbol{M},\nu}$ is a constant, we have that the regret bound of Equation (48) is of order, up to logarithmic terms,

$$T/\lambda + \sqrt{T} + \sqrt{T/\lambda\,(1+\ell/\lambda)}\Big(\lambda + T^{3/2}/\ell + \underbrace{T/\sqrt{\ell} + T/\ell}_{\leqslant T^{3/2}/\ell} + \sqrt{T}\Big),$$

i.e., of order $T^c$ where $(\,\cdot\,)_+$ denotes the non-negative part and

$$c = \max\big\{1 - b,\ 1/2,\ (1-b)/2 + (a-b)_+/2 + \max\{b,\ 3/2 - a,\ 1/2\}\big\}\,;$$

an optimization over $a \in [0,1]$ and $b \in [0,1]$ leads to $a = b = 3/4$ and $c = 7/8$, which concludes the proof. $\qquad\square$

**Elliptic potential with staged updates.** It only remains to prove the following extension of the classic elliptic potential lemma (see Lemma A.4 in Appendix A) to updates in stages.

**Lemma E.6.** *Consider vectors $\boldsymbol{y}_t \in \mathbb{R}^d$ with $\|\boldsymbol{y}_t\|_2 \leqslant 1$, a parameter $\lambda \geqslant 1$, and the Gram matrices $V_0 = \lambda \boldsymbol{I}_d$ and*

$$V_t = \lambda \boldsymbol{I}_d + \sum_{\tau=1}^{t} \boldsymbol{y}_\tau \boldsymbol{y}_\tau^\top \quad for \quad t \geqslant 1\,.$$

*For all integers $S \geqslant 1$ and $\ell \geqslant 1$, we have:*

$$\sum_{s=1}^{S} \sum_{\tau=(s-1)\ell+1}^{s\ell} \big\| V_{(s-1)\ell}^{-1} \boldsymbol{y}_\tau \big\|_2 \leqslant \frac{1}{\sqrt{\lambda}} \sum_{s=1}^{S} \sum_{\tau=(s-1)\ell+1}^{s\ell} \big\| V_{(s-1)\ell}^{-1/2} \boldsymbol{y}_\tau \big\|_2$$

$$\leqslant \frac{1}{\sqrt{\lambda}} \sqrt{2\,dS\ell\,(1+\ell/\lambda)\,\log\big(1+S\ell/(d\lambda)\big)}\,.$$

The sum in Lemma E.6 differs from the sum bounded in Lemma A.4 in two ways: first, it involves terms of the form

$$\big\| V_{t-1}^{-1} \boldsymbol{y}_t \big\|_2 \quad \text{instead of} \quad \big\| V_{t-1}^{-1/2} \boldsymbol{y}_t \big\|_2\,,$$

which leads to an additional $1/\sqrt{\lambda}$ multiplicative term in our bound, and second, the matrices $V$ are actually "frozen" within stages, which entails the other additional multiplicative factor $\sqrt{1+\ell/\lambda}$. The proof below focuses on these two modifications.

*Remark* E.7. Carpentier et al. (2020) provide some general study of the sums

$$\sum_{t=1}^{T} \big\| V_{t-1}^{-p/2} \boldsymbol{y}_t \big\|_2\,,$$

for $p \in (0,+\infty)$. For $p = 2$, as in Lemma E.6 up to staging, they obtain an upper bound of order $\sqrt{Td/\lambda}$. This corresponds, up to logarithmic factors and up to the $\sqrt{1+\ell/\lambda}$ term due to staging, to the right-most term of Lemma E.6. Therefore, the first inequality of Lemma E.6, while relying on the simple lower bound $V_{t-1} \succeq \lambda \boldsymbol{I}_d$ (see the proof above), looks sharp enough.

*Proof.* We use $V_{t-1} \succeq \lambda \boldsymbol{I}_d$ for all $t \geqslant 1$ to get, for all $s \geqslant 1$ and all $\tau \geqslant 1$,

$$\big\| V_{(s-1)\ell}^{-1} \boldsymbol{y}_\tau \big\|_2 = \big\| V_{(s-1)\ell}^{-1/2} V_{(s-1)\ell}^{-1/2} \boldsymbol{y}_\tau \big\|_2 \leqslant \frac{1}{\sqrt{\lambda}} \big\| V_{(s-1)\ell}^{-1/2} \boldsymbol{y}_\tau \big\|_2\,, \tag{50}$$

which yields the first inequality stated in the lemma. We prove below that for all $s \geqslant 1$,

$$\sum_{\tau=(s-1)\ell+1}^{s\ell} \big\| V_{(s-1)\ell}^{-1/2} \boldsymbol{y}_\tau \big\|_2 \leqslant \sqrt{2\ell\,(1+\ell/\lambda)\,\ln\frac{\det(V_{s\ell})}{\det\big(V_{(s-1)\ell}\big)}}\,. \tag{51}$$

The second inequality then follows from Equation (51) and the application of a Cauchy-Schwarz inequality:

$$\sum_{s=1}^{S} \sum_{\tau=(s-1)\ell+1}^{s\ell} \|V_{(s-1)\ell}^{-1/2}\, \boldsymbol{y}_\tau\|_2 \leqslant \sum_{s=1}^{S} \sqrt{2\ell\,(1+\ell/\lambda)\,\ln\frac{\det(V_{s\ell})}{\det(V_{(s-1)\ell})}}$$

$$\leqslant \sqrt{2S\ell\,(1+\ell/\lambda)\sum_{s=1}^{S}\ln\frac{\det(V_{s\ell})}{\det(V_{(s-1)\ell})}} = \sqrt{2S\ell\,(1+\ell/\lambda)\ln\frac{\det(V_{S\ell})}{\det(V_0)}}\,,$$

together with the fact that $\det(V_0) = \lambda^d$ and that the upper bound $d\log(\lambda + S\ell/d)$ on $\ln\big(\det(V_{S\ell})\big)$ is given by Lemma A.6. We are thus only left to prove Equation (51).

To do so, we show below that

$$\forall j \in [\ell-1], \qquad V_{(s-1)\ell}^{-1} \preceq (1+\ell/\lambda)\, V_{(s-1)\ell+j}^{-1}\,, \tag{52}$$

which, keeping in mind that $\|V^{-1/2}\boldsymbol{y}\|_2 = \sqrt{\boldsymbol{y}^\top V \boldsymbol{y}}$, directly entails that

$$\sum_{\tau=(s-1)\ell+1}^{s\ell} \|V_{(s-1)\ell}^{-1/2}\, \boldsymbol{y}_\tau\|_2 \leqslant \sqrt{1+\ell/\lambda} \sum_{\tau=(s-1)\ell+1}^{s\ell} \|V_{\tau-1}^{-1/2}\, \boldsymbol{y}_\tau\|_2\,.$$

The bound of Equation (51) is then obtained via the arguments between Equations (22) to (23) in the proof of Lemma A.4, applied within a stage, i.e., within the $\ell$ rounds from $(s-1)\ell+1$ to $s\ell$.

To prove Equation (52), we recall that

$$V_{(s-1)\ell+j} - V_{(s-1)\ell} = \sum_{\tau=(s-1)\ell+1}^{(s-1)\ell+j} \boldsymbol{y}_\tau \boldsymbol{y}_\tau^\top\,,$$

where $\boldsymbol{y}_\tau \boldsymbol{y}_\tau^\top \preceq \boldsymbol{I}_d$ since $\|\boldsymbol{y}_\tau\|_2 \leqslant 1$. We also recall that by definition, $V_{(s-1)\ell} \succeq \lambda \boldsymbol{I}_d$. Therefore,

$$V_{(s-1)\ell+j} - V_{(s-1)\ell} \preceq j\, \boldsymbol{I}_d \preceq (j/\lambda)\, V_{(s-1)\ell}\,, \qquad \text{thus} \quad V_{(s-1)\ell+j} \preceq (1+\ell/\lambda)\, V_{(s-1)\ell}\,,$$

which implies Equation (52) after inverting both sides. $\qquad\square$

## E.3. Handling Regret Defined in Terms of Actual Rewards

The end of Section 2.3 stated that the results achieved in this article go beyond the mere case of the pseudo-regret

$$R_T = \sum_{t=1}^{T} \max_{a \in \mathcal{A}} \sum_{h \in [H]} \boldsymbol{b}_t(h)\, \boldsymbol{\varphi}(a, \boldsymbol{x}_t)^\top \boldsymbol{\theta}_h^\star - \sum_{t=1}^{T} \sum_{h \in [H]} \boldsymbol{b}_t(h)\, \boldsymbol{\varphi}(a_t, \boldsymbol{x}_t)^\top \boldsymbol{\theta}_h^\star$$

and also yield a control of a regret defined in terms of actual rewards:

$$R_T^{\text{actual}} = \sum_{t=1}^{T} r_t(a_t^\star) - \sum_{t=1}^{T} r_t(a_t)\,, \qquad \text{where} \qquad a_t^\star \in \operatorname*{argmax}_{a \in \mathcal{A}} \sum_{h \in [H]} \boldsymbol{b}_t(h)\, \boldsymbol{\varphi}(a, \boldsymbol{x}_t)^\top \boldsymbol{\theta}_h^\star\,.$$

Below, we actually sketch the proof that $R_T^{\text{actual}}$ is close to $R_T$ with high probability, up to an additive term of order $T^{5/8}$. That proof sketch actually follows the (long and complex) proof provided above for Theorem 4.2.

**First step: noise terms.** We first control the noise terms. It suffices to mimic Lemma E.3 and get a scalar version thereof, where terms of the form $\widehat{\boldsymbol{b}}_\tau \otimes \boldsymbol{\varphi}(a_\tau, \boldsymbol{x}_\tau)$ are replaced by the scalar multiplier 1. This shows that the sums of the noise terms are small, and more precisely, that with probability at least $1 - \delta/3$,

$$\sum_{t=1}^{T} r_t(a_t^\star) \qquad \text{is } \sqrt{T/\delta}\text{-close to} \qquad \sum_{t=1}^{T} \boldsymbol{\varphi}(a_t^\star, \boldsymbol{x}_t)^\top \boldsymbol{\theta}_{h_t}^\star$$

$$\text{and} \qquad \sum_{t=1}^{T} r_t(a_t) \qquad \text{is } \sqrt{T/\delta}\text{-close to} \qquad \sum_{t=1}^{T} \boldsymbol{\varphi}(a_t, \boldsymbol{x}_t)^\top \boldsymbol{\theta}_{h_t}^\star\,.$$

**Second step: conditional expectations of $h_t$.** A scalar version of Lemma E.2, based on stage lengths $\ell = \lceil T^{3/4} \rceil$, ensures that with probability at least $1 - \delta/3$,

$$\sum_{t=1}^{T} \boldsymbol{\varphi}(a_t, \boldsymbol{x}_t)^\top \boldsymbol{\theta}_{h_t}^\star \qquad \text{is } T^{5/8}/\sqrt{\delta}\text{--close to} \qquad \sum_{t=1}^{T} \boldsymbol{\varphi}(a_t, \boldsymbol{x}_t)^\top \sum_{h \in [H]} \bar{\boldsymbol{b}}_t(h) \, \boldsymbol{\theta}_h^\star .$$

The same proof technique as in Lemma E.2, but without stages (and in a scalar fashion), similarly entails that with probability at least $1 - \delta/3$,

$$\sum_{t=1}^{T} \boldsymbol{\varphi}(a_t^\star, \boldsymbol{x}_t)^\top \boldsymbol{\theta}_{h_t}^\star \qquad \text{is } \sqrt{T/\delta}\text{--close to} \qquad \sum_{t=1}^{T} \boldsymbol{\varphi}(a_t^\star, \boldsymbol{x}_t)^\top \sum_{h \in [H]} \boldsymbol{b}_t(h) \, \boldsymbol{\theta}_h^\star ;$$

we use here that $a_t^\star$ is $\sigma(\boldsymbol{x}_{1:t})$–measurable.

**Third step: relating posterior probabilities.** Finally, the difference between

$$\sum_{t=1}^{T} \boldsymbol{\varphi}(a_t, \boldsymbol{x}_t)^\top \sum_{h \in [H]} \bar{\boldsymbol{b}}_t(h) \, \boldsymbol{\theta}_h^\star \qquad \text{and} \qquad \sum_{t=1}^{T} \boldsymbol{\varphi}(a_t, \boldsymbol{x}_t)^\top \sum_{h \in [H]} \boldsymbol{b}_t(h) \, \boldsymbol{\theta}_h^\star$$

is bounded by

$$\sum_{t=1}^{T} \left\| \boldsymbol{b}_t - \bar{\boldsymbol{b}}_t \right\|_1 ,$$

which is exactly the quantity that Lemma E.5 controls: it is of order $T^{1/4}$ given the value $\ell = \lceil T^{3/4} \rceil$ picked.

**Conclusion.** Collecting all bounds, we see that $R_T^{\text{actual}}$ is $T^{5/8}\sqrt{\delta}$–close to $R_T$, with probability at least $1 - \delta$.

# F. Numerical Simulations

The focus of this article is primarily theoretical: the simulations are intended mainly to illustrate the practical behavior of the proposed algorithms, specifically their convergence and their performance relative to relevant baselines. Accordingly, the purpose of this appendix is threefold.

**Goal 1: Illustrate the impact of taking into account the latent dynamics.** First, we compare the belief-based LinUCB strategies (Box A, and its special case Box B) with a baseline given by plain LinUCB (introduced by Abbasi-Yadkori et al., 2011 and restated in Box D of this appendix). This baseline ignores the latent state dynamics and treats the observed contexts as if they were directly sampled from a stochastic environment and the rewards as if they depended only on the observed contexts and actions. Alternative baselines we could most immediately think of are based on latent-state bandit models but do not incorporate contextual information into the reward model: this includes, for example, the works by Zhou et al. (2021) and Azizzadenesheli et al. (2016), as well as Nelson et al., 2022 in their original form.

**Goal 2: Illustrate the impact of the hyper-parameters $\ell$ and confidence bonuses $\varepsilon_{t,a}$.** The belief-based strategies considered in this article (stated in Box A, with Box B being a special case for stages of length $\ell = 1$) depend on two hyper-parameters: the stage lengths $\ell \geqslant 1$, and the form of the confidence bonuses. We considered two forms in this article: one in Section 4.2 to deal with rewards stemming from the most complex reward model (1), where rewards depend directly on the latent states, and one in Appendix A suited to the simpler reward model (2), where rewards are functions of the beliefs. While the theory developed in this article did not consider the case of $\ell \geqslant 2$ and the confidence bonuses of Appendix A, in the experiments, we go beyond these original designs and provide a more comprehensive study of performance according to these hyper-parameters. The goal is to disentangle the effect of the form of the confidence bonuses $\varepsilon_{t,a}$ from the effect of the update schedule $\ell$. Since both the staging scheme and the larger confidence bonuses of Section 4.2 are introduced in the theory to cope with the direct dependence of rewards on the latent states, these simulations help determine whether they are merely technical tools for the analysis or whether they also provide a practical advantage.

**Goal 3: Provide a realistic application with a larger-scale HMM.** Third, these simulations also illustrate that the proposed algorithms can be run in a realistic application setting. The simulations are carried out on a light computing setup (8 cores, 16 threads, 32 GB RAM) and without using a GPU. Even so, the hidden state dynamics in our simulations are already nontrivial: the underlying HMM (described in detail below) features 2 hidden states and emission distributions over 20 different values. By comparison, Nelson et al. (2022) consider a smaller simulated setting with 2 hidden states and 4 emission values. Similarly, small emission spaces are also used in the simulations of Zhou et al. (2021) and Azizzadenesheli et al. (2016).

## F.1. Data Preparation; Variables; Latent States and Contexts

We use partially simulated but realistic data. A brief summary of the hypothetical simulation background in banking industry is the following: a bank aims to optimize its marketing strategy for a credit product. Each potential client is described by a context (a client profile), and the bank chooses one of three marketing actions: *Call*, *Email*, or *No action*. Rewards are generated by a latent-state-dependent linear function with additive noise. We assume that the environment switches between two unobserved economic latent states: *inflation* and *recession*. The latent state affects both the distribution of client profiles and the reward function. For instance, during a recession, clients tend to have lower revenue and may respond differently to marketing actions than during inflation. The objective is to maximize cumulative rewards by adapting the actions over time.

**Data set.** Our simulations are based on the "Default of Credit Card Clients" dataset from UCI Machine Learning Repository (Yeh, 2009), originally provided by Yeh & Lien (2009). The dataset is distributed under the Creative Commons Attribution 4.0 International (CC BY 4.0) license and was designed to benchmark algorithms for predicting credit card default probabilities. It contains socio-demographic variables, debt levels, payment histories, and a binary target indicating whether a client defaulted in the next month.

For the purposes of these simulations, each row is interpreted as a potential marketing opportunity. We discard some variables and create additional features as described below. Our reprocessing is close to the simulation setup of Li & Stoltz (2022, Appendix F), who study a "market share expansion for loans" application based on the same dataset.

**Variables kept.** We keep the following variables, with mild preprocessing:.

- *Age*—client age in years at the time of the campaign, discretized into five levels using cutoffs 27, 31, 37, and 43 (level 1 denoting the younger age category and level 5 the oldest);
- *Education*—client's education level, regrouped into four levels (others, high school, university, and graduate school);
- *Marital status*—client's marital status with three levels (single, married, and other).

**Variables created.**   We construct two additional variables:

- *Revenue*—a proxy of client's revenue derived from the current debt level in the original dataset by multiplying the latter by 0.2; the value obtained is further discretized into four levels with cutoffs 10K, 36K, and 54K (level 1 denoting the lowest revenue category and level 4 being the highest).

- *Risk score*—following Li & Stoltz (2022, Appendix F), we fit a probability of default using a XGBoost model (Chen & Guestrin, 2016) on the original dataset to estimate the probability that a client defaults on the loan in the following month. Full details of the XGBoost hyper-parameters are provided in Li & Stoltz (2022, Appendix F) and the number of trees selected by cross validation equals now 1,111 (this number is slightly different than in the reference due to differences in the versions of Python and of some packages). We then discretize the predicted probabilities of default into five equal quantile bins to obtain a finite risk score, with level 1 denoting the lowest risk and level 5 the highest. (It is unnecessary here to recalibrate the predicted probabilities by dividing them by 4 and cap the resulting value to 20%, as in the reference, since we are only interested in the resulting quantile-based discretization.)

**Latent states.**   We consider two latent states, referred to as *Inflation* ($h = 1$) and *Recession* ($h = 2$). As in the theoretical model, the latent state affects both the context distribution and the reward function.

The contexts $\boldsymbol{x}_t$ are given by the variables described above, i.e., are 5–uples of the form (Age, Education, Marital status, Revenue, Risk score). We assume that the latent state only influences the last two components $\boldsymbol{x}'_t$, formed by Revenue and Risk score, and that the first three components, denoted by $\boldsymbol{w}_t$, are independent of the latent state $h_t$ conditional to $\boldsymbol{x}'_t$. We also assume that the learner is aware of this fact, which entails (by an application of the Bayes' rule) that

$$\forall t \geqslant 1, \ \forall h \in [2], \qquad \mathbb{P}(h_t = h \mid \boldsymbol{x}_{1:t}) = \mathbb{P}(h_t = h \mid \boldsymbol{x}'_{1:t}),$$

so that belief estimates should be computed based solely on the subcontexts $\boldsymbol{x}'_{1:t}$. The estimation procedures and regret guarantees remain unchanged up to this adaptation. Assuming this piece of knowledge is reasonable in practice, as the bank usually has prior domain knowledge about which covariates are likely to reflect the latent economic regime.

However, rewards may and will depend on the entire context $\boldsymbol{x}_t = (\boldsymbol{w}_t, \boldsymbol{x}'_t)$. The modeling above is handy for computational reasons: subcontexts $\boldsymbol{x}'_t$ only take $4 \times 5 = 20$ different values while full contexts $\boldsymbol{x}_t$ take $5 \times 4 \times 3 \times 20 = 1{,}200$ different values; the belief estimation procedure would be computationally expensive to implement if all five components of the contexts truly depended on the latent states.

We now indicate how we pick the first three components $\boldsymbol{w}_t$ independently of the latent states $h_t$ conditional to the last two components $\boldsymbol{x}'_t$.

**Context generation.**   The generation of the last two components $\boldsymbol{x}'_t$ through a HMM is described in Appendix F.2 below. We rather explain here how we generated $\boldsymbol{w}_t$ based on $\boldsymbol{x}'_t$: we do so only based on the value of $\boldsymbol{x}'_t$, which justifies the conditional independence to the latent state $h_t$.

More precisely, the dataset constructed above contains about 30,000 statistical units and therefore preserves realistic empirical dependence among the original covariates. At round $t \geqslant 1$, given the reduced context $\boldsymbol{x}'_t$ generated, we identify the (thousands of) statistical units in the data set with the same values of Revenue and Risk score and sample (with replacement) one such unit: its values for Age, Education and Marital status form the first three components $\boldsymbol{w}_t$ of the full context $\boldsymbol{x}_t = (\boldsymbol{w}_t, \boldsymbol{x}'_t)$.

### F.2. Parameters for the HMM and the Reward Functions

We picked somewhat arbitrary HMM and reward parameters: the goal is not to claim that these parameters are calibrated to a specific market, but rather to create a simulated environment in which the latent state has a visible effect on both contexts and rewards.

*Table 1.* Emission probabilities $\nu_h$ by latent state $h \in [2]$: for readability, the 20 values are presented through $5 \times 4$ tables, with rows corresponding to Risk score levels and columns, to Revenue levels.

| *Inflation* ($h = 1$) | | | | | *Recession* ($h = 2$) | | | |
| --- | --- | --- | --- | --- | --- | --- | --- | --- |
| Risk \ Revenue | 1 | 2 | 3 | 4 | Risk \ Revenue | 1 | 2 | 3 | 4 |
| 1 | 0.0407 | 0.1242 | 0.0956 | 0.0665 | 1 | 0.0093 | 0.0373 | 0.0202 | 0.0089 |
| 2 | 0.0048 | 0.0760 | 0.0628 | 0.0661 | 2 | 0.0420 | 0.0320 | 0.0098 | 0.0361 |
| 3 | 0.0697 | 0.0109 | 0.0664 | 0.1634 | 3 | 0.0855 | 0.0341 | 0.0265 | 0.0070 |
| 4 | 0.0010 | 0.0058 | 0.0025 | 0.0967 | 4 | 0.1101 | 0.0636 | 0.0913 | 0.0228 |
| 5 | 0.0050 | 0.0310 | 0.0107 | 0.0002 | 5 | 0.0986 | 0.0480 | 0.1423 | 0.0746 |

**HMM Parameters.** We recall that there are two values for the latent state, *Inflation* ($h = 1$) and *Recession* ($h = 2$). We consider the transition matrix

$$M = \begin{bmatrix} \mathbb{P}(\text{Inflation} \to \text{Inflation}) & \mathbb{P}(\text{Inflation} \to \text{Recession}) \\ \mathbb{P}(\text{Recession} \to \text{Inflation}) & \mathbb{P}(\text{Recession} \to \text{Recession}) \end{bmatrix} = \begin{bmatrix} 0.85 & 0.15 \\ 0.2 & 0.8 \end{bmatrix}$$

with unique stationary distribution $\boldsymbol{\pi} = (4/7,\ 3/7)$, as well as the emission distributions $\nu_1$, $\nu_2$ reported in Table 1.

Given the expression for $M$, the latent state is persistent: remaining in the same latent state is substantially more likely than switching to the other state. The emission probabilities reflect the intended economic interpretation: the inflation state is associated more strongly with lower risk and higher revenue profiles, whereas the recession state places more probability mass on higher risk and lower revenue profiles.

We draw the initial latent state $h_1 \sim \boldsymbol{\pi}$, and then, successively, for all $t \geqslant 1$, draw the reduced context $\boldsymbol{x}'_t$ from the emission distribution $\nu_{h_t}$ and draw the next latent state $h_{t+1}$ according to the distribution read in the row $h_t$ of $M$.

**Reward model.** We recall that the action space $\mathcal{A}$ contains three actions: $\mathcal{A} = \{\text{No action, Call, Email}\}$. At round $t \geqslant 1$, given a latent state $h_t$, a context $\boldsymbol{x}_t$, and an action $a$, the realized reward is generated according to Equation (1), in the specific form

$$r_t(a) = \underbrace{\boldsymbol{\varphi}(a, \boldsymbol{x}_t)^\top \boldsymbol{\theta}^\star_{h_t}}_{=\mu_{h_t}(a, \boldsymbol{x}_t)} + \eta_t(a), \qquad \text{where} \qquad \eta_t(a) \sim \mathcal{N}(0, 0.2).$$

The linear expected part $\mu_{h_t}(a, \boldsymbol{x}_t)$ of the reward is defined through coefficients listed in Tables 2 and 3, as follows: for all $h \in [2]$, action $a \in \mathcal{A}$, and context $\boldsymbol{x}$,

$$\begin{aligned}
\mu_h(a, \boldsymbol{x}) = {}& \beta_0^h + \beta_{\text{age}}^h(\text{Age}) + \beta_{\text{edu}}^h(\text{Education}) + \beta_{\text{mar}}^h(\text{Marital Status}) + \beta_{\text{rev}}^h(\text{Revenue}) + \beta_{\text{risk}}^h(\text{Risk score}) \\
& + \mathbb{1}_{\{a=\text{Call}\}} \beta_{\text{call}}^h + \mathbb{1}_{\{a=\text{Email}\}} \beta_{\text{email}}^h \\
& + \mathbb{1}_{\{a=\text{Call}\}} \beta_{\text{call}\times\text{risk}}^h(\text{Risk score}) + \mathbb{1}_{\{a=\text{Email}\}} \beta_{\text{email}\times\text{risk}}^h(\text{Risk score}) \\
& + \mathbb{1}_{\{a=\text{Call}\}} \beta_{\text{call}\times\text{rev}}^h(\text{Revenue}) + \mathbb{1}_{\{a=\text{Email}\}} \beta_{\text{email}\times\text{rev}}^h(\text{Revenue}).
\end{aligned}$$

For a context $\boldsymbol{x} = (i_1, i_2, i_3, i_4, i_5)$, coefficients marked like $\beta_{\text{rev}}^h(\text{Revenue})$ and $\beta_{\text{risk}}^h(\text{Risk Score})$ refer to $\beta_{\text{rev}}^h(i_4)$ and $\beta_{\text{risk}}^h(i_5)$, respectively.

The coefficients are listed in Table 2 and Table 3. In words, the tables specify, for each latent state, an intercept, additive effects for the context variables, additive effects for the active marketing actions, and action-specific additive interactions with Risk score and Revenue. For the passive action (No action), only the intercept and additive context effects remain.

### F.3. Algorithms and Hyper-parameters Thereof

We consider $T = 50,000$ and use the first 250 rounds as a warm start for all algorithms considered, during which actions are drawn uniformly at random. We compare the belief-based LinUCB strategies (Box A, and its special case Box B) with a baseline given by plain LinUCB (introduced by Abbasi-Yadkori et al., 2011 and restated next in Box D).

#### F.3.1. STAGED LINUCB STRATEGIES ON ESTIMATED BELIEFS

These strategies correspond to Box A and Box B.

*Table 2.* Coefficients for the linear reward model under inflation $h = 1$.

| Intercept | | | 0.05 | | |
|---|---|---|---|---|---|
| **Context variables** | | | Coefficients for each level | | |
| | Level 1 | Level 2 | Level 3 | Level 4 | Level 5 |
| *Risk score* | 0.067 | 0.05 | 0.033 | 0.017 | 0 |
| *Revenue* | 0 | 0.017 | 0.025 | 0.033 | |
| *Age* | 0 | 0.008 | 0.017 | 0.008 | 0 |
| *Education* | 0 | 0 | 0.017 | 0.033 | |
| *Marital status* | 0 | 0 | 0.033 | | |
| Action Variables | | | Single coefficient | | |
| *Call* | | | $-0.2$ | | |
| *Email* | | | 0.025 | | |
| Action $\times$ Risk Score | | | Coefficients for each level | | |
| | Risk Score: 1 | Risk Score: 2 | Risk Score: 3 | Risk Score: 4 | Risk Score: 5 |
| *Call* | $-0.2$ | $-0.1$ | 0 | 0.3 | 0.4 |
| *Email* | 0.05 | 0.025 | 0 | $-0.2$ | $-0.25$ |
| Action $\times$ Revenue | | | Coefficients for each level | | |
| | Revenue: 1 | Revenue: 2 | Revenue: 3 | Revenue: 4 | |
| *Call* | 0.2 | 0 | 0 | $-0.2$ | |
| *Email* | $-0.1$ | 0 | 0 | 0.025 | |

*Table 3.* Coefficients for the linear reward model under recession $h = 2$.

| Intercept | | | 0.017 | | |
|---|---|---|---|---|---|
| **Context variables** | | | Coefficients for each level | | |
| | Level 1 | Level 2 | Level 3 | Level 4 | Level 5 |
| *Risk score* | 0.067 | 0.05 | 0.033 | 0.017 | 0 |
| *Revenue* | 0 | 0.017 | 0.025 | 0.033 | |
| *Age* | 0 | 0.008 | 0.017 | 0.008 | 0 |
| *Education* | 0 | 0 | 0.017 | 0.033 | |
| *Marital status* | 0 | 0 | 0.033 | | |
| Action Variables | | | Single coefficient | | |
| *Call* | | | 0.15 | | |
| *Email* | | | $-0.1$ | | |
| Action $\times$ Risk Score | | | Coefficients for each level | | |
| | Risk Score: 1 | Risk Score: 2 | Risk Score: 3 | Risk Score: 4 | Risk Score: 5 |
| *Call* | 0.25 | 0.2 | 0 | $-0.1$ | $-0.3$ |
| *Email* | 0.15 | 0.1 | 0 | 0.1 | 0.05 |
| Action $\times$ Revenue | | | Coefficients for each level | | |
| | Revenue: 1 | Revenue: 2 | Revenue: 3 | Revenue: 4 | |
| *Call* | $-0.2$ | 0 | 0 | 0.1 | |
| *Email* | 0.1 | 0 | 0 | 0.05 | |

Three hyper-parameters need to be set: the stage lengths $\ell \geqslant 1$ (with $\ell = 1$ corresponding to per-round updates as in Box B), the regularization parameter $\lambda > 0$, and the form of the confidence bonuses $\varepsilon_{t,a}$. We take as belief-estimation subroutine $\mathcal{B}$ the spectral method described in Box C of Section C.4 to estimate HMM parameters, together with the Bayes' updates rules of Equations (25) to (27).

**Stage length $\ell \geqslant 1$.**   The theory suggests a reference stage length of order $T^{3/4}$. Since $T = 50,000$ is relatively small from the viewpoint of the asymptotic analysis, we rather report results for $\ell \in \{1, 15, 37, 224\}$, which are integer-rounded values of $T^0, T^{1/4}, T^{1/3}$, and $T^{1/2}$.

**Regularization parameter $\lambda > 0$.**   We tune $\lambda$ over the logarithmic grid $\{10^k : k \in \{-9, -3, -2, -1, 0, 1, 2\}\}$. The range of this base–10 grid is chosen to cover several orders of magnitude and to ensure that the best-performing value of $\lambda$ is not attained at one endpoint of the grid for the strategies implemented. Importantly, the marginal effect of further decreasing $\lambda$ below $10^{-3}$ becomes negligible. Thus, we include $\lambda = 10^{-9}$ as a representative value at this order of magnitude to approximate the case of a very weak regularization while avoiding the numerical instability suffered when inverting the Gram matrix for even smaller values of $\lambda$.

**Confidence bonuses $\varepsilon_{t,a}$: two forms.**   We considered two forms of confidence bonuses in this article: one in Section 4.2 to deal with rewards stemming from the most complex reward model (1), where rewards depend directly on the latent states, and one in Appendix A suited to the simpler reward model (2). We will respectively refer to these two forms as the "complex form" and the "simplified form".

**Confidence bonuses $\varepsilon_{t,a}$: "complex form".**   We recall the expression stated in Theorem 4.2:

$$\varepsilon_{t,a} = U_{\text{belief}}(t, \delta/2) + f_t \left\| G^{-1}_{(s_t-1)\ell}\left(\widehat{\boldsymbol{b}}_t \otimes \boldsymbol{\varphi}(a, \boldsymbol{x}_t)\right) \right\|_2, \qquad \text{where}$$

$$f_t = \lambda\sqrt{H}\, C_{\boldsymbol{\theta}^\star} + 4\sqrt{\frac{s_T(s_t-1)(1+s_t\gamma)\ell}{\delta(1-\gamma)}} + \sqrt{\frac{4s_T}{\delta}}C_\eta(s_t-1)\ell + \frac{2(s_t-1)\gamma}{1-\gamma} + \sum_{\tau=1}^{(s_t-1)\ell} U_{\text{belief}}(\tau, \delta/2).$$

We actually omit the $\sqrt{s_T/\delta}$ terms coming from the union bounds mentioned at the beginning of Appendix E.2, so that the dominant contribution in the formula above for $\varepsilon_{t,a}$ is

$$4\sqrt{\frac{(s_t-1)(1+s_t\gamma)\ell}{\delta(1-\gamma)}} \left\| G^{-1}_{(s_t-1)\ell}\left(\widehat{\boldsymbol{b}}_t \otimes \boldsymbol{\varphi}(a, \boldsymbol{x}_t)\right) \right\|_2.$$

In the simulation, we therefore use confidence bonuses $\varepsilon_{t,a}^{\text{cplx}}$ of the "complex form"

$$\varepsilon_{t,a}^{\text{cplx}} = C\, s_t\sqrt{\ell} \left\| G^{-1}_{(s_t-1)\ell}\left(\widehat{\boldsymbol{b}}_t \otimes \boldsymbol{\varphi}(a, \boldsymbol{x}_t)\right) \right\|_2, \tag{53}$$

where the multiplicative exploration constant $C$ controls the exploration level and is tuned over the logarithmic grid $\{5 \times 10^k : k \in \{-6, -5, -4, -3, -2, -1\}\}$. This grid is chosen to cover a wide range of exploration strengths and to ensure that the best-performing value of $C$ lies in the interior of the grid for all the strategies implemented.

**Confidence bonuses $\varepsilon_{t,a}$: "simplified form".**   These confidence bonuses are derived, for the value $\ell = 1$ (and only for this value), from Lemma A.2 together with some crude boundings. Their dominant term is of the original form given by the left-hand side below (looking at the proof), even though we rather stated an upper bound thereof in Lemma A.2 (based on $G_{t-1} \succeq \lambda \boldsymbol{I}_{dH}$), given by the right-hand side below:

$$\left\| G^{-1}_{t-1}\left(\widehat{\boldsymbol{b}}_t \otimes \boldsymbol{\varphi}(a, \boldsymbol{x}_t)\right) \right\|_2 \sum_{\tau=1}^{t-1} \left\| \boldsymbol{b}_\tau - \widehat{\boldsymbol{b}}_\tau \right\|_1 \leqslant \left\| \widehat{\boldsymbol{b}}_t \otimes \boldsymbol{\varphi}(a, \boldsymbol{x}_t) \right\|_{G^{-1}_{t-1}} \frac{1}{\sqrt{\lambda}} \sum_{\tau=1}^{t-1} \left\| \boldsymbol{b}_\tau - \widehat{\boldsymbol{b}}_\tau \right\|_1.$$

The right-hand side provides a simpler and more readable expression to derive the confidence bonuses $\varepsilon_{t,a}$ in Theorem A.1 but for a fairer comparison with the "complex form" of confidence bonuses (53), and due to their similar expressions, we

---

**BOX D: PLAIN LINUCB STRATEGY (ABBASI-YADKORI ET AL., 2011)**

**Known parameters:** finite action set $\mathcal{A}$; context set $\mathcal{X}$; transfer function $\boldsymbol{\varphi} : \mathcal{A} \times \mathcal{X} \to \mathbb{R}^d$;

**Unknown parameters:** HMM parameters, given by a transition matrix $\boldsymbol{M} = (M_{h,h'})_{(h,h') \in [H]}$ and emission distributions $(\nu_h)_{h \in [H]}$ over $\mathcal{X}$; reward parameters $\boldsymbol{\theta}_h^\star \in \mathbb{R}^d$, for $h \in [H]$

**Inputs:** regularization parameter $\lambda > 0$; closed-form expression for the confidence bonuses $\varepsilon_{t,a}$

**Initialization:** the learner sets $\widehat{\boldsymbol{\theta}}_0 = (1/\lambda)\,\mathbf{1} \in \mathbb{R}^d$

**For** rounds $t \geqslant 1$, **the learner:**

1. Observes the context $\boldsymbol{x}_t$, drawn independently by the environment from $\nu_{h_t}$;

2. Computes the estimated mean rewards $\quad \widehat{r}_t(a) = \boldsymbol{\varphi}(a, \boldsymbol{x}_t)^\top \widehat{\boldsymbol{\theta}}_{t-1} \quad$ for all $a \in \mathcal{A}$;

3. Picks an action $a_t \in \underset{a \in \mathcal{A}}{\operatorname{argmax}} \big\{ \widehat{r}_t(a) + \varepsilon_{t,a} \big\}$;

4. Obtains and observes the reward $\quad r'_t(a_t) = \sum_{h \in [H]} \boldsymbol{b}_t(h)\, \boldsymbol{\varphi}(a_t, \boldsymbol{x}_t)^\top \boldsymbol{\theta}_h^\star + \eta_t(a_t)$;

5. Computes $\quad \widehat{\boldsymbol{\theta}}_t = V_t^{-1} \sum_{\tau=1}^{t} \boldsymbol{\varphi}(a_\tau, \boldsymbol{x}_\tau) r_\tau(a_\tau) \quad$ where $\quad V_t = \sum_{\tau=1}^{t} \boldsymbol{\varphi}(a_\tau, \boldsymbol{x}_\tau) \boldsymbol{\varphi}(a_\tau, \boldsymbol{x}_\tau)^\top + \lambda \boldsymbol{I}_d$.

**end**

---

prefer resorting to the tighter left-hand side above. Replacing the cumulative belief-estimation error by its order $\sqrt{t}$, we thus consider, in the case $\ell = 1$ only, confidence bonuses proportional to

$$\sqrt{t}\, \left\| G_{t-1}^{-1} \Big( \widehat{\boldsymbol{b}}_t \otimes \boldsymbol{\varphi}(a, \boldsymbol{x}_t) \Big) \right\|_2.$$

The analysis in Appendix A was only performed for the case $\ell = 1$ of no stages, but we extend it in the simulations to staged updates, by considering

$$\varepsilon_{t,a}^{\mathrm{simpl}} = C\,\sqrt{s_t \ell}\, \left\| G_{(s_t-1)\ell}^{-1} \Big( \widehat{\boldsymbol{b}}_t \otimes \boldsymbol{\varphi}(a, \boldsymbol{x}_t) \Big) \right\|_2$$

as the confidence bonuses of the "simplified form". The multiplicative exploration constant $C$ is tuned over the same grid as above.

### F.3.2. PLAIN LINUCB STRATEGY BY ABBASI-YADKORI ET AL. (2011)

As a baseline, we consider the LinUCB strategy by Abbasi-Yadkori et al. (2011) in its standard form, see Box D. This baseline ignores the latent-state dynamics altogether and therefore does not exploit either the HMM structure or the belief estimates. In particular, the rewards in the simulation are still generated by the true latent-state-dependent model, but the plain version of LinUCB treats them as if they arose from a standard linear contextual bandit model based only on the observed context and action. Thus, the comparison of the LinUCB strategies exploiting estimated beliefs to this plain version of LinUCB indicates whether latent-state-aware models bring a practical benefit.

We recall in Box D the plain LinUCB strategy of Abbasi-Yadkori et al. (2011), as slightly adapted by Li & Stoltz (2022, Appendix E) to take care of the existence of a transfer function $\varphi$ taking into account the action; in particular, the confidence bonuses used therein are of the form

$$C \ln(t)\, \big\| \boldsymbol{\varphi}(a, \boldsymbol{x}_t) \big\|_{V_{t-1}^{-1}},$$

where the matrices $V_{t-1}$ are defined in Box D. Plain LinUCB relies on per-round updates. We use the same grids of exploration constants $C$ and regularization parameters $\lambda$ as for the staged LinUCB on estimated beliefs; see Appendix F.3.1.

### F.4. Performance Reported: Empirical Averages of Pseudo-Regrets

**Disclaimer.** We ran $N = 100$ independent simulation, using random seeds $1951, \ldots, 2050$. Because $N = 100$ is relatively small, the results below should be interpreted as illustrative only. This choice also reflects our computational

budget: the simulations were run on a modest CPU-only setup, and the goal here is to visualize the practical behavior of the algorithms rather than to provide an extensive empirical benchmark.

**Additional indexations by runs.** For run $i \in [N]$, let $h_t^{(i)}$, $\boldsymbol{x}_t^{(i)}$, $\boldsymbol{b}_t^{(i)}$, and $a_t^{(i)}$ denote, respectively, the realized latent state, the realized context, the true belief, and the action selected by the policy at round $t$. In particular,

$$\forall h \in [H], \qquad \boldsymbol{b}_t^{(i)}(h) = \mathbb{P}\big(h_t^{(i)} = h \,\big|\, \boldsymbol{x}_{1:t}^{(i)}\big).$$

In the simulation, these true beliefs are computed via the Bayes' update rule of Equations (25) to (27), performed with the true HMM parameters and the realized contexts $\boldsymbol{x}_{1:t}^{(i)}$. They are used only for the evaluation of the strategies, not by the strategies themselves.

**Empirical average of pseudo-regrets.** We report pseudo-regrets rather than cumulative rewards for three reasons. First, as shown next in Appendix F.5, the performance of the variants of the staged LinUCB strategy on estimated beliefs is often close to each other, while pseudo-regret makes their difference easier to visualize. Second, since the results are averaged over only $N = 100$ runs, realized rewards would include additional Gaussian noise. Third, the consideration of cumulative pseudo-regrets directly indicates whether an algorithm exhibits sublinear or approximately linear regret over time.

That being said, we thus report pseudo-regrets. The pseudo-regret of run $i$ up to round $t$ is defined, with the notation above and given the definition of Equation (5), by

$$R_T^{(i)} = \sum_{\tau=1}^{t} \max_{a \in \mathcal{A}} \sum_{h \in [H]} \boldsymbol{b}_\tau^{(i)}(h)\, \boldsymbol{\varphi}\big(a, \boldsymbol{x}_t^{(i)}\big)^\top \boldsymbol{\theta}_h^\star - \sum_{\tau=1}^{t} \sum_{h \in [H]} \boldsymbol{b}_t^{(i)}(h)\, \boldsymbol{\varphi}\big(a_t^{(i)}, \boldsymbol{x}_t^{(i)}\big)^\top \boldsymbol{\theta}_h^\star.$$

In the simulations, we report the empirical averages

$$\bar{R}_t = \frac{1}{N} \sum_{i=1}^{N} R_t^{(i)}$$

over time, together with bands equal to $\pm 2$ times the standard errors of the series $\big(R_t^{(i)}\big)_{i \in [N]}$.

### F.5. Outcomes of Simulations

**Overview of the performance by strategies.** Figure 2 compares the pseudo-regret of the baseline strategy, *Plain LinUCB* to the one of the strategies introduced in this article: the staged LinUCB strategy on estimated beliefs with complex-form confidence bonuses, abbreviated as *LinUCB-Belief-Complex* on the pictures and tables, and of the staged LinUCB strategy on estimated beliefs with simplified-form confidence bonuses, abbreviated as *LinUCB-Belief-Simplified*. We do so for $\ell \in \{1, 15, 37, 224\}$, using for each value of $\ell$ the best exploration constants $C$ and regularization parameters $\lambda$ selected in hindsight from the grids considered.

The first observation is that the baseline *Plain LinUCB*, which does not leverage the latent-state structure, exhibits approximately linear pseudo-regret, even with the best $C$ and $\lambda$ in hindsight. By contrast, both *LinUCB-Belief-Complex* and *LinUCB-Belief-Simplified* achieve clearly sublinear pseudo-regret for all values of $\ell$ considered: this highlights the importance of exploiting the latent-state dynamics in the algorithm design.

The second set of observations is that *LinUCB-Belief-Complex* performs generally better than *LinUCB-Belief-Simplified*. Moreover, for a fixed strategy, the value of $\ell$ is not too influential. That being said, for *LinUCB-Belief-Complex*, the staged variants generally perform slightly (but not significantly) better than the per-round version $\ell = 1$.

**Sensitivity to exploration constants.** Figures 3 and 4 compare the pseudo-regrets of *Plain LinUCB*, *LinUCB-Belief-Complex*, and *LinUCB-Belief-Simplified* over grids of $\ell$ and $C$, using, for each pair $(\ell, C)$, the best regularization parameter $\lambda$ selected in hindsight from its grid. We see that, no matter the stage lengths $\ell$, too large values of $C$ lead to much larger pseudo-regrets, that have a nearly-linear behavior for the initial values of $T$ (and must later exhibit a sublinear behavior). The parameter $C$ is thus critical to tune.

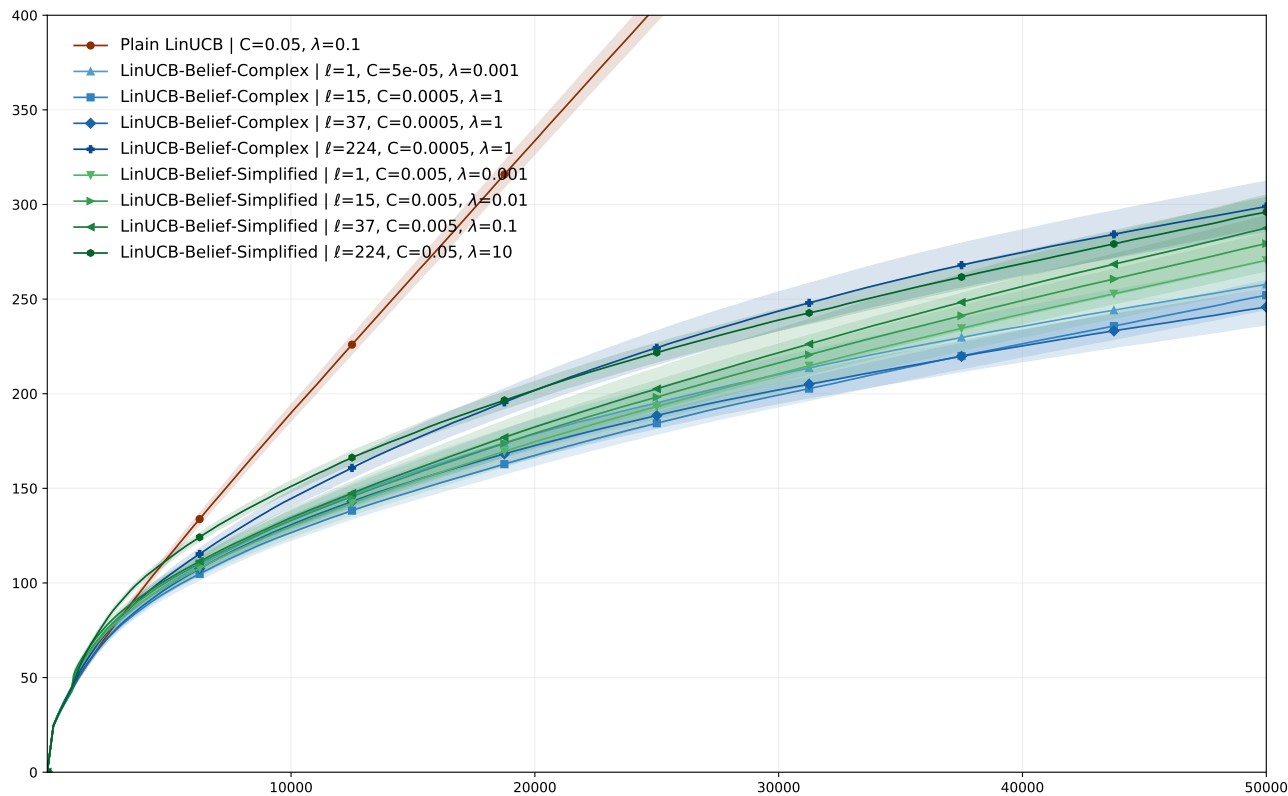

*Figure 2.* Pseudo-regrets averaged over 100 runs. Solid lines correspond to averages and shaded areas to $\pm 2$ standard errors. The figure compares *Plain LinUCB*, *LinUCB-Belief-Complex*, and *LinUCB-Belief-Simplified*, for $\ell \in \{1, 15, 37, 224\}$, using the best exploration constants $C$ and regularization parameters $\lambda$ selected in hindsight from the grids considered.

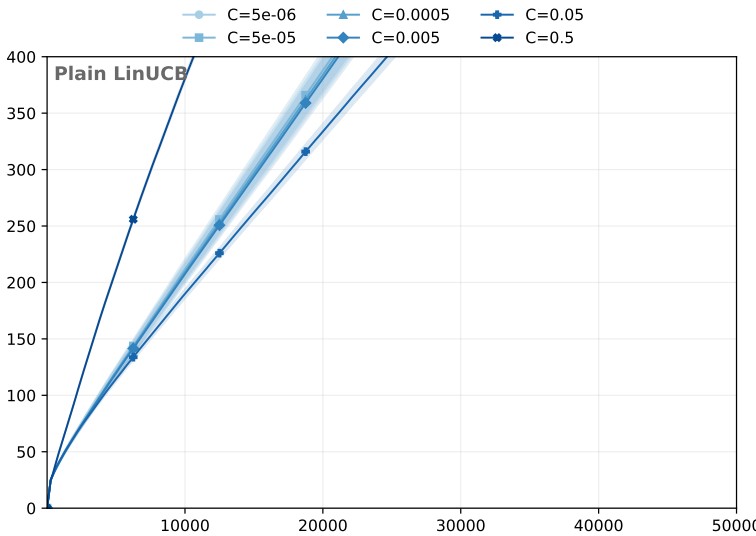

*Figure 3.* Pseudo-regrets averaged over 100 runs. Solid lines correspond to averages and shaded areas to $\pm 2$ standard errors. The figure compares *Plain LinUCB* for different values of $C$, using, for each $C$, the best regularization parameter $\lambda$ selected in hindsight from its grid.

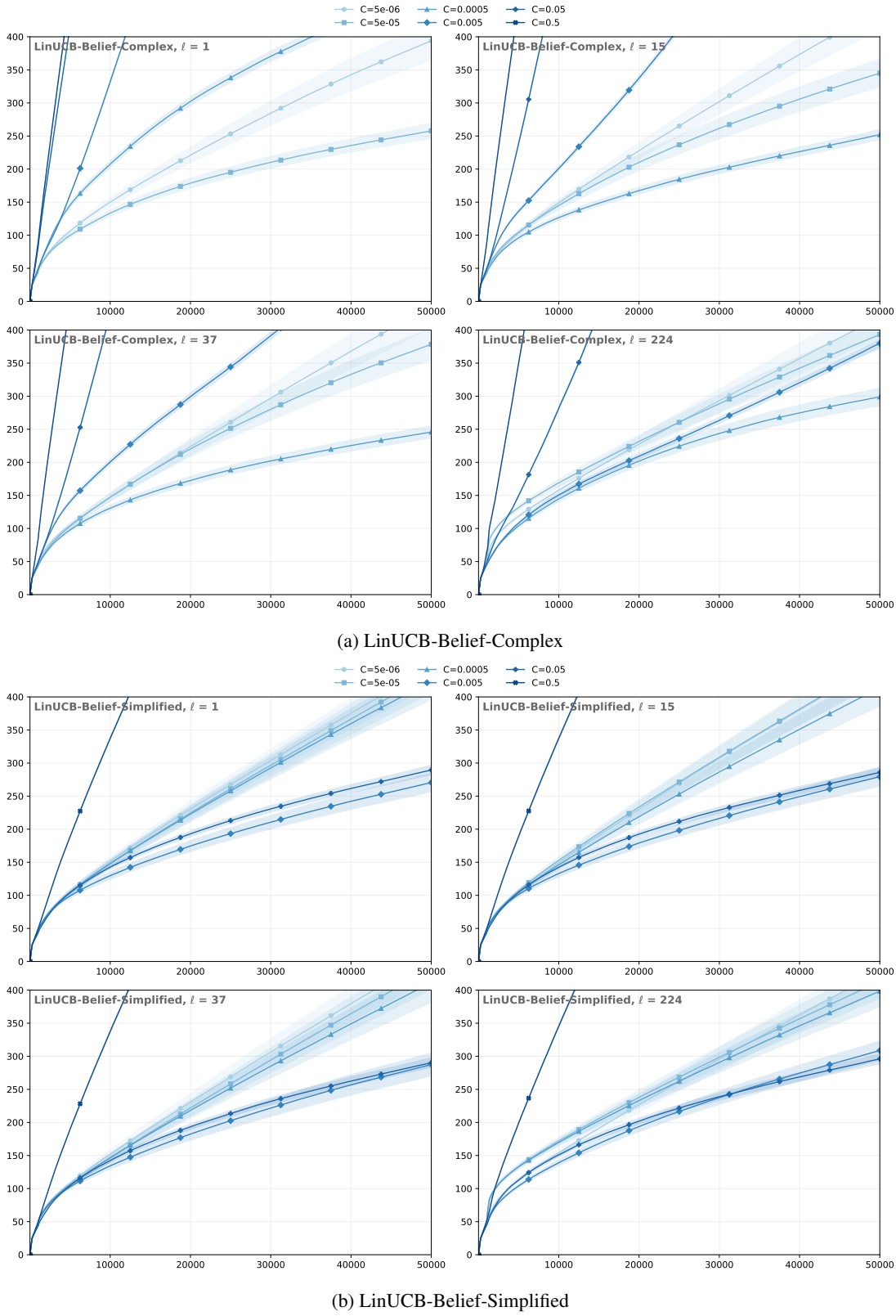

*Figure 4.* Pseudo-regrets averaged over 100 runs. Solid lines correspond to averages and shaded areas to ±2 standard errors. The figures compare LinUCB-Belief-Complex (top graphs) and LinUCB-Belief-Simplified (bottom graphs) for different values of $\ell$ and $C$, using, for each pair $(\ell, C)$, the best regularization parameter $\lambda$ selected in hindsight from its grid.

*Table 4.* Pseudo-regrets of *Plain LinUCB* averaged over 100 runs for each pair $(\lambda, C)$. The table reports averages (with $\pm 2$ standard errors in parentheses).

| | | | *Plain LinUCB* | | | |
|---|---|---|---|---|---|---|
| $\lambda$ / $C$ | 5e-06 | 5e-05 | 0.0005 | 0.005 | 0.05 | 0.5 |
| 1e-09 | 917 (45) | 915 (44) | 911 (43) | 911 (44) | 760 (15) | 1458 (6) |
| 0.001 | 914 (44) | 916 (45) | 909 (43) | 910 (44) | 758 (14) | 1455 (6) |
| 0.01 | 921 (44) | 917 (42) | 915 (43) | 916 (44) | 765 (16) | 1456 (6) |
| 0.1 | 931 (44) | 927 (44) | 918 (43) | 899 (43) | **754 (16)** | 1454 (6) |
| 1 | 930 (41) | 933 (40) | 930 (42) | 909 (39) | 757 (15) | 1453 (6) |
| 10 | 915 (33) | 916 (33) | 920 (32) | 916 (31) | 788 (17) | 1445 (6) |
| 100 | 1224 (52) | 1221 (53) | 1211 (53) | 1179 (51) | 946 (13) | 1403 (6) |

**Detailed results for triplets** $(\ell, C, \lambda)$**.**   Tables 4 to 6 report the detailed pseudo-regret results for each strategy over the full grids of $\ell$, $C$, $\lambda$. What we wanted to check is that the grids of $C$ and $\lambda$ were sufficiently large in the sense that, for each strategy and each value of stage length $\ell$, the best-performing pair $(\lambda, C)$ is not achieved at a boundary of the grids.

**Computational costs; link to the code.**   *LinUCB-Belief-Complex* and *LinUCB-Belief-Simplified* have comparable computational costs in our implementation. For each triplet $(\ell, \lambda, C)$, one series of $N = 100$ runs takes approximately 15 minutes. The full implementation is available at `https://github.com/zhenli1989/bandits_latent_states`.

*Table 5.* Pseudo-regrets of *LinUCB-Belief-Complex* averaged over 100 runs for each triplet $(\ell, \lambda, C)$. The table reports averages (with $\pm 2$ standard errors in parentheses).

| $\ell$ | $\lambda \, / \, C$ | *LinUCB-Belief-Complex* | | | | | |
|---|---|---|---|---|---|---|---|
| | | 5e-6 | 5e-5 | 5e-4 | 5e-3 | 5e-2 | 5e-1 |
| $\ell = 1$ | 1e-09 | 394 (29) | 259 (13) | 521 (7) | 2180 (8) | 4000 (6) | 4614 (6) |
| | 0.001 | 416 (28) | **258 (11)** | 521 (7) | 2181 (8) | 3999 (6) | 4614 (5) |
| | 0.01 | 432 (34) | 259 (11) | 520 (8) | 2182 (8) | 4001 (6) | 4615 (6) |
| | 0.1 | 417 (30) | 262 (13) | 519 (7) | 2179 (8) | 4000 (6) | 4615 (6) |
| | 1 | 490 (31) | 262 (12) | 514 (7) | 2174 (8) | 3999 (6) | 4616 (5) |
| | 10 | 678 (26) | 457 (18) | 494 (6) | 2147 (8) | 4004 (6) | 4635 (5) |
| | 100 | 658 (13) | 650 (11) | 488 (9) | 1965 (7) | 4001 (5) | 4709 (6) |
| $\ell = 15$ | 1e-09 | 443 (29) | 359 (21) | 261 (9) | 1038 (8) | 3041 (7) | 4377 (5) |
| | 0.001 | 448 (27) | 354 (22) | 266 (8) | 1037 (8) | 3039 (7) | 4377 (5) |
| | 0.01 | 443 (31) | 345 (21) | 264 (9) | 1035 (8) | 3039 (7) | 4377 (5) |
| | 0.1 | 473 (33) | 348 (21) | 256 (8) | 1035 (8) | 3039 (7) | 4377 (5) |
| | 1 | 564 (31) | 387 (21) | **252 (8)** | 1028 (8) | 3036 (7) | 4378 (5) |
| | 10 | 678 (24) | 644 (22) | 319 (8) | 997 (8) | 3020 (7) | 4391 (5) |
| | 100 | 657 (13) | 655 (12) | 598 (13) | 842 (7) | 2906 (6) | 4436 (5) |
| $\ell = 37$ | 1e-09 | 448 (31) | 383 (26) | 255 (11) | 754 (8) | 2639 (8) | 4226 (6) |
| | 0.001 | 437 (33) | 378 (25) | 253 (11) | 755 (7) | 2639 (7) | 4226 (6) |
| | 0.01 | 471 (33) | 382 (29) | 255 (11) | 753 (8) | 2639 (7) | 4227 (6) |
| | 0.1 | 450 (28) | 378 (24) | 250 (9) | 753 (7) | 2639 (7) | 4226 (6) |
| | 1 | 556 (32) | 433 (26) | **246 (9)** | 747 (7) | 2634 (7) | 4227 (5) |
| | 10 | 670 (28) | 653 (23) | 367 (11) | 719 (7) | 2613 (7) | 4237 (5) |
| | 100 | 659 (12) | 657 (12) | 637 (11) | 610 (8) | 2459 (7) | 4263 (5) |
| $\ell = 224$ | 1e-09 | 426 (28) | 393 (22) | 323 (15) | 410 (8) | 1842 (9) | 3794 (6) |
| | 0.001 | 420 (30) | 425 (24) | 338 (14) | 417 (8) | 1840 (9) | 3796 (6) |
| | 0.01 | 424 (32) | 410 (29) | 337 (14) | 417 (8) | 1840 (8) | 3795 (6) |
| | 0.1 | 451 (30) | 401 (28) | 305 (16) | 413 (8) | 1839 (9) | 3795 (6) |
| | 1 | 525 (29) | 474 (27) | **299 (13)** | 395 (8) | 1832 (8) | 3794 (6) |
| | 10 | 651 (20) | 655 (21) | 512 (17) | 380 (7) | 1793 (8) | 3789 (6) |
| | 100 | 660 (12) | 659 (12) | 654 (11) | 490 (11) | 1586 (8) | 3745 (5) |

*Table 6.* Pseudo-regrets of *LinUCB-Belief-Simplified* averaged over 100 runs for each triplet $(\ell, \lambda, C)$. The table reports averages (with $\pm 2$ standard errors in parentheses).

| | | | *LinUCB-Belief-Simplified* | | | | |
|---|---|---|---|---|---|---|---|
| $\ell$ | $\lambda / C$ | 5e-6 | 5e-5 | 5e-4 | 5e-3 | 5e-2 | 5e-1 |
| | 1e-09 | 451 (30) | 462 (32) | 430 (30) | 279 (14) | 318 (7) | 1508 (9) |
| | 0.001 | 446 (29) | 455 (29) | 424 (28) | **271 (14)** | 318 (8) | 1505 (9) |
| | 0.01 | 455 (30) | 436 (29) | 435 (31) | 284 (13) | 317 (8) | 1504 (9) |
| $\ell = 1$ | 0.1 | 513 (34) | 509 (37) | 441 (30) | 276 (14) | 316 (8) | 1504 (8) |
| | 1 | 571 (35) | 568 (32) | 521 (29) | 288 (15) | 311 (8) | 1499 (9) |
| | 10 | 688 (26) | 688 (25) | 680 (27) | 536 (20) | 289 (7) | 1464 (9) |
| | 100 | 659 (13) | 659 (13) | 658 (13) | 651 (11) | 430 (12) | 1249 (8) |
| | 1e-09 | 453 (31) | 459 (30) | 414 (28) | 280 (13) | 320 (8) | 1506 (9) |
| | 0.001 | 452 (28) | 453 (30) | 418 (30) | 285 (15) | 321 (8) | 1506 (8) |
| | 0.01 | 483 (29) | 475 (31) | 432 (30) | **279 (15)** | 319 (8) | 1508 (9) |
| $\ell = 15$ | 0.1 | 535 (38) | 495 (38) | 443 (33) | 287 (17) | 319 (8) | 1506 (9) |
| | 1 | 571 (32) | 575 (33) | 514 (28) | 287 (16) | 311 (8) | 1499 (9) |
| | 10 | 679 (25) | 682 (24) | 673 (24) | 524 (22) | 286 (7) | 1464 (8) |
| | 100 | 657 (12) | 657 (12) | 657 (12) | 651 (10) | 426 (12) | 1247 (8) |
| | 1e-09 | 452 (34) | 440 (31) | 419 (31) | 291 (17) | 322 (8) | 1509 (9) |
| | 0.001 | 459 (34) | 433 (32) | 411 (30) | 290 (16) | 325 (8) | 1510 (9) |
| | 0.01 | 476 (30) | 451 (33) | 419 (32) | 291 (16) | 323 (8) | 1508 (9) |
| $\ell = 37$ | 0.1 | 506 (30) | 481 (31) | 424 (29) | **288 (17)** | 320 (8) | 1508 (8) |
| | 1 | 567 (35) | 560 (33) | 510 (34) | 295 (19) | 311 (7) | 1503 (9) |
| | 10 | 670 (31) | 669 (25) | 662 (23) | 532 (19) | 290 (7) | 1465 (8) |
| | 100 | 659 (12) | 659 (12) | 657 (12) | 648 (10) | 428 (13) | 1247 (8) |
| | 1e-09 | 437 (27) | 413 (26) | 398 (24) | 320 (15) | 343 (9) | 1528 (9) |
| | 0.001 | 428 (29) | 434 (27) | 417 (22) | 340 (18) | 347 (9) | 1528 (9) |
| | 0.01 | 455 (29) | 432 (31) | 430 (24) | 339 (17) | 347 (9) | 1529 (9) |
| $\ell = 224$ | 0.1 | 496 (33) | 456 (35) | 403 (27) | 330 (16) | 346 (9) | 1528 (9) |
| | 1 | 529 (28) | 524 (30) | 457 (25) | 309 (14) | 337 (9) | 1521 (9) |
| | 10 | 652 (21) | 649 (21) | 651 (21) | 525 (20) | **296 (8)** | 1482 (9) |
| | 100 | 660 (12) | 659 (12) | 660 (12) | 652 (11) | 437 (12) | 1252 (8) |

