# OpenReview forum: "A Direct Approach for Handling Contextual Bandits with Latent State Dynamics"
_ICML.cc/2026/Conference — ICML 2026 regular_

### Official Review · Reviewer_BT6j · 2026-03-08

**Soundness:** 3
**Presentation:** 3
**Significance:** 3
**Originality:** 3
**Overall Recommendation:** 4
**Confidence:** 3

**Summary:**

The main problem addressed is a contextual bandit environment in which unobserved regimes evolve according to a (hidden) Markov model and influence both context distributions and rewards. There’s an emphasis made comparing against another paper (Nelson et al (2022)) which assumes rewards depend on posterior beliefs and with that the model reduces to a linear bandit. In this paper, rewards are modeled depending directly on the underlying regime, which better reflects real-world applications. It is shown that the O(T^1/2) regret from the other paper comes from belief-linearization enabling a reduction to linear bandits. A staged UCB algorithm is presented that estimates latent-state beliefs and reward parameters online, and sub-linear regret is proven under some (mild) conditions. An important element in this is the belief estimation based on a carefully constructed filtration.The approach allows arbitrary regime switching.  Experiments on data derived from a credit-card dataset demonstrate this consistent sub-linear regret and improved performance.

**Compliance With Llm Reviewing Policy:**

Affirmed.

**Final Justification:**

After considering the author rebuttal and the reviewer discussion, my overall assessment has slightly improved, and I lean more clearly toward acceptance.

The rebuttal adequately addressed several of my concerns. In particular, the authors clarified the computational setup and the scope of the experiments, and provided a more convincing justification for their choice of baselines, including the plan to incorporate a more directly comparable latent-model baseline. I also appreciate the clarification regarding the regret bounds and the trade-off between generality and stronger assumptions.

My main positive assessment remains unchanged: the paper presents a technically solid and conceptually clear treatment of contextual bandits with latent Markovian regimes, and provides a meaningful advance over prior work such as Nelson et al. by avoiding belief-linearization and introducing a novel analysis based on filtration and HMM forgetting. I believe these ideas are likely to be of wider interest.

Some weaknesses remain, in particular the relatively limited empirical evaluation and the still somewhat unclear practical implications of the approach, as well as the fact that the regret guarantees are weaker than in more restrictive settings. However, I do not consider these issues to be blocking.

So overall, I would be comfortable recommending acceptance.

**Key Questions For Authors:**

- What compute would be needed for the current experiments, and is it reasonable to run either larger experiments or against a benchmark that does include a latent-model inside?
- Did you look into using stronger assumptions to get to stronger regret-bounds, or was the focus on this relatively general problem definition?

**Limitations:**

Yes, these are adequately discussed.

**Strengths And Weaknesses:**

- Strengths:
- Conceptually clear paper that shows why the reduction in Nelson et al. works, and also why it fails in this more general setting. This is really another way of studying the problem, without the belief-linearization, which breaks some earlier tools used.
- The technical set-up is interesting and well developed with interesting ideas. Especially the belief estimation using the filtration designed and HMM forgetting properties works well. The reward model is natural and logical to use. On the models some relevant restrictions are necessary to come to the proofs. Feels like there’s been a trade-off towards generality over sharper bounds (for instance bounded second moments instead of sub-Gaussian). Staged UCB set-up works well.
- Probably would say moderately significant right now, it’s an interesting new way to define the problem, but right now not clear yet (also because of the assumptions used in this analysis) if this remains theoretical or also develops practically.
- Weaknesses
- Simulation is relatively limited: unclear how compute-heavy and with that useful this would be in the end (also stated in the paper ‘constrained by computational resources’), benchmark could be more relevant through something closer to Nelson et al. (2022) comparison - now it’s a benchmark against a model that doesn’t use any latent model.
- Also acknowledged from the paper, the regret bound is worse than classical results, and if possible it would strengthen the contribution to better discuss whether the regret bound reflects an inherent difficulty of the state-dependent setting or comes primarily from the analysis tools used here.
- The introduction’s framing around Nelson et al. makes the contribution feel reactive rather than proactively positioned as a broader modeling advance. section 3 reads really as a comparison, and then also shows what is new, while this is not really the case and is really a wider type of model. It could be presented earlier more as a more general new work, and with that also emphasise the main contribution (which I think is the setup of the filtration and then using the HMM forgetting properties to control correlations). Section 5 is now a bit heavy in theory and could benefit from some intuition before diving into the formal statements to guide reading.

---

> ### Author Rebuttal · Authors · 2026-03-28
>
> We thank the reviewer for the time taken in evaluating the submission.
>
>
> **On not positioning the submission as a broader modeling advance**
>
> The current exposition was picked to keep a modest and humble tone acknowledging the earlier work by Nelson et al., 2022. However, we happily read in all reviews that reviewers actually share what we thought deep inside, i.e., that our work really goes beyond the simplified model by Nelson et al. We agree with the exposition and summary suggested here and would completely restructure the submission accordingly, as indicated in the answer to Reviewer PX2C. It should consist of heavy copy-pastes with some easy adaptations.
>
>
> **On regret bounds**
>
> Yes, we have a probably not unreasonable assumption to get a $T^{3/4}$ rate also in the most complex reward model---and this $T^{3/4}$ looks like a rate that can be well explained, given the analysis of Appendix E for the simplified model, with a typical $T^{1/2}$  being worsened to a $T^{3/4}$ rate therein because of the parametric (thus at $\sqrt{T}$ rate) estimation of the HMM parameters.
> This assumption would be to assume that the smallest eigenvalue of the Gram matrix $G_t$ grows linearly with $t$, instead of only exploiting the $\geq \lambda$ lower bound. A similar assumption is present in Nelson et al. (but we do not need it in Appendix E to replicate and extend their results in the simplified reward model).
> That being said,  our focus was Indeed to remain relatively general.
>
>
> **On simulations and practical devlopments**
>
> For practical relevance: see our answer to Reviewer PX2C.
>
> For simulations details:
>
> 1. Like much of the bandits literature, our paper is primarily theoretical, and the experiments are intented mainly to illustrate the practical behavior of the proposed method, specifically, its convergence and its performance relative to relevant baselines. For this reason, we conducted the experiment on a modest setup (8 cores, 16 threads and 32GB RAM) without using a GPU. That being said, the experiments are still carried out on a simulated but realistic banking credit marketing setting designed to demonstrate the practical applicability of the algorithm. In particular, we consider 2 hidden states and 20 observations for HMM emission matrix. By comparison, Nelson et al. consider a smaller simulated setting with 2 hidden states and 4 observations in the finite-context case. Similar small emission matrices are also used by Zhou et al. 2021 and Azizzadenesheli et al. 2016 (both cited in our submission).
>
> 2. Regarding the baselines, we so far compared against classical contextual linear bandit algorithms because the most directly related latent-state bandit models do not incorporate contextual information into the reward model. This includes, for example, Zhou et al 2021,  Azizzadenesheli et al 2016, Nelson et al. 2022 (in their original form). As a result, standard contextual linear bandits provided the most direct available baseline.
>
> 3. That being said, we agree that the Box B strategy (actually a contextual extension of Nelson et al., 2022) is a meaningful additional baseline, that corresponds to using the Box A strategy with $\ell = 1$. We will add its performance.

---

> > ### Author Rebuttal · Reviewer_BT6j · 2026-04-03
> >
> > Thanks you for this further, and clear, explanation, this completely clarifies the questions I had.

---

### Official Review · Reviewer_wEYb · 2026-03-12

**Soundness:** 3
**Presentation:** 4
**Significance:** 3
**Originality:** 3
**Overall Recommendation:** 4
**Confidence:** 2

**Summary:**

This paper studies contextual bandits in environments where rewards and contexts are influenced by latent states that evolve according to a hidden Markov model (HMM). Prior work (Nelson et al., 2022) reduces this setting to a linear bandit problem by modeling rewards as linear functions of posterior state probabilities and assuming the HMM parameters are known. In contrast, this paper proposes a direct formulation where rewards depend on the true hidden states.

**Compliance With Llm Reviewing Policy:**

Affirmed.

**Final Justification:**

I keep my moderate positive score

**Key Questions For Authors:**

NA

**Limitations:**

yes (see Sec. 6)

**Strengths And Weaknesses:**

Pros:
The paper is well-written and easy to follow, with a clear problem formulation and a logical presentation of the proposed approach. The methodology and theoretical analysis appear to be solid, and the arguments are presented in a structured and convincing manner. Overall, the work seems technically sound and accessible to readers.

Cons:
I don't find any major weaknesses. However, I am not a domain expert in this specific line of research, so I will defer to the comments and identified weaknesses from other reviewers.

---

> ### Author Rebuttal · Authors · 2026-03-28
>
> We thank the reviewer for the time taken in evaluating the submission.
>
> Indeed, the other reviewers raised extremely interesting and valid points, which we hope to have answered to eveyrone's satistfaction. We refer to the answers written for each of them.

---

> > ### Author Rebuttal · Reviewer_wEYb · 2026-04-01
> >
> > Thank you for your response. I will follow up on the discussion with the other reviewers.

---

### Official Review · Reviewer_8dNF · 2026-03-13

**Soundness:** 2
**Presentation:** 1
**Significance:** 4
**Originality:** 4
**Overall Recommendation:** 4
**Confidence:** 3

**Summary:**

The paper studies linear contextual bandits with an HMM that determines the distribution over contexts and the true parameters.
By combining the UCB approach and HMM parameter estimation, the cumulative regret bound of $\tilde{\mathcal{O}}(T^{7/8})$ is achieved.
The algorithm first estimates the hidden state using the spectral method and relying on the forgetting property assumption of the HMM. Then, it computes the upper confidence bound based on the estimation, addressing uncertainty from both the state and the parameter.

**Compliance With Llm Reviewing Policy:**

Affirmed.

**Final Justification:**

I have mixed feelings about this submission. In summary, I find this work novel and interesting, but it fails to effectively communicate its merits and validity. As my main technical concern is resolved, I have raised my score.

---

The main issue was whether estimating the raw HMM parameters is possible.
The authors proposed a fix for the potential permutations of states, but the authors' justification of whether permutation is the only transformation that needs addressing added more confusion.
Propositions C.3 and C.4, which they referred to, do not include any statement about permutations, and the fact that the norm is bounded does not address the issue.

What resolved my concern was finding a sentence in Azizzadenesheli et al. (2016) myself which explicitly states that the estimation is good up to permutations.
This resolved my concern, but given that the authors' responses consist of a mix of relevant and irrelevant facts about the issue, I am uncertain whether the authors are properly aware of the potential pitfalls.
I add more details about the matter at the end in case there was some misunderstanding.

I have remaining concern about the presentation as well. There are many sentences that are unnecessarily complicated and hard to parse on the first read. At least to me, these appear to only (and possibly overly) focused on factual correctness rather than readability. It is not really ground for rejection, but I believe there is room for improvement.

---

Let me additionally describe my concern regarding the estimation of HMM parameters here.
Suppose there are two HMMs whose states are a permutation of each other and denote the permutation by $\sigma$.
Suppose $b$ and $b'$ are belief vectors of the two HMMs respectively at some time.
Then, they have the relationship of $\sigma b = b'$.
The two HMMs have the exact same observation distribution, hence any algorithm would output the same vector $\hat{b}$ for both HMMs.
More precisely, the output would have the same distribution because the distribution of the input is the same.
However, if $\lVert b - \hat{b} \rVert_1$ is small, then it is highly unlikely that $\lVert b' - \hat{b} \rVert_ 1 = \lVert \sigma b - \hat{b} \rVert_1$ is also small, which is why I think the current statements of Propositions C.3 and C.4 can not be true: if they are true for one HMM, then they are not true for another one.

The authors emphasized that the guarantee is an upper bound on the norm in both responds, but that does not help because only $b'$ is permuted and $\hat{b}$ is not.
The authors' claim that the guarantee is invariant of the state indexation gives the impression that they believe the spectral method outputs two different vectors $\hat{b}$ and $\sigma \hat{b}$ for the two HMMs, which cannot be true.

Again, this issue is resolved by reading Azizzadenesheli et al. (2016), where it explicitly mentions that there exists an unknown permutation $\sigma'$ such that $\lVert b - \sigma' \hat{b}\rVert_1$ is small, and I am highlighting this to ensure the authors and I share a common understanding.

**Key Questions For Authors:**

1. (From Weakness 1) Could the authors confirm that Assumption 4.1 and Lemma 4.3 are correctly stated? I don't think learning the vector $b_ t$ itself is possible.

2. (From Weakness 2) Could the authors provide more intuition or justification for the definition of the regret? In addition, could the authors provide more details about Remark 2.4? Remark 2.4 mentions that the second sum in Eq. (2) is close to the actual sum of the rewards, but if "up to a factor of the same order of magnitude" means within a constant factor of each other, then the actual regret could scale linearly in $T$. On the contrary, if they differ only by some additive terms, then Eq. (2) would make sense.

**Limitations:**

yes

**Strengths And Weaknesses:**

Strengths
1. The assumed setting is novel and realistic. It carries significance in practice.
2. The method of combining the state-estimation with UCB is original.


Weaknesses

1. I think there are some missing details in Assumption 4.1 and Lemma 4.3. If the learner has no information about $(M, \nu)$, I don't see how it can approximate the belief $b_ t$ when permutations of states are not distinguishable. Specifically, if one makes a new HMM by permuting $M$ and $\nu$ by the same permutation of states, then the sequences of $x_ t$ that the two HMMs emit have the same distribution. The learning target $b_ t$ will also get permuted, so it seems impossible to learn $b_ t$ with respect to the $\ell_ 1$ norm when its permutated versions would not be distinguishable. The validity of Assumption 4.1 is currently my main concern.

2. I think the choice of regret as in Eq. (2) requires more justification, especially the second sum.
I interpreted the benchmark term $\sum_ {t=1}^T \max_ {a \in \mathcal{A}} \sum_ {h \in [H]} b_ t(h) \phi(a, x_ t)^\top \theta_ h^\ast$ as the reward of an agent that is fully aware of the HMM parameters but does not observe or use the reward signal.
This agent would maximize $\mathbb{E}[ r_ t(a) \mid x_ {1:t}]$.
However, the second term $\sum_ {h \in [H]} b_ t(h) \phi(a_ t, x_ t)^\top \theta_ h^\ast$ does not admit the same interpretation.
As mentioned in the paper, $a_ t$ is not $\sigma(x_ {1:t})$ measurable, so this term is not $\mathbb{E}[r_ t \mid x_ {1:t}]$.
Hence, I am uncertain whether it is correct to call this value pseudo-regret.
Ironically, overcoming this discrepancy is presented as the main challenge, which further necessitates justifying the definition.

3. The presentation is very hard to follow due to the structure of the paper and the style of writing.

- There are a lot of run-on sentences frequently interrupted by commas, parentheses, and colons. Some instances include the second and third sentences in the abstract, the first sentences in Section 2, Section 3, and Appendix A, respectively, and the second sentences in Section 3.3 and Appendix C, respectively. It is quite hard to parse these sentences on the first read.

- The style of discussion in this paper is incoherent. I believe the authors were trying to add more details, but sometimes it distracts the reader from the main flow. Taking Section 3 as an example, the main challenge is presented before explaining how this paper or Nelson et al. (2022) tackle the problem, so it is quite confusing to understand why they are challenges and in which sense they are overcome. I am still uncertain whether introducing reward models (3) and (4) was necessary and why $a_ t$ must be $\mathcal{U}_ t$-measurable (deploying a randomized algorithm would make $a_ t$ not even $\mathcal{F}_ t^{\mathrm{obs}}$-measurable).

4. The proposed bound $\tilde{\mathcal{O}}(T^{7/8})$ is quite large, and there is no justification or a lower bound as to whether this is a tight or at least a sensible bound. It might be helpful if the authors could explain what causes this dependency. I would like to note that I find this weakness relatively minor considering that this paper addresses this problem for the first time.

---

> ### Author Rebuttal · Authors · 2026-03-28
>
> We thank the reviewer for the time taken in evaluating the submission and for the careful and thoughtful comments and questions on the maths!
>
> **Weakness 1: On missing details in Assumption 4.1 and Lemma 4.3**
>
> Yes, we had overlooked the fact that spectral methods only provide estimates that are good up to permutations; since the evaluation criteria thereof are in norms, this does not pose a problem if the target is only to estimate the HMM parameters or the beliefs in these norms.
>
> However, our algorithm estimates the reward parameters $\boldsymbol{\theta}^\star_h$ in a way that implicitly requires the tracking of the specific hidden states; put differently, even if the hidden-state labels are only notational, this labeling needs to be consistent at least after a certain round for our algorith to be meaningful. We thank the reviewer for pointing out that an argument was therefore missing.
>
> Fortunately, that missing argument is rather elementary (and we will of course incorporate it). Namely, Assumption 4.2 implies, in particular, that all emission distributions $\nu_h$ are different and thus are separated: we denote $\Delta = \min_{h \ne h'} \Arrowvert \nu_h - \nu_{h'} \Arrowvert > 0$. Based on that and on the bound of Proposition C.3, we can show that after a constant time of the order $1/\Delta^2$, the estimates $\hat\nu_{t,h}$ are $\Delta/4$--close to their true values $\nu_h$ and $3\Delta/4$ separated from any other distribution $\nu_{h'}$. Therefore, by triangle inequalities, at a given step $t+1$, when obtaining the estimates $\hat\nu_{t+1,\rho(h)}$, where $\rho$ is some permutation of the hidden states, we can perform an alignment to the ordering used in round $t$, i.e., identify $\rho^{-1}$, by looking at which permutation $\pi$ of $[H]$ minimizes the norm between $\hat\nu_{t+1,\pi(\rho(h))}$ and $\hat\nu_{t}$, and then 'align' states with this $\pi$, which is unique and equals $\rho^{-1}$. We may then show the desired consistency of the labels after a constant time of order $1/\Delta^2$, under the conditions of Proposition C.3., which is sufficient for the rest of the analysis.
>
> **Weakness 2: Regret definition**
>
> Thanks for giving us an opportunity to expand on Remark 2.4.  We considered this specific notion of pseudo-regret because this was the one also considered by Nelson et al. However, Remark 2.4 indicates that the second term $\sum_t \sum_{h \in [H]} \boldsymbol{b}_t(h) \varphi(a_t, x_t)^\top \theta_h^\star$ is close, up to a factor of order $T^{7/8}$ (not a constant factor, not a $T$ factor) of the true rewards $\sum_t r_t(a_t)$. The argument to show so is similar to the arguments used in Appendix D:
>
> 1. By Hoeffding-Azuma, with the filtration $\cal F_t^{\mathrm{all}}$, we have that $\sum_t r_t(a_t)$ is close to $\sum_t \varphi(a_t, x_t)^\top \theta_{h_t}$, i.e., we get rid of the noise terms $\eta_t(a_t)$
>
> 2. Then, as in Lemma D.2, we can show that $\sum_t \varphi(a_t, x_t)^\top \theta_{h_t}$ is close to $\sum_t \sum_{h \in [H]} \bar b_t(h) \varphi(a_t, x_t)^\top \theta_{h}$
>
> 3. Finally, as in Lemmas D.4-D.5, we have that $\sum_t \sum_{h \in [H]} \bar b_t(h) \varphi(a_t, x_t)^\top \theta_{h}$ is close to $\sum_t \sum_{h \in [H]} b_t(h) \varphi(a_t, x_t)^\top \theta_{h}$, which is the cumulative pseudo-reward considered in the definition of regret
>
> We will of course add these details as an additional subsection at the end of Appendix D, with a pointer in Remark 2.4 (and a rewriting of the latter to state more explicitly how the two objectives differ: by a quantity of the order $T^{7/8}$.
>
> **On the presentation**
>
> We acknowledge the shortcomings on presentation and explain, in our answer to Reviewer PX2C how we will address them. The bottom line was that we were taking an humble and modest tone while, as we thought deep inside and as all reviewers concur, our contributions are significant enough for the exposition to depart from the preliminary results in Nelson et al.
>
> **On the measurability of the $a_t$**
>
> If algorithms are randomized (which we do not need), it suffices to put the external randomizations they use in $\cal U_t$. However, note that even in the non-randomized case, the $a_t$ are not measurable w.r.t. $\cal F_ t^{\mathrm{obs}}$ (as they also depend on rewards, which themselves depend on hidden states). These were exactly the kind of challenges we had to overcome. But we agree that discussing these complex statistical dependencies later in the article (as promised in our answer to Reviewer PX2C) will improve the flow of exposition.
>
> **Explanations for the $T^{7/8}$ order of magnitude**
>
> We actually have explanations for that and also a possibility to improve them into a $T^{3/4}$ rate, see our answer to Reviewer BT6j.

---

> > ### Author Rebuttal · Reviewer_8dNF · 2026-04-01
> >
> > I deeply appreciate the author's response to my questions. While they mostly addressed my concerns, I have a few additional questions related to my original ones.
> >
> > ---
> >
> > **Weakness 1** The authors provided how to handle the learned vector and matrix if they are permuted. While I find the method valid, I feel that the 'if they are permuted' assumption is not justified, possibly because I am not familiar with the spectal method for HMMs. What is the exact guarantee of the spectal methods, and where can I find it? Is it guaranteed that there exists a permutation such that the learned vector and the matrix are close to the true parameters? It might be the case that the learned vector and matrix are correct up to some linear transformation, not necessarily a permutation, which would invalidate the proposed method. As mentioned, this is my main concern and is not fully resolved.
> >
> > **Weakness 2** Thank you for the explanation. The details the authors added about Remark 2.4 convinced me that the provided guarantees are actually meaningful. However, the authors did not address whether Eq. (2) is the right definition to consider, other than mentioning that it is inherited from Nelson et al. (2022). The definition makes sense in Nelson et al. (2022), but does not reflect the changed setting in this paper, and I am still not convinced whether one can call it "pseudo-regret". It might make more sense if the authors redefine the regret using the actual rewards and bound it by combining what is presented in the manuscript and what is explained in the rebuttal.
> >
> > **Measurability of $a_t$** Although I do not think this is a serious issue, the authors' response to this part added more confusion. My question was *why* $a_t$ must be $\mathcal{U}_t$-measurable, but the rebuttal only says that the analysis is challenging *because* $a_t$ must be $\mathcal{U}_t$-measurable. Could the authors explain why?
> > In addition, I could not understand claim that even if $a_t$ is chosen deterministically, it is not $\mathcal{F}_t^{\mathrm{obs}}$-measurable. $\mathcal{F}_t^{\mathrm{obs}}$ contains all information that the agent can utilize to choose $a_t$. The manuscript also states that $a_t \in \mathcal{U}_t \subset \mathcal{F}_t^{\mathrm{obs}}$.
> > Could the authors clarify this part as well?

---

> > > ### Author Response · Authors · 2026-04-02
> > >
> > > We thank again the reviewers for the follow up questions, and here are the questions:
> > >
> > > **Weakness 1**
> > >
> > > The reviewer wonders 'what is the exact guarantees of the spectal methods', and asks for specific pointers; more generally, we feel that the reviewer would like to make sure that beliefs and HMM parameters can indeed be estimated up to permutations of the hidden states, and would like to get a sense of the corresponding results in the literature.
> > >
> > > We summarized these in Propositions C.3 and C.4 (buried deep in the appendices of our submission, admittedly). All guarantees are 'global' and independent of the indexations of states: transition matrices, emission distributions, and beliefs are estimated in norms (Euclidian, or total-variation distance). Such criteria are invariant under permutations but would be affected by, e.g., linear transformations; permutations seems the only way of preserving these criteria and we acknowledge the initial shortcoming in not keeping track over time of a consistent labeling.
> > >
> > > Meanwhile, we noted that the article by Azizzadenesheli et al. 2016 (see Step 3 of the proof of Lemma 8) already sketched the alignment procedure we mentioned earlier to solve the 'up to permutations' issue.
> > >
> > > **Weakness 2**
> > >
> > > Right, we had not come back to the question of whether or not the pseudo-regret of Equation (2) is meaningful.
> > >
> > > Actually, as the reviewer points out, this is a void question as being able to consider the true regret (with actual payoffs, as indicated in Remark 2.4 and justified in the previous answer) is more important and makes the consideration of any pseudo-regret void. We agree with that and will rewrite this section to deal with the true regret (and add a short section in Appendix E to provide the details of Remark 2.4).
> > >
> > > The behind-the-scene reason for the writing of Section 2.4 is that we realized only 2 days before the submission that the actual regret could be controlled and we opted for a cheap way to acknowledge that.
> > >
> > > **Measurability of $a_t$**
> > >
> > > First, there was a very unfortunate and very confusing typo in our answer; we meant 'However, note that even in the non-randomized case, the $a_t$ are not measurable w.r.t. $\cal F^{\mathrm{ctx}} = \sigma(x_{1:t})$ (as they also depend on rewards, which themselves depend on hidden states).' The confusion on our end comes from the fact that the 'obs' filtration was relative to contexts till late before the submission and we had forgotten that we had changed the names of several filtrations. Again, sorry sorry sorry for the confusion.
> > >
> > > Now, to get back to the inital question: we did not want to say that $a_t$ must be $\cal U_t$-measurable, but it took us a long time to identify that making sure that it is $\cal U_t$-measurable (i.e., working in stages) was super convenient to solve several issues we faced in the analysis and that we tried to detail in Section 3.2. We will rework the wording therein to make sure that we do not convey the message of a mandatory $\cal U_t$-measurability, but rather say that it is handy.

---

### Official Review · Reviewer_PX2C · 2026-03-14

**Soundness:** 3
**Presentation:** 2
**Significance:** 3
**Originality:** 3
**Overall Recommendation:** 3
**Confidence:** 3

**Summary:**

The authors of the paper handle linear contextual bandits where the contexts move via a hidden Markov model with a fairly general reward model that generalizes various well-studied problem settings, including linear bandits, contextual bandits, and an earlier studied version with simpler reward structures by Nelson et al. 2022. The regret baseline is based on beliefs about the latent states.

**Compliance With Llm Reviewing Policy:**

Affirmed.

**Key Questions For Authors:**

Please see my comments before.

**Limitations:**

Yes

**Strengths And Weaknesses:**

The belief estimation module for HMM is basically imported from existing literature and it fits the problem at hand. This step incurs an additional regret cost that is unavoidable in this context without assuming knowledge of the Markov model behind latent states.

I found the use of an intermediate $\sigma$-algebra $\cal U_t$ and fast mixing of the latent states to be an interesting approach to avoid the issues in estimation arising from various statistical dependencies that the reward structure introduces.

It is imperative that the authors describe some practical scenarios where such a complex reward structure is relevant. It is indeed a generalization of many reward models and the analysis stands as one that encompasses all. However, strictly from a contribution standpoint, it would be useful to understand where this model is useful.

I found the description of the paper a bit too "high-level", even though the arguments are logical. The logic is less line-by-line readable in my opinion, and it requires an intimate familiarity with Nelson et al., 2022 to parse. It would be useful if the authors could disentangle this work from the aforementioned paper. A lot of real-estate is spent in discussing the belief estimation and how the problem considered subsumes earlier models etc. The reader has to wait for a long time to get to the algorithm. It remains nebulous how the regret breaks down into the various terms that are in turn bounded. Each module within the sections in the paper, as it stands, is described well, but I had to collate information from the various parts of the paper to comprehend the algorithm; the regret analysis is still somewhat nebulous, despite being fairly familiar with proof techniques for bandits.

---

> ### Author Rebuttal · Authors · 2026-03-28
>
> We thank the reviewer for the time taken in evaluating the submission.
>
> **On practical scenarios**
>
> Indeed, we should have motivated the extension in the introduction. If one is ready to believe that 'ordinary' linear contextual bandits are a setting of practical interest, then our setting is basically the extension of this to cases where the environment can be in different states. For instance, in economic problems, there could be finitely many underlying states like 'crisis', 'steady', 'growth' and the rewards, as well as the contexts, would directly depend on that underlying state. We take in our simulation a credit example, in which both rewards and contexts like revenues of applicants heavily depend on the state of Economy.
>
> **Too high-level exposition**
>
> We understand the issue. The current exposition was picked to keep a modest and humble tone acknowledging the earlier work by Nelson et al., 2022. However, we happily read in all reviews that reviewers actually share what we thought deep inside, i.e., that our work really goes beyond the simplified model by Nelson et al. Given that, we would now suggest to change the exposition as follows:
> - Introduce in Section 2 our HMM-based setting, with rewards depending on the hidden states
> - Have a super-brief Section 3 stating the simplified setting of Nelson et al., with rewards depending on beliefs, but deferring all (technical) discussions of challenges overcome to the end of the main body
> - In Section 4, note that the strategy we introduced may handle the simplified setting of Nelson et al when taking $\ell = 1$, i.e., not considering stages (where we recall that we actually extend the original setting of Nelson et al. 2022 to context-dependent rewards)
> - In Section 5, first state the $T^{3/4}$ high-probability bound obtained in the (extension of the) simplified setting of Nelson et al., referring to the beginning of the appendices (see below)
> - In Section 5, then state the fast-forgetting condition (only used for the most complex reward model), the $T^{7/8}$ regret bound, and sketch its proof, with a discussion of the challenges overcome
> - In terms of appendices, we would rather first prove the $T^{3/4}$ bound for the simplified reward model (it should take 3 self-contained pages) and then only provide the proofs for the most complex reward model, rather than the other way round; this would help readers to identify better what parts need to be changed. Currently, that simplified proof comes after the proof for the most complex case and we agree that it would be better the other way round.
> Long story made short, we have clear ideas on how to restructure the submission to effectively disentangle our contributions from the earlier one by Nelson et al. This should be achieved with some heavy copy-pastes together with some easy adaptations.

---

> > ### Author Rebuttal · Reviewer_PX2C · 2026-04-01
> >
> > The proposed changes will fully address my concerns.

---

### Decision · Program_Chairs · 2026-04-30

**Decision:**

Accept (regular)

**Comment:**

The submission propose a rather novel solution to contextual finite armed bandits, where the reward function is governed by a hidden state that evolves according to a latent Markov chain. The authors did a good job in differentiating from Nelson et al. (2023). To solve the proposed problem, the authors need to estimate the beliefs and correspondingly construct the UCBs on the latent reward vectors, which depend on the hidden state. The corresponding technical tools involve the consideration of the filtration ${\cal U}_t$ and the UCB constructed in Line 4 in Box A (and also Theorem 5.2). These tools are quite novel, and the proposed techniques shall find interesting use in other online learning problem with hidden state transition.

Nevertheless, I urge the authors to provide full clarity on the ``up to permutation'' estimation guarantee on the HMM parameters in the revision of the paper, and to extract the discussions from Azizzadenesheli et al. (2016, Step 3 of the proof of Lemma 8) rather than just citing it.

Minor comment: For transpose (for example in equation (4)), please use $\top$ instead of $T$. In addition, Line 197L: typo: may be be estimated.